# A Unified Principle of Pessimism for Offline Reinforcement Learning under Model Mismatch

**Yue Wang**
Department of Electrical and Computer Engineering
University of Central Florida
Orlando, FL, 32816
`yue.wang@ucf.edu`

**Zhongchang Sun**
Department of Electrical Engineering
University at Buffalo
Buffalo, NY, 14260
`zhongcha@buffalo.edu`

**Shaofeng Zou**
School of Electrical, Computer and Energy Engineering
Arizona State University
Tempe, AZ 85287
`zou@asu.edu`

## Abstract

In this paper, we address the challenges of offline reinforcement learning (RL) under model mismatch, where the agent aims to optimize its performance through an offline dataset that may not accurately represent the deployment environment. We identify two primary challenges under the setting: inaccurate model estimation due to limited data and performance degradation caused by the model mismatch between the dataset-collecting environment and the target deployment one. To tackle these issues, we propose a unified principle of pessimism using distributionally robust Markov decision processes. We carefully construct a robust MDP with a single uncertainty set to tackle both data sparsity and model mismatch, and demonstrate that the optimal robust policy enjoys a near-optimal sub-optimality gap under the target environment across three widely used uncertainty models: total variation, $\chi^2$ divergence, and KL divergence. Our results improve upon or match the state-of-the-art performance under the total variation and KL divergence models, and provide the first result for the $\chi^2$ divergence model.

## 1 Introduction

Reinforcement learning (RL) [40] learns a policy to maximize cumulative rewards through online interactions with the environment. However, in real-world applications such as autonomous vehicles [16] and health care [61], the trial-and-error nature of interacting with the environment can be both costly and dangerous, rendering online learning impractical. To circumvent these challenges, the concept of offline RL has emerged [17, 18], seeking to learn an optimal policy from a pre-collected dataset, eliminating the need for real-time interaction with the environment.

Despite its potential, offline RL faces two significant challenges that impact its performance. The first challenge stems from the nature of the offline dataset itself. Offline RL demonstrates impressive performance only when the dataset is of high quality, as exemplified by [41, 9, 52, 18]. However, in most RL scenarios, the data collection process can be expensive, constrained, and subject to specific behavior policies. This often results in a dataset with insufficient samples and limited coverage. The limited coverage of state-action pairs in the dataset presents a substantial hurdle, making it challenging to fully learn the environment model and the model learned may be inaccurate, leading to a poorly performing policy. and hindering the discovery of the optimal policy, particularly when

38th Conference on Neural Information Processing Systems (NeurIPS 2024).

the behavior policy is sub-optimal. Existing solutions involve introducing pessimism during policy learning, by penalizing the reward function for state-action pairs that are not adequately covered by the dataset, known as the lower confidence bound (LCB) approach [15, 33, 19], demonstrating potential both numerically and theoretically in solving offline RL.

The second critical challenge in offline RL pertains to its vulnerability to model mismatch. As the offline dataset is collected in advance, it only encapsulates the environment's information at the time of data collection. In practical applications, environments often undergo variations due to e.g., unexpected perturbations, heterogeneity, or non-stationarity. This inherent variability introduces a model mismatch between the target environment and the one where the dataset is collected, leading to significant performance degradation when deploying the learned policy in the real environment. Robust RL or distributionally robust optimization (DRO) framework is then proposed to address this challenge [13, 28], which incorporates the pessimism principle to effectively tackle model uncertainty. By constructing an uncertainty set containing 'plausible' environments, robust RL optimizes the worst-case performance among them, providing a performance guarantee for the real environment.

To summarize, there are two sources of uncertainty in offline RL: **Uncertainty from limited data:** This stems from the inherent uncertainty introduced by limited offline datasets and lack of exploration; **Uncertainty from model-mismatch:** This arises from distribution shifts between data-collection and deployment environments, as well as between data-collection distribution and the distribution induced by the optimal policy. Each challenge can be addressed through its corresponding principle of pessimism, as in previous works: limited data coverage can be mitigated with reward estimation penalties (LCB), while model mismatch can be tackled with distribution estimation penalties (DRO). However, existing approaches often address these uncertainties separately, e.g., [36, 5, 24], leading to methodological redundancy or complexity (more discussions and comparisons can be found in Section 5). In this paper, we propose a unified framework that integrates both principles of pessimism into a **single** robust Markovian decision process (MDP) model for offline RL, offering a clear and streamlined conceptual formulation and providing improved or matched theoretical guarantees compared to existing methods. Our major contributions can be summarized as follows.

**A unified distributionally robust formulation for offline RL under model mismatch.** As discussed before, offline RL faces challenges including limited dataset coverage and model mismatch. Specifically, we first tackle the challenge of model mismatch using the approach of distributionally robust MDP, which optimizes the worst-case performance over an uncertainty set specified by e.g., total variation, $\chi^2$ or Kullback-Leibler divergence. We then show that the uncertainty from data sparsity can be transformed into a data-dependent penalization term that is added to the radius of the constructed uncertainty set. Our formulation hence unifies the two principles of pessimism to a **single** DRO problem that tackles both sources of uncertainty without additional structures and enjoys an easier implementation compared to existing works.

**Augmented design of radius to achieve tight theoretical guarantees.** Designing the uncertainty set for the model mismatch is typically straightforward, often relying on domain knowledge [39]. However, incorporating an additional penalty term to address data sparsity presents significant challenges. Balancing the conservativeness for less-visited state-action pairs estimation and the overall performance of learned policies requires careful consideration. In our work, we conduct a meticulous analysis to understand how the penalty radius impacts the performance of the learned policy. We then design penalty radii for three widely studied uncertainty set models: total variation, $\chi^2$ divergence, and KL divergence. Our analysis provides insights into the performance of learned policies under these penalty radii, showcasing the versatility of our framework across both metric-based and non-metric-based models. Moreover, our designs enable us to derive tight theoretical guarantees. Specifically, we improve upon existing results for robust offline RL under the total variation model [5], match the state-of-the-art performance under the KL divergence model [36], and present the first result under the $\chi^2$ divergence model.

## 2 Preliminaries

### 2.1 Markov Decision Process (MDP)

An MDP is specified by a tuple $(\mathcal{S}, \mathcal{A}, \mathsf{P}, r, \gamma)$, where $\mathcal{S}$ and $\mathcal{A}$ are the state and action spaces, respectively, $\mathsf{P} = \{\mathsf{P}_s^a \in \Delta(\mathcal{S}), s \in \mathcal{S}, a \in \mathcal{A}\}$[1] is the transition kernel, $r : \mathcal{S} \times \mathcal{A} \to [0, 1]$ denotes the reward function, and $\gamma \in [0, 1)$ is the discount factor. Let $S$ be the number of states and $A$ be the number of actions. For any transition kernel $\mathsf{P}$, $\mathsf{P}_s^a = (p_{s,s'}^a)_{s' \in \mathcal{S}}$, where $p_{s,s'}^a$ denotes the probability of transiting from state $s$ to state $s'$ after taking action $a$. The reward of the transition when taking action $a$ at state $s$ is denoted by $r(s, a)$.

For a stationary policy $\pi$ that maps from state $s \in \mathcal{S}$ to a distribution over action $a \in \mathcal{A}$, it specifies the probability of the agent taking actions at each state. The value function of a policy $\pi$ is defined as

$$V_\mathsf{P}^\pi(s) \triangleq \mathbb{E}_\mathsf{P} \left[ \sum_{t=0}^{\infty} \gamma^t r(s_t, a_t) | s_0 = s, \pi \right], \tag{1}$$

where $\mathbb{E}_\mathsf{P}$ denotes the expectation with respect to the distribution induced by the transition kernel $\mathsf{P}$. Let $\rho$ be the distribution of the initial state $s$, the value function under the distribution $\rho$ is denoted by $V_\mathsf{P}^\pi(\rho) = \mathbb{E}_{s \sim \rho}[V_\mathsf{P}^\pi(s)]$. Let $d_\mathsf{P}^\pi(s)$ and $d_\mathsf{P}^\pi(s, a)$ denote the state occupancy measure and state-action occupancy measure of policy $\pi$ when the initial state $s$ follows distribution $\rho$: $d_\mathsf{P}^\pi(s) = (1 - \gamma) \sum_{t=0}^{\infty} \gamma^t \mathbb{P}(s_t = s | s_0 \sim \rho, \pi, \mathsf{P})$, $d_\mathsf{P}^\pi(s, a) = d_\mathsf{P}^\pi(s)\pi(a|s)$.

### 2.2 Distributionally Robust MDP

A robust MDP is defined as $(\mathcal{S}, \mathcal{A}, \mathcal{P}, r, \gamma)$, where the transition kernel is not fixed but lies in some uncertainty set $\mathcal{P}$. In this work we consider the $(s, a)$-rectangular uncertainty set:

$$\mathcal{P} = \bigotimes_{s,a} \mathcal{P}_s^a, \ \mathcal{P}_s^a = \{q \in \Delta(\mathcal{S}) : D(q, \mathsf{P}_s^a) \le R\}, \tag{2}$$

where $\bigotimes_{s,a}$ means the uncertainty sets for every state-action pair are independently defined, $\mathsf{P}_s^a$ is some nominal transition kernel for $(s, a)$-pair, $D$ is some function that measures the difference between two distributions, e.g., total variation and KL divergence, and $R$ is the radius of the uncertainty set. The robust MDP aims to find the policy that optimizes the *worst-case* performance among all possible transition kernels from $\mathcal{P}$. Such a worst-case performance can be characterized by the robust value function $V_\mathcal{P}^\pi(s)$:

$$V_\mathcal{P}^\pi(s) \triangleq \min_{\zeta \in \mathcal{P}} V_\zeta^\pi(s), \tag{3}$$

which is shown to be the unique solution to the robust Bellman equation [13]:

$$V_\mathcal{P}^\pi(s) = \sum_a \pi(a|s) \left( r(s, a) + \gamma \sigma_{\mathcal{P}_s^a}(V_\mathcal{P}^\pi) \right), \tag{4}$$

with $\sigma_{\mathcal{P}_s^a}(V) \triangleq \min_{q \in \mathcal{P}_s^a} q^\top V$ being the support function of a vector $V$ on the uncertainty set $\mathcal{P}_s^a$.

Similar to the standard MDP setting, $V_\mathcal{P}^\pi(\rho)$ can also be defined w.r.t. the initial state distribution $\rho$. The goal here is then to find the optimal robust policy $\arg \max_\pi V_\mathcal{P}^\pi(\rho)$.

## 3 Problem Formulation

In offline RL, the agent does not receive new samples by interacting with the environment. Instead, it is given a previously collected dataset $\mathcal{D}$ which consists of $N$ tuples $\{(s_i, a_i, s_i', r_i) : i = 1, \cdots, N\}$. The dataset is generated according to some distribution $\mu$, i.e., $(s_i, a_i) \sim \mu$, and $s_i' \sim \mathsf{P}_{s_i}^{a_i}$ follows the nominal transition kernel $\mathsf{P}$, and $r_i = r(s_i, a_i)$ is a deterministic reward function. As discussed, due to the potential model mismatch between the environment that generates the offline dataset and the one in which the learned policy is going to be deployed, we design an uncertainty set that

---

[1] $\Delta(\mathcal{S})$ denotes the probability simplex defined on $\mathcal{S}$.

captures such a model mismatch as in (2), aiming to learn a policy that performs well under the true deployment environment through the robust RL framework:

$$\pi^* = \arg\max_\pi V_\mathcal{P}^\pi(\rho), \text{ where } \mathcal{P} = \bigotimes_{s,a} \mathcal{P}_s^a, \ \mathcal{P}_s^a = \{q \in \Delta(\mathcal{S}) : D(q, \mathsf{P}_s^a) \leq R\}. \tag{5}$$

where $R$ represents the similarities of the two environments. $D$ and $R$ are generally designed by domain experts, so we do not focus on their design and consider them as pre-settled in this work.

A key challenge here is that the nominal transition kernel $\mathsf{P}_0$ is unknown, but only an offline dataset generated from $\mathsf{P}_0$ is given. Due to the limited exploration and distributional shift, the dataset $\mathcal{D}$ may not cover all possible states or transitions that the agent might encounter in the environment.

To guarantee that a provable efficient algorithm can be designed based on the dataset $\mathcal{D}$, we adopt an assumption on the distributional mismatch between the dataset distribution and the occupancy measure induced by a comparator policy $\pi^*$. A comparator policy refers to a policy having satisfying worst-case performance and is set to be the optimal robust policy in many cases, but our approach can be applied to an arbitrary policy that may not be optimal.

**Assumption 1.** *(Robust single-policy clipped concentrability [36]). The data distribution $\mu$ satisfies*

$$C^{\pi^*} \triangleq \max_{(s,a,\mathsf{Q}) \in \mathcal{S} \times \mathcal{A} \times \mathcal{P}} \frac{\min\{d_\mathsf{Q}^{\pi^*}(s,a), \frac{1}{S}\}}{\mu(s,a)} < +\infty. \tag{6}$$

In Assumption 1, we only require that the dataset covers the state-action pairs that are visited by the comparator policy, known as partial coverage. When there is no model mismatch, Assumption 1 reduces to the single-policy clipped concentrability assumption in [19] for non-robust offline RL.

We hence formulate the offline RL problem with model mismatch as the following concrete problem: *Solve the robust RL problem* (5) *using only a fixed offline dataset $\mathcal{D}$ satisfying Assumption 1.*

## 4 Framework Design and Main Results

To simultaneously address the two sources of uncertainty (1) uncertainty arising from the model mismatch between the data-collected environment and the deployment environment; and (2) uncertainty in the nominal (data-collection) model estimation attributed to an insufficient amount of data and limited coverage of the offline dataset, we develop a unified principle of pessimism, and theoretically characterize the finite sample complexity of the proposed algorithms. Our algorithms are easier and more efficient to implement than existing approaches, and yield improved or matching results.

Denote $N(s,a) = \sum_{i=1}^N \mathbf{1}_{(s_i,a_i)=(s,a)}$ as the number of samples that transit from $(s,a)$ in $\mathcal{D}$, where $\mathbf{1}$ is the indicator function. The empirical transition kernel is then obtained as

$$\hat{\mathsf{P}}_{s,s'}^a = \begin{cases} \frac{\sum_{i=1}^N \mathbf{1}_{(s_i,a_i,s_i')=(s,a,s')}}{N(s,a)}, & \text{if } N(s,a) > 0 \\ \frac{1}{S}, & \text{if } N(s,a) = 0, \end{cases} \tag{7}$$

and we also define the empirical reward function as

$$\hat{r}(s,a) = \begin{cases} \frac{\sum_{(s_i,a_i)=(s,a)} r_i}{N(s,a)}, & \text{if } N(s,a) > 0 \\ 0, & \text{if } N(s,a) = 0. \end{cases} \tag{8}$$

The MDP $\hat{M} = (\mathcal{S}, \mathcal{A}, \hat{\mathsf{P}}, \hat{r})$ with the empirical transition kernel $\hat{\mathsf{P}}$ and empirical reward function $\hat{r}$ is referred to as the empirical MDP, representing our estimation of the nominal model from the dataset. A straightforward approach is to construct an uncertainty set $\hat{\mathcal{P}}$ by replacing the nominal kernel $\mathsf{P}$ in (5) by $\hat{\mathsf{P}}$:

$$\hat{\mathcal{P}}_s^a = \{q \in \Delta(\mathcal{S}) : D(q, \hat{\mathsf{P}}_s^a) \leq R\} \tag{9}$$

and solve the corresponding robust RL problem. Although it tackles the model mismatch uncertainty through the uncertainty set, it generally results in a sub-optimal performance due to the lack of consideration of the estimation error within the empirical model.

To tackle this estimation error, previous works introduce some additional structures other than the above DRO framework (9), including a penalty term in reward [36] or an additional uncertainty set and making it a bi-level DRO [5] (more discussions and comparisons can be found in Section 5). In this work, we construct a **single, unified** uncertainty set of environments, showing that such an estimation error can be incorporated into the distribution uncertainty set and developing a unified principle of pessimism to address both uncertainties. Specifically, for any $s \in \mathcal{S}$ and $a \in \mathcal{A}$, we set

$$\tilde{\mathcal{P}}_s^a = \{q \in \Delta(\mathcal{S}) : D(q, \hat{\mathsf{P}}_s^a) \leq R + \kappa_s^a\}, \tag{10}$$

where $\kappa_s^a$ is a function (will be specified later) inversely proportional to $N(s, a)$ that measures the degree of confidence when estimating the empirical transition kernel; and $R$ accounts for the model mismatch. Intuitively, for the less-observed state-action pairs, $\kappa_s^a$ becomes larger and we are less confident and more pessimistic when estimating the transition kernel $\hat{\mathsf{P}}_s^a$, and vice versa.

We claim and show later that the additional uncertainty or pessimism by enlarging the uncertainty set effectively tackles the uncertainty from the limited dataset, and the whole uncertainty set results in a conservative estimation of the worst-case performance. We then optimize the worst-case performance under this uncertainty set $\tilde{\mathcal{P}}$, which can be efficiently solved through the standard robust dynamic programming approach [28, 13]. Our algorithm is presented as Algorithm 1. For the convenience of analysis, we modify the vanilla robust value iteration algorithm [13] by setting the output policy to select actions that occur in the dataset when there is a tie. Such an output policy always exists (shown in Lemma 11 in the Appendix) and enables us to derive a tighter analysis of the sub-optimality gap. It is also worth noting that for all the uncertainty set models we consider, our Algorithm 1 can be efficiently applied with a polynomial computational complexity and a linear convergence rate [13].

---

**Algorithm 1** Robust Value Iteration for Offline RL with Model Mismatch

**Input:** $\mathcal{D}, V = 0$
Estimate the empirical reward $\hat{r}$ and empirical uncertainty set $\tilde{\mathcal{P}}$ according to (8) and (10)
**repeat**
  $V(\cdot) \leftarrow \max_{a \in \mathcal{A}} \{\hat{r}(\cdot, a) + \gamma \sigma_{\tilde{\mathcal{P}}_\cdot^a}(V)\}$
**until** convergence
**for** $s \in \mathcal{S}$ **do**
  $\tilde{\pi}(s) \in \left\{\arg\max_{a \in \mathcal{A}} \hat{r}(s, a) + \gamma \sigma_{\tilde{\mathcal{P}}_s^a}(V)\right\} \cap \{a : N(s, a) > 0\}$
**end for**
**Output:** $\tilde{\pi}$

---

In the following sections, we consider three widely used uncertainty set models: total variation, $\chi^2$ divergence, and KL divergence models. We show that, for all three models, the uncertainty from the dataset can be incorporated into the uncertainty set as an additional term in the radius, and with a carefully designed uncertainty set, the optimal robust policy $\tilde{\pi}$ obtained has a near-optimal performance under the target uncertainty set $\mathcal{P}$, i.e., our approach effectively and efficiently solves the offline RL problems under model mismatch.

## 4.1 Total Variation (TV)

We first consider the case of TV defined uncertainty set[2]. Specifically, the uncertainty set is constructed using $D(q, p) = \frac{1}{2}\|q - p\|_1$ in eq. (10). Note that when the radius is greater than 2, the defined uncertainty set reduces to the whole probability simplex $\Delta(\mathcal{S})$. Our first attempt is to set $\kappa_s^a$ large enough such that the true uncertainty set $\mathcal{P}$ falls into the constructed one $\tilde{\mathcal{P}}$, such that the robust value function of $\tilde{\mathcal{P}}$ lower bounds the true robust value function $V_{\mathcal{P}}^\pi$, as a conservative estimation.

We note that as long as the true nominal transition kernel $\mathsf{P}$ and the empirical transition kernel $\hat{\mathsf{P}}$ are close: $\|\mathsf{P}_s^a - \hat{\mathsf{P}}_s^a\| \leq \kappa_s^a$ holds with high probability, the triangle inequality implies $\mathcal{P} \subseteq \tilde{\mathcal{P}}$ with high probability. Therefore, the optimal robust policy $\tilde{\pi}$ provides a performance guarantee for the original problem (5). Following this approach, we can show the following results.

---

[2]Our approach described can also be applied to uncertainty sets defined by other metrics, e.g., Wasserstein distance and Hellinger distance.

**Theorem 1.** *Consider TV defined uncertainty set. For each state-action pair $(s, a)$, set $\kappa_s^a = \sqrt{\frac{S \log \frac{SA}{\delta}}{2N(s,a)}}$. Then with probability at least $1 - 2\delta$, the output policy $\tilde{\pi}$ of algorithm 1 satisfies that*

$$V_{\mathcal{P}}^{\pi^*}(\rho) - V_{\mathcal{P}}^{\tilde{\pi}}(\rho) \leq \tilde{\mathcal{O}}\left(\sqrt{\frac{S^2 C^{\pi^*}}{(1-\gamma)^4 N}}\right), \tag{11}$$

*where $\tilde{\mathcal{O}}$ notation absorbs universal constants and log terms.*

**Remark 1.** *To achieve an $\epsilon$ sub-optimality gap, a dataset of size $N = \tilde{\mathcal{O}}(S^2 C^{\pi^*}(1-\gamma)^{-4}\epsilon^{-2})$ is required. This result matches the previous one in [5].*

The construction above is based on distribution, to ensure the resulting uncertainty set $\tilde{\mathcal{P}}$ is larger than the target one $\mathcal{P}$. However, such a distribution-based construction can result in an overly large uncertainty set and an overly pessimistic policy, as observed in [47, 19]. To improve, one observation is that we can design a smaller uncertainty set such that the resulting robust value function $V_{\tilde{\mathcal{P}}}^{\pi}$ approximately lower bounds the true robust value function, without using the distribution-based framework. We hence further design a novel uncertainty set that turns out to be less conservative (the intuition of such a design will be discussed in the next section).

**Theorem 2.** *For any $(s, a)$, let $\kappa_s^a = \frac{\log \frac{SA}{\delta}}{N(s,a)}$. For the output policy $\tilde{\pi}$ of algorithm 1, with probability at least $1 - 4\delta$, it holds that*

$$V_{\tilde{\mathcal{P}}}^{\pi^*}(\rho) - V_{\tilde{\mathcal{P}}}^{\tilde{\pi}}(\rho) \leq \tilde{\mathcal{O}}\left(\sqrt{\frac{C^{\pi^*}S + \frac{1}{\mu_{\min}}}{N(1-\gamma)^3}}\right). \tag{12}$$

*Here, $\mu_{\min} \triangleq \min_{s,a}\{\mu(s, a) : \mu(s, a) > 0\}$ denotes the smallest non-zero entry of the distribution $\mu$ that generates the dataset.*

**Remark 2.** *Combining the two results, we showed that our approach can obtain an $\epsilon$-optimal robust policy when the dataset is of size $\tilde{\mathcal{O}}\left(\frac{C^{\pi^*}S}{(1-\gamma)^3\epsilon^2}\min\{\frac{S}{1-\gamma}, \frac{1}{\mu_{\min}}\}\right)$ for the TV defined model. This result is better than the previous result in [5], illustrating our approach is more efficient. Moreover, our framework is much simpler than the one in [5] and can be effectively solved in a polynomial time (see detailed discussion in Section 5).*

### 4.2 $\chi^2$ Divergence

We then study the uncertainty models with the $\chi^2$ divergence: $D(p, q) = \sum_s q(s)\left(1 - \frac{p(s)}{q(s)}\right)^2$. Note that $\chi^2$ divergence is not a metric, implying the failure of the triangle inequality and the distribution-based design as in the previous section. While it is possible to design $\kappa$ such that $D(\hat{\mathsf{P}}_s^a, \mathsf{P}_s^a) \leq \kappa_s^a$ [31], it does not imply $\mathcal{P}_s^a \subset \tilde{\mathcal{P}}_s^a$ and no lower bound guarantee can be obtained.

To address this issue, we similarly adapt the value function-based construction, by taking a closer look at the error decomposition. The main idea of our distribution-based construction for the TV defined model is to construct an uncertainty set that is large enough to include the true transition kernel with high probability. Then the sub-optimality gap can be decomposed as

$$V_{\mathcal{P}}^{\pi^*} - V_{\mathcal{P}}^{\tilde{\pi}} = \underbrace{V_{\mathcal{P}}^{\pi^*} - V_{\tilde{\mathcal{P}}}^{\tilde{\pi}}}_{\Delta_1} + \underbrace{V_{\tilde{\mathcal{P}}}^{\tilde{\pi}} - V_{\mathcal{P}}^{\tilde{\pi}}}_{\Delta_2 \leq 0}, \tag{13}$$

and $\Delta_2 \leq 0$ from $\mathcal{P}_s^a \subset \tilde{\mathcal{P}}_s^a$, which however fails under non-metric models. On the other hand, if we further decompose $\Delta_2$ as

$$V_{\tilde{\mathcal{P}}}^{\tilde{\pi}} - V_{\mathcal{P}}^{\tilde{\pi}} = \underbrace{V_{\tilde{\mathcal{P}}}^{\tilde{\pi}} - V_{\hat{\mathcal{P}}}^{\tilde{\pi}}}_{\Delta_{21}} + \underbrace{V_{\hat{\mathcal{P}}}^{\tilde{\pi}} - V_{\mathcal{P}}^{\tilde{\pi}}}_{\Delta_{22}}, \tag{14}$$

where $\hat{\mathcal{P}}$ is the empirical uncertainty set (9). We note that $\Delta_{22}$ is the concentration error due to the limited dataset, which is independent of the term $\kappa_s^a$ (ignoring the dependence of $\tilde{\pi}$ on $\kappa$ for discussion

connivance); $\Delta_{21}$ will be a negative term since $\hat{\mathcal{P}} \subset \tilde{\mathcal{P}}$, thus we should set the term $\kappa$ such that the negative bound on $\Delta_{21}$ cancels out with the concentration bound on $\Delta_{22}$, leading to a non-zero yet tight overall bound on $\Delta_2$. Instead of ensuring the uncertainty set inclusion, we directly ensure the bound on the value function difference $\Delta_2$ is small. Clearly, the clue function-based design can be applied to both non-metric and metric models, and is less conservative than the distribution-based for the metric models, since closeness in distribution is stronger than the closeness in value functions [47]. This observation results in our second design for the total variation model in Theorem 2, showing an improvement in sample complexity.

Based on this intuition, we present our radius design and results.

**Theorem 3.** *Consider $\chi^2$ divergence-defined uncertainty set. For any $(s,a)$, let $\kappa_s^a = \tilde{\mathcal{O}}\left(\sqrt{\frac{1+R}{N(s,a)}}\right)$. When $N \geq \tilde{\mathcal{O}}\left(\frac{S}{C^{\pi^*}\mu_{\min}^2}\right)$, with probability at least $1 - 4\delta$, the output policy $\tilde{\pi}$ of algorithm 1 satisfies*

$$V_{\mathcal{P}}^{\pi^*}(\rho) - V_{\mathcal{P}}^{\tilde{\pi}}(\rho) \leq \tilde{\mathcal{O}}\left(\sqrt{\frac{C^{\pi^*}S}{N(1-\gamma)^4}}\right). \tag{15}$$

**Remark 3.** *To achieve an $\epsilon$-optimal robust policy under the $\chi^2$ model, our approach requires the total number of samples*

$$N = \tilde{\mathcal{O}}\bigg(\underbrace{\frac{C^{\pi^*}S}{(1-\gamma)^4\epsilon^2}}_{\epsilon\text{-dependent}} + \underbrace{\frac{S}{C^{\pi^*}\mu_{\min}^2}}_{\text{burn-in cost}}\bigg).$$

*Our sample complexity has two parts: the $\epsilon$-dependent part which dominates as the accuracy $\epsilon$ decreases, and a fixed amount of burn-in cost, whose existence is because we cannot expect to learn a near-optimal policy when the dataset is too limited. When the desired accuracy $\epsilon$ decreases, the first term dominates the overall complexity, resulting in the asymptotic result presented in the theorem.*

Our approach and result stand for the first concrete study for offline robust RL with $\chi^2$ divergence-defined uncertainty sets. It is also worth noting that our sample complexity asymptotically matches the sample complexity of the model-based robust RL with a generative model [38]. These observations hence demonstrate the optimality of our results and the effectiveness of our approach.

### 4.3 KL Divergence

In this section, we consider the KL divergence defined uncertainty set, i.e., $D(p,q) = \sum_s p(s) \log \frac{p(s)}{q(s)}$. Similarly, KL divergence is not a metric, we hence adapt our design discussed above for $\chi^2$ models here. The following theorem presents our design of the uncertainty set and sample complexity results.

**Theorem 4.** *Consider KL divergence defined uncertainty set. For any $(s,a)$, let $\kappa_s^a = C_1\sqrt{\frac{\log\frac{2(1+R)N^3S}{(1-\gamma)\delta}}{N(s,a)\hat{\mathsf{P}}_{\min}}}$, if $N \geq \frac{8\log\frac{1}{\delta}}{\mu_{\min}\mathsf{P}_{\min}}$, then with probability at least $1 - 4\delta$, the output policy $\tilde{\pi}$ of algorithm 1 satisfies*

$$V_{\mathcal{P}}^{\pi^*}(\rho) - V_{\mathcal{P}}^{\tilde{\pi}}(\rho) \leq \tilde{\mathcal{O}}\left(\sqrt{\frac{C^{\pi^*}S}{(1-\gamma)^4N\mathsf{P}_{\min}}}\right). \tag{16}$$

*Here, $\mathsf{P}_{\min}$ and $\hat{\mathsf{P}}_{\min}$ represents the minimal non-zero entry of the nominal transition kernel $\mathsf{P}$ and empirical nominal kernel $\hat{\mathsf{P}}$.*

**Remark 4.** *Similarly, for the KL divergence model, our approach requires a total number of $\tilde{\mathcal{O}}\left(\frac{C^{\pi^*}S}{(1-\gamma)^4\mathsf{P}_{\min}\epsilon^2} + \frac{1}{\mathsf{P}_{\min}\mu_{\min}}\right)$ samples to find an $\epsilon$-optimal policy. The sample complexity also contains two parts, an asymptotically dominated term and a fixed burn-in cost. Our result matches the one of [36] and is better than the one of [5].*

To summarize, our unified framework can be adapted to different uncertainty models and solve offline robust RL problems, offering improved or matched sample complexity results and a more efficient

implementation. More discussion can be found in Section 5. We also provide some numerical experiments to verify the effectiveness and efficiency of our algorithm, which can be found in Appendix A. Under all three uncertainty set models, our algorithm enjoys a smaller or similar sample complexity compared to LCB approaches, and always outperform the non-robust dynamic programming approach.

# 5 Related Works

## 5.1 Comparison with prior art

In this section, we compare our works with the most related existing works [36, 5], where offline RL with model mismatch is studied. Compared to them, our methods enjoy three major advantages: (1). A more unified and straightforward framework of single pessimism principle; (2). Improved or matched sample complexity; And (3). Enhanced computational complexity.

**Unified framework for double pessimism principles.** In [36], the two principles of pessimism are separately employed, where a reward penalty term $b$ is used to penalize less visited state-action pairs in addition to the uncertainty set that accounts for the model mismatch, and the update rule is

$$V(s) \leftarrow \max_a \left\{ r(s,a) + \gamma \sigma_{\hat{\mathcal{P}}_s^a}(V) - b(s,a) \right\}. \tag{17}$$

Our approach, on the other hand, enjoys a straightforward and simple formulation. Specifically, we incorporate the two principles of pessimism into a single uncertainty set, developing our unified principle. Moreover, it is noted that design of the penalty term $b$ is complicated, whereas our addition term $\kappa$ has a simple and clear form. It is worth noting that although the LCB approach and ours share a similar updating rule, there is a fundamental difference in the algorithm design motivation and analysis. In the LCB approach, the penalty term $b$ is designed such that the resulting estimation $V$ is less than the true robust value function $V_{\mathcal{P}}^\pi$, to ensure the conservativeness of LCB approaches; Whereas our resulting estimation $V_{\hat{\mathcal{P}}}^\pi$ is not necessarily less than $V_{\mathcal{P}}^\pi$, since we directly tackle the distribution uncertainty but not through value function estimations. This hence requires novel technique innovations in our analysis.

In another closely related work [5], a double pessimism principle is adopted to address the two sources of uncertainty, under TV and KL divergence models. Specifically, a TV distance-based uncertainty set $\hat{\mathcal{P}}$ is first constructed to tackle the model estimation uncertainty from the dataset; Then centered at each transition kernel $\hat{\mathsf{P}} \in \hat{\mathcal{P}}$, a second layer of uncertainty set $\Phi(\hat{\mathsf{P}})$ to reflect the distributional robustness to model mismatch. They then take the optimal policy of

$$V_{pess^2}^\pi = \inf_{\hat{\mathsf{P}} \in \hat{\mathcal{P}}} \inf_{\tilde{\mathsf{P}} \in \Phi(\hat{\mathsf{P}})} V_{\tilde{\mathsf{P}}}^\pi \tag{18}$$

as the output policy. Although their approach indicates that the data estimation uncertainty can also be captured by a distributional uncertainty set, they still employ the two principles separately, although both in the form of DRO formulations. Instead of designing two uncertainty sets as in [5], we unify the two types of pessimism and use a **single** uncertainty set with a composed design of the radius to address both the model estimation uncertainty and the model mismatch.

**Improved or Matched Sample Complexity.** In terms of sample complexity, the comparison are included in Table 1. Compared to [36], our theoretical result matches theirs under the KL divergence model, and we moreover develop results for the other two uncertainty set models that are not considered therein. The numerical experiment results can be found in Appendix A, which further verify our discussion. Compared with [5], our results achieve better sample complexity in both KL and TV models. In the TV model, our sample complexity outperforms [5] in terms of dependence on $S$ and $(1-\gamma)$; For the KL model, our complexity is linearly dependent on $S$, while [5] has a quadratic dependence. Furthermore, as noted in [5], their result's exponential term can be replaced by utilizing both $\mathsf{P}_{\min}$ and $\mu_{\min}$, while our (asymptotic) complexity result depends solely on $\mathsf{P}_{\min}$.

**Enhanced Computational Complexity.** Our algorithms are also better in terms of computational complexity or practical implementations than both baselines.

The two-layer optimization problem in [5] is an extension of the model studied in [42] to the robust setting, both involving non-rectangular uncertainty sets that are NP-hard to solve [50]. This creates

| | TV DISTANCE | $\chi^2$ DIVERGENCE | KL DIVERGENCE |
|---|---|---|---|
| [36] | $\times$ | $\times$ | $\tilde{\mathcal{O}}\big(\frac{C^{\pi^*}S}{(1-\gamma)^4\epsilon^2\mathsf{P}_{\min}}\big)$ |
| [5] | $\tilde{\mathcal{O}}\big(\frac{C^{\pi^*}_{rob}S^2}{(1-\gamma)^4\epsilon^2}\big)$ | $\times$ | $\tilde{\mathcal{O}}\big(\frac{C^{\pi^*}_{rob}S^2\exp((1-\gamma)^{-1})}{(1-\gamma)^4\epsilon^2}\big)$ |
| OUR WORK | $\tilde{\mathcal{O}}\big(\frac{C^{\pi^*}S}{(1-\gamma)^3\epsilon^2}\min\{\frac{S}{1-\gamma},\frac{1}{\mu_{\min}}\}\big)$ | $\tilde{\mathcal{O}}\big(\frac{C^{\pi^*}S}{(1-\gamma)^4\epsilon^2}\big)$ | $\tilde{\mathcal{O}}\big(\frac{C^{\pi^*}S}{(1-\gamma)^4\epsilon^2\mathsf{P}_{\min}}\big)$ |

Table 1: Comparison with related works on offline RL under model mismatch. In [5], $C^{\pi^*}_{rob}$ is the robust partial coverage coefficient [5], which is similar to $C^{\pi^*}$, and the $\exp((1-\gamma)^{-1})$ term can be eliminated with an additional cost on $\mathsf{P}^{-1}_{\min}$ and $\mu^{-1}_{\min}$.

uncertainty regarding the solvability of their models. Specifically, due to the unsolvability of the non-robust model in [42], an adversarial training-based algorithm is designed in [34] with only an experimental convergence guarantee, highlighting the implementation challenges of [5]. In contrast, our algorithm can be implemented with polynomial complexity. Specifically, the total computational complexity of our algorithm in the TV, CS and KL models are $\mathcal{O}(S^2A\log S)$,$\mathcal{O}(S^2A\log S)$, and $\tilde{\mathcal{O}}(S^2A)$.

Compared to [36], our algorithms also offer better computational complexity. Specifically, the penalty term in [36] requires a complicated computation involving a minimum operator, resulting in an additional max-min structure in their algorithm update. The comparison/max-min operator in the LCB algorithm is executed $SA$ times per step, significantly increasing computational complexity. In contrast, our algorithms have a simple structure and do not require additional operators, making them more computationally efficient. We also use numerical experiment to further illustrate our enhanced computational efficiency, in Appendix A.

## 5.2 Other related works

We then discuss some of the other related works.

**Offline RL without model mismatch.** Offline RL focuses on learning an optimal policy from a pre-collected dataset, and the target deployment environment is identical to the one where the dataset is collected. Many previous works make the global coverage assumption, i.e., the behavior policy can cover all state-action pairs, e.g., [35, 7, 27, 58, 59, 14, 45, 20, 21, 64, 41, 9, 52, 18, 2, 10]. This assumption is too restrictive and is often violated in practice since it requires the history data to cover all the state-action pairs [12, 1, 11]. Recently, a relaxed partial coverage setting was proposed, which assumes that the density ratio between the occupancy measure induced by a single target policy and the behavior policy is finite for all state-action pairs, and the goal is to learn a policy that is no worse than the target policy. The partial coverage assumption only requires that the history data visit the state-action pairs that the target policy will visit. Under the partial coverage assumption, optimal policy can be learned for offline RL incorporated with the pessimism principle facing the uncertainty, e.g., [15, 42, 51, 53, 33, 62, 60, 37, 19, 63, 47]. However in our setting, we also consider the potential model mismatch between the two environments, which possibly due to e.g., non-stationarity, heterogeneity and sim-to-real gap, and formulate the problem as offline RL under model mismatch.

**Robust RL.** Robust RL [13, 28, 54] aims to tackle the model mismatch in RL, by optimizing the worst-case performance over the uncertainty set. Existing works mainly focus on the online setting [48, 49, 46, 4, 8, 25, 23] or with a generative model [57, 55, 29, 38]. Studies for robust RL with an offline dataset, besides the two mentioned above [36, 5], are developed in recent works including [24, 44, 65, 30, 56, 26, 57]. These works either focus on the linear MDPs, or adapt strong assumptions including global coverage or absorbing states. More importantly, all these works employ the two pessimism principles separately through the LCB penalty and DRO uncertainty set. Compared to them, we focus on general MDPs and develop our unified framework.

## 6 Conclusion and Discussions

In this paper, we investigated the offline RL problem under model mismatch under the most general partial coverage setting, where two sources of uncertainty are presented: inaccurate estimation of transition dynamics due to limited dataset coverage, and model mismatch between training and testing

environments. We developed a unified DRO-based framework containing a single uncertainty set with a composed radius of two parts to tackle the two sources of uncertainty discussed above. Our approach can be implemented in a much easier and more straightforward way than existing approaches and can be applied to both metric-based and non-metric-based uncertainty models. Specifically, we investigated three types of uncertainty sets defined by total variation, $\chi^2$ divergence, and KL divergence. Our methodology can be easily extended to handle other uncertainty set models. Among them, we obtain near-optimal sample complexity results that improve or match the existing results under the total variation and KL divergence models and provide the first algorithm and finite sample complexity analysis for the uncertainty set defined by the $\chi^2$ divergence.

**Limitations.** It is in our future interest to extend our unified framework to address large-scale problems. This includes robust MDPs with latent structures, such as linear MDPs, as opposed to previous work that uses the LCB + DRO framework, for example [24, 44], and more general MDPs with function approximation techniques.

## 7 Acknowledgements

Yue Wang is supported by DARPA under Agreement No. HR0011-24-9-0427. The work of Zhongchang Sun and Shaofeng Zou is supported by the National Science Foundation under Grants CCF-2438429 and ECCS-2438392 (CAREER).

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

# A  Experiments

In this section, we provide simulation results to demonstrate the performance of our algorithm. We consider two problems: the Frozen-Lake problem [6] and the gambler problem [40, 65, 36].

For the Frozen-Lake problem, an agent aims to cross a $4 \times 4$ frozen lake from Start to Goal without falling into any Holes. The reward is set to be 1 when the agent reaches Goal and 0 otherwise. Due to the slippery nature of the frozen lake, the agent may not always move along the intended direction.

We formulate the gambler problem as an infinite-horizon MDP. A gambler engages in a betting game based on a sequence of coin flips. The gambler wins the stake when the coin lands on heads and loses it when the coin lands on tails. The probability of heads for the coin flip is $p = 0.6$. The game begins with an initial balance and ends when the gambler's balance either reaches 20 or 0. The reward is set to be 1 when the state reaches 20 and 0 otherwise.

## A.1  Comparison under the total variation uncertainty set model

We first evaluate our algorithm under the total variation uncertainty set. We adapt the construction in [36] to design an DRVI-LCB approach for the TV defined uncertainty set as a baseline, and implement the two variants of our algorithm: distribution-based design and value function-based design. The robustness level $R$ is set to be 0.1. We generate the dataset according to $\mu(s, a) = \frac{\mathbf{1}_{a=\pi^*(s)}}{2} + \frac{\mathbf{1}_{a=\eta}}{2}$ where $\eta$ is a random action, and $\pi^*(s)$ denotes the optimal robust policy. Clearly the dataset satisfies the partial coverage assumption 1.

We run the algorithms independently for 10 times and plot the mean of the sub-optimality gap and mean plus and minus the standard deviations of the 10 runs as the envelop. It can be seen from Fig.1 and Fig. 2 that the value function-based construction has a smaller sub-optimality gap and converges faster than the distribution-based construction and the DRVI-LCB, which demonstrates that our algorithm is less conservative and more effective.

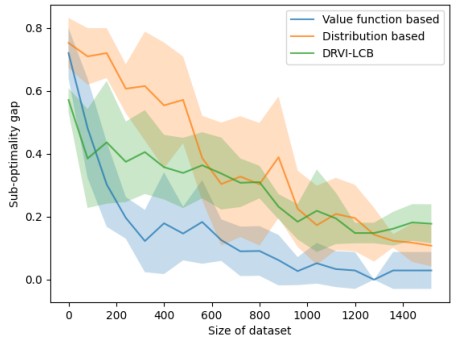
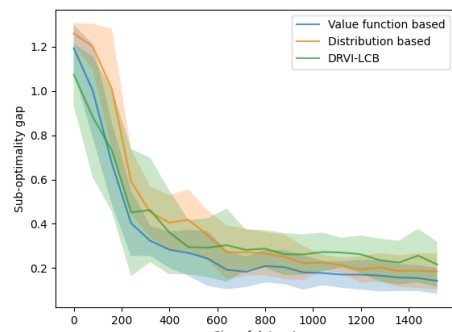

Figure 1: Frozen-Lake: TV Distance Defined Uncertainty Set

Figure 2: Gambler: TV Distance Defined Uncertainty Set

## A.2  Comparison under the $\chi^2$ divergence uncertainty set model

We then compare our algorithm with the DRVI-LCB under the $\chi^2$ divergence model. Similarly, we adapt their penalty term design for the uncertainty set, and run the experiment under the two environments. We similarly generate dataset following $\mu$ constructed above and run the algorithm for 10 times. The robustness level $R$ is set to be 0.1. It can be seen from Fig.3 and Fig.4 that the non-robust DP converges much slower to the optimal policy, whereas our approach has a similar convergence rate to the DRVI-LCB to the optimal policy, which validates our theoretical results.

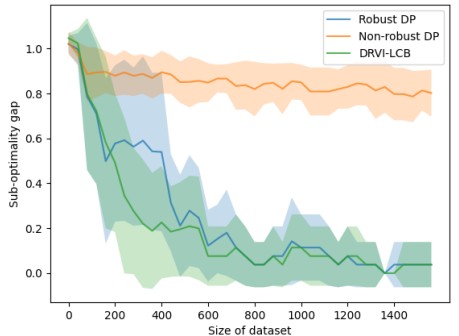 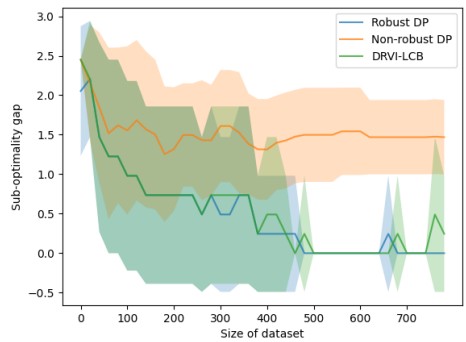

Figure 3: Frozen-Lake: $\chi^2$ Divergence Defined Uncertainty Set

Figure 4: Gambler: $\chi^2$ Divergence Defined Uncertainty Set

## A.3 Comparison under the KL divergence uncertainty set model

We then compare the performance of our algorithm and the one in [36] under the KL divergence model. We similarly generate dataset following $\mu$ constructed above and run the algorithm for 10 times. The robustness level $R$ is set to be $0.1$. It can be seen from Fig.5 and Fig.6 that the non-robust DP converges much slower to the optimal policy, whereas our approach has a similar convergence rate to the DRVI-LCB to the optimal policy, which validates our theoretical results.

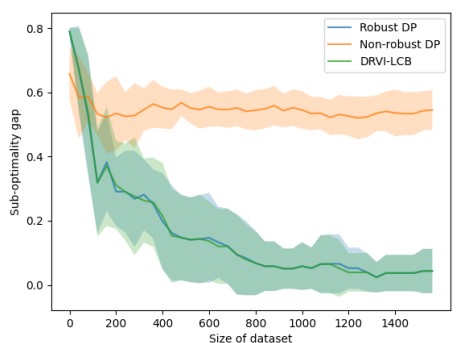 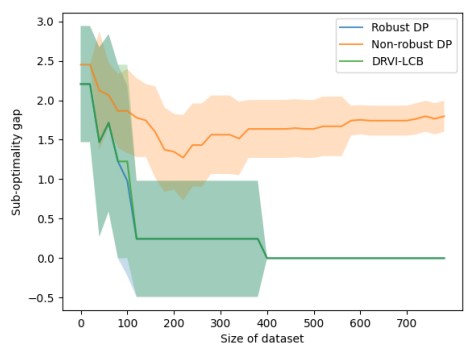

Figure 5: Frozen-Lake: KL Divergence Defined Uncertainty Set

Figure 6: Gambler: KL Divergence Defined Uncertainty Set

We further run the three algorithms on the two uncertainty sets for the gambler problem. The robustness level $R$ is set to be $0.2$. From Fig. 4 and Fig. 6, it can be seen that the non-robust DP converges much slower, whereas our approach solves the problem efficiently. Compared to LCB approaches, our method enjoys a similar performance and convergence rate, further demonstrating the effectiveness and efficiency of our approach.

## A.4 Execution time

To illustrate our computational efficiency, we implemented the LCB algorithm from [36] and our DRO algorithm under the KL model, monitoring the execution time of both methods while learning the same policy from the same dataset. We plotted the execution time for each dataset versus the size of the dataset in three environments: Gambler's game, Frozen Lake, and N-chain. As shown in Appendix A.4, our algorithm consistently requires less execution time across all three environments, demonstrating lower computational complexity.

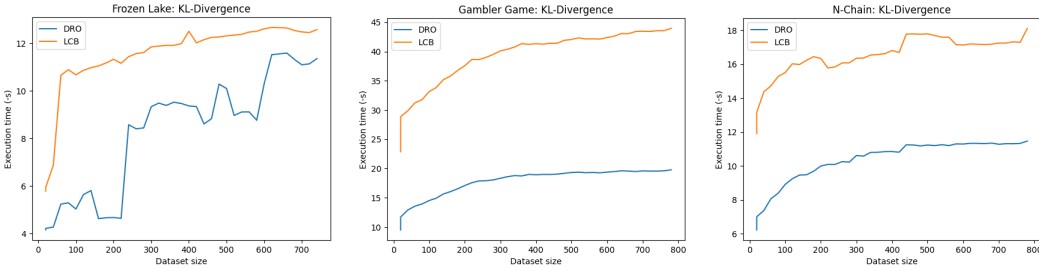

Figure 7: Execution time: DRO vs LCB [36]

# B  Proofs of Section 4.1

**Lemma 1.** *For each state-action pair* $(s, a)$, *set*

$$\kappa_s^a = \sqrt{\frac{S \log \frac{SA}{\delta}}{2N(s,a)}}.$$

*Then with probability at least* $1 - \delta$, $\mathcal{P}_s^a \subseteq \tilde{\mathcal{P}}_s^a$.

We then present the sub-optimality gap in the following theorem.

*Proof.* For the $(s, a)$ pairs with $N(s, a) = 0$, the statement is trivial due to the fact that $\tilde{\mathcal{P}}_s^a = \Delta(\mathcal{S})$. It is hence sufficient to consider the pairs with $N(s, a) > 0$.

By directly applying the Hoeffding's inequality [22], we have that for each pair $(s, a, x)$,

$$\mathbb{P}(|\hat{\mathsf{P}}_{s,x}^a - \mathsf{P}_{s,x}^a| \geq k) \leq \mathbf{exp}\left(-2N(s,a)k^2\right). \tag{19}$$

It hence can be further shown that

$$\|\hat{\mathsf{P}}_s^a - \mathsf{P}_s^a\| \leq \sqrt{\frac{S \log \frac{SA}{\delta}}{2N(s,a)}} \tag{20}$$

simultaneously for any $(s, a)$-pair with probability at least $1 - \delta$.

Now consider any $q \in \mathcal{P}_s^a$, it holds that $\|q - \mathsf{P}_s^a\| \leq R$. Hence

$$\|q - \hat{\mathsf{P}}_s^a\| \leq \|q - \mathsf{P}_s^a\| + \|\hat{\mathsf{P}}_s^a - \mathsf{P}_s^a\| \leq R + \sqrt{\frac{S \log \frac{SA}{\delta}}{2N(s,a)}}, \tag{21}$$

which implies that $\mathcal{P}_s^a \subseteq \tilde{\mathcal{P}}_s^a$ for any $(s, a)$ with probability at least $1 - \delta$. $\square$

**Theorem 5.** *For the output policy* $\tilde{\pi}$ *of algorithm 1, with probability at least* $1 - 2\delta$, *it holds that*

$$V_{\mathcal{P}}^{\pi^*}(\rho) - V_{\mathcal{P}}^{\tilde{\pi}}(\rho) \leq \frac{2}{(1-\gamma)^2} \sqrt{\frac{8 \log \frac{SA}{\delta} \log \frac{4SA}{\delta}}{2N\mu_{\min}}}, \tag{22}$$

*Proof.* First note that

$$V_{\mathcal{P}}^{\pi^*} - V_{\mathcal{P}}^{\tilde{\pi}} = V_{\mathcal{P}}^{\pi^*} - V_{\tilde{\mathcal{P}}}^{\tilde{\pi}} + V_{\tilde{\mathcal{P}}}^{\tilde{\pi}} - V_{\mathcal{P}}^{\tilde{\pi}}. \tag{23}$$

Due to the fact that $\mathcal{P}_s^a \subseteq \tilde{\mathcal{P}}_s^a$ with probability $1 - \delta$, it holds that

$$V_{\tilde{\mathcal{P}}}^{\tilde{\pi}} \leq V_{\mathcal{P}}^{\tilde{\pi}}, \tag{24}$$

and hence

$$V_{\mathcal{P}}^{\pi^*} - V_{\mathcal{P}}^{\tilde{\pi}} \leq V_{\mathcal{P}}^{\pi^*} - V_{\tilde{\mathcal{P}}}^{\tilde{\pi}}. \tag{25}$$

It can be further bounded as

$$V_{\mathcal{P}}^{\pi^*}(s) - V_{\tilde{\mathcal{P}}}^{\tilde{\pi}}(s)$$

$$= r(s, \pi^*(s)) + \gamma \sigma_s^*(V_{\mathcal{P}}^{\pi^*}) - \max_a \{ r(s,a) + \gamma \tilde{\sigma}_s^a(V_{\tilde{\mathcal{P}}}^{\tilde{\pi}}) \}$$

$$\overset{(a)}{\leq} r(s, \pi^*(s)) + \gamma \sigma_s^*(V_{\mathcal{P}}^{\pi^*}) - r(s, \pi^*(s)) - \gamma \tilde{\sigma}_s^*(V_{\tilde{\mathcal{P}}}^{\tilde{\pi}})$$

$$= \gamma \sigma_s^*(V_{\mathcal{P}}^{\pi^*}) - \gamma \tilde{\sigma}_s^*(V_{\tilde{\mathcal{P}}}^{\tilde{\pi}})$$

$$= \gamma \sigma_s^*(V_{\mathcal{P}}^{\pi^*}) - \gamma \sigma_s^*(V_{\tilde{\mathcal{P}}}^{\tilde{\pi}}) + \gamma \sigma_s^*(V_{\tilde{\mathcal{P}}}^{\tilde{\pi}}) - \gamma \tilde{\sigma}_s^*(V_{\tilde{\mathcal{P}}}^{\tilde{\pi}})$$

$$\leq \gamma (\mathsf{P}_{\tilde{V}})_s^*(V_{\mathcal{P}}^{\pi^*} - V_{\tilde{\mathcal{P}}}^{\tilde{\pi}}) + \gamma \sigma_s^*(V_{\tilde{\mathcal{P}}}^{\tilde{\pi}}) - \gamma \tilde{\sigma}_s^*(V_{\tilde{\mathcal{P}}}^{\tilde{\pi}}), \tag{26}$$

where $(a)$ is from $\tilde{\pi} = \arg\max_\pi V_{\tilde{\mathcal{P}}}^\pi$, and the last inequality is from the fact $\mathsf{P}_{\tilde{V}} \in \mathcal{P}_s^*$ and hence $\sigma_s^*(V_{\mathcal{P}}^{\pi^*}) \leq (\mathsf{P}_{\tilde{V}})_s^* V_{\mathcal{P}}^{\pi^*}$.

Applying the inequality above recursively implies

$$V_{\mathcal{P}}^{\pi^*}(s) - V_{\tilde{\mathcal{P}}}^{\tilde{\pi}}(s) \leq \frac{1}{1-\gamma} \sum_x d_{\mathsf{P}_{\tilde{V}}}^*(x) |\sigma_x^*(V_{\tilde{\mathcal{P}}}^{\tilde{\pi}}) - \tilde{\sigma}_x^*(V_{\tilde{\mathcal{P}}}^{\tilde{\pi}})|. \tag{27}$$

Note that

$$\sum_x d_{\mathsf{P}_{\tilde{V}}}^*(x) |\sigma_x^*(V_{\tilde{\mathcal{P}}}^{\tilde{\pi}}) - \tilde{\sigma}_x^*(V_{\tilde{\mathcal{P}}}^{\tilde{\pi}})|$$

$$= \sum_x d_{\mathsf{P}_{\tilde{V}}}^*(x) |\sigma_x^*(V_{\tilde{\mathcal{P}}}^{\tilde{\pi}}) - \hat{\sigma}_x^*(V_{\tilde{\mathcal{P}}}^{\tilde{\pi}}) + \hat{\sigma}_x^*(V_{\tilde{\mathcal{P}}}^{\tilde{\pi}}) - \tilde{\sigma}_x^*(V_{\tilde{\mathcal{P}}}^{\tilde{\pi}})|$$

$$\leq \sum_x d_{\mathsf{P}_{\tilde{V}}}^*(x) |\sigma_x^*(V_{\tilde{\mathcal{P}}}^{\tilde{\pi}}) - \hat{\sigma}_x^*(V_{\tilde{\mathcal{P}}}^{\tilde{\pi}})| + \sum_x d_{\mathsf{P}_{\tilde{V}}}^*(x) |\hat{\sigma}_x^*(V_{\tilde{\mathcal{P}}}^{\tilde{\pi}}) - \tilde{\sigma}_x^*(V_{\tilde{\mathcal{P}}}^{\tilde{\pi}})|$$

$$\leq \sum_x d_{\mathsf{P}_{\tilde{V}}}^*(x) |\sigma_x^*(V_{\tilde{\mathcal{P}}}^{\tilde{\pi}}) - \hat{\sigma}_x^*(V_{\tilde{\mathcal{P}}}^{\tilde{\pi}})| + \frac{1}{1-\gamma} \sqrt{\frac{8 \log \frac{4SA}{\delta} \log \frac{SA}{\delta}}{2N\mu(s, \pi^*(s))}}, \tag{28}$$

where the last inequality is from

$$\|\hat{\sigma}^*(V_{\tilde{\mathcal{P}}}^{\tilde{\pi}}) - \tilde{\sigma}^*(V_{\tilde{\mathcal{P}}}^{\tilde{\pi}})\|$$

$$\overset{(a)}{=} \max_s \left\{ \left| \max_{0 \leq \lambda \leq \frac{1}{1-\gamma}} \left\{ \hat{\mathsf{P}}_s^*(V_{\tilde{\mathcal{P}}}^{\tilde{\pi}} - \mu) - R\mathbf{Span}(V_{\tilde{\mathcal{P}}}^{\tilde{\pi}} - \lambda) \right\} \right. \right.$$

$$\left. \left. - \max_{0 \leq \lambda \leq \frac{1}{1-\gamma}} \left\{ \hat{\mathsf{P}}_s^*(V_{\tilde{\mathcal{P}}}^{\tilde{\pi}} - \lambda) - (\kappa_s^* + R)\mathbf{Span}(V_{\tilde{\mathcal{P}}}^{\tilde{\pi}} - \lambda) \right\} \right| \right\}$$

$$\overset{(b)}{\leq} \max_s \left\{ \left| \max_{0 \leq \lambda \leq \frac{1}{1-\gamma}} \left\{ \hat{\mathsf{P}}_s^*(V_{\tilde{\mathcal{P}}}^{\tilde{\pi}} - \lambda) + R\mathbf{Span}(V_{\tilde{\mathcal{P}}}^{\tilde{\pi}} - \lambda) - \hat{\mathsf{P}}_s^*(V_{\tilde{\mathcal{P}}}^{\tilde{\pi}} - \lambda) - (\kappa_s^* + R)\mathbf{Span}(V_{\tilde{\mathcal{P}}}^{\tilde{\pi}} - \lambda) \right\} \right| \right\}$$

$$\leq \max_s \left\{ \max_{0 \leq \lambda \leq \frac{1}{1-\gamma}} |\kappa_s^* \mathbf{Span}(V_{\tilde{\mathcal{P}}}^{\tilde{\pi}} - \lambda)| \right\}, \tag{29}$$

where $(a)$ is from the dual form of $\sigma_{\mathcal{P}}(V)$ and $(b)$ is from the fact that $|\max_x F(x) - \max_x G(x)| \leq \max_x |F(x) - G(x)|$.

From the definition of $\kappa_s^a = \sqrt{\frac{S \log \frac{SA}{\delta}}{2N(s,a)}}$, it holds that $\kappa_s^* = \sqrt{\frac{S \log \frac{SA}{\delta}}{2N(s,\pi^*(s))}} \leq \sqrt{\frac{8S \log \frac{4SA}{\delta} \log \frac{SA}{\delta}}{2N\mu(s,\pi^*(s))}}$ and hence

$$|\kappa_s^* \mathbf{Span}(V_{\tilde{\mathcal{P}}}^{\tilde{\pi}} - \lambda)| \leq \frac{1}{1-\gamma} \sqrt{\frac{8S \log \frac{4SA}{\delta} \log \frac{SA}{\delta}}{2N\mu(s,\pi^*(s))}}. \tag{30}$$

To further bound (28), we first note that

$$\frac{1}{1-\gamma} \sum_{s,a} d_{\mathsf{P}_{\tilde{V}}}^*(s,a) |\sigma_s^*(V_{\tilde{\mathcal{P}}}^{\tilde{\pi}}) - \hat{\sigma}_s^*(V_{\tilde{\mathcal{P}}}^{\tilde{\pi}})|$$

$$\leq \frac{1}{1-\gamma} \sum_{s,a} d_{\mathsf{P}_{\tilde{V}}}^*(s,a) \max_{0 \leq \lambda \leq \frac{1}{1-\gamma}} |\mathsf{P}_s^a[V_{\tilde{\mathcal{P}}}^{\tilde{\pi}}]_\lambda - \hat{\mathsf{P}}_s^a[V_{\tilde{\mathcal{P}}}^{\tilde{\pi}}]_\lambda|, \tag{31}$$

which is from the dual form of $\sigma_{\mathcal{P}}(V)$, i.e.,

$$\sigma_{\mathcal{P}_s^a}(V) = \max_{0 \le \lambda \le \frac{1}{1-\gamma}} \{\mathsf{P}_s^a[V]_\lambda - \mathbf{Span}([V]_\lambda)\}, \tag{32}$$

and $[V]_\lambda(s) = \min\{V(s), \lambda\}$. We then involve (20), and it holds that

$$\frac{1}{1-\gamma} \sum_{s,a} d^*_{\mathsf{P}_{\tilde{V}}}(s,a) |\sigma_s^*(V_{\tilde{\mathcal{P}}}^{\tilde{\pi}}) - \hat{\sigma}_s^*(V_{\tilde{\mathcal{P}}}^{\tilde{\pi}})|$$

$$\overset{(a)}{\le} \frac{1}{1-\gamma} \sum_{s,a} d^*_{\mathsf{P}_{\tilde{V}}}(s,a) \max_{0 \le \lambda \le \frac{1}{1-\gamma}} |\mathsf{P}_s^a[V_{\tilde{\mathcal{P}}}^{\tilde{\pi}}]_\lambda - \hat{\mathsf{P}}_s^a[V_{\tilde{\mathcal{P}}}^{\tilde{\pi}}]_\lambda|$$

$$\le \frac{1}{1-\gamma} \sum_{s,a} d^*_{\mathsf{P}_{\tilde{V}}}(s,a) \left( \sqrt{\frac{S \log \frac{SA}{\delta}}{2N(s,a)}} \frac{1}{1-\gamma} \right)$$

$$\le \frac{1}{(1-\gamma)^2} \sum_{s,a} d^*_{\mathsf{P}_{\tilde{V}}}(s,a) \left( \sqrt{\frac{8S \log \frac{SA}{\delta} \log \frac{4SA}{\delta}}{2N\mu(s,\pi^*(s))}} \right)$$

$$\le \frac{1}{(1-\gamma)^2} \sqrt{\frac{8S \log \frac{SA}{\delta} \log \frac{4SA}{\delta}}{2N\mu(s,\pi^*(s))}}. \tag{33}$$

Plugging this inequality to (27) and from the definition of $C^{\pi^*}$, we have that

$$V_{\mathcal{P}}^{\pi^*}(s) - V_{\tilde{\mathcal{P}}}^{\tilde{\pi}}(s) \le \frac{4S \log \frac{4SA}{\delta}}{(1-\gamma)^2} \sqrt{\frac{C^{\pi^*}}{N}}, \tag{34}$$

which completes the proof. $\square$

**Theorem 6.** *For the output policy $\tilde{\pi}$ of algorithm 1, with probability at least $1 - 4\delta$, it holds that*

$$V_{\mathcal{P}}^{\pi^*}(\rho) - V_{\tilde{\mathcal{P}}}^{\tilde{\pi}}(\rho) \le \tilde{\mathcal{O}} \left( \frac{(C^{\pi^*}S + \frac{1}{\mu_{\min}})}{N(1-\gamma)^2} + \sqrt{\frac{(C^{\pi^*}S + \frac{1}{\mu_{\min}})}{N(1-\gamma)^3}} \right). \tag{35}$$

*Proof.* Note that Lemma 8 of [36] states that with probability $1 - \delta$, for any $(s,a)$ pair,

$$N(s,a) \ge \frac{N\mu(s,a)}{8 \log \frac{4SA}{\delta}}. \tag{36}$$

We hence conduct our proof under the occurrence of this event. Note that for any $s$,

$$V_{\mathcal{P}}^{\pi^*}(s) - V_{\tilde{\mathcal{P}}}^{\tilde{\pi}}(s) = \underbrace{V_{\mathcal{P}}^{\pi^*}(s) - V_{\mathcal{P}}^{\tilde{\pi}}(s)}_{(A)} + \underbrace{V_{\mathcal{P}}^{\tilde{\pi}}(s) - V_{\tilde{\mathcal{P}}}^{\tilde{\pi}}(s)}_{(B)}. \tag{37}$$

From Lemma 2 and Lemma 3, the above bound can be bounded as

$$V_{\mathcal{P}}^{\pi^*}(s) - V_{\tilde{\mathcal{P}}}^{\tilde{\pi}}(s)$$

$$\le (32 + 4\sqrt{8}) \sqrt{\frac{16c_1 \log \frac{4SAN}{\delta} \log \frac{4SA}{\delta}}{(1-\gamma)^2 N\mu_{\min}}} \sqrt{\frac{1}{\max\{R, 1-\gamma\}}}$$

$$+ \frac{2(8K_1 + 16) \log \frac{4SA}{\delta}}{N\mu_{\min}(1-\gamma)^2}$$

$$+ 4\sqrt{\frac{16K_2^2 C^{\pi^*} S \log \frac{4SA}{\delta}}{\gamma^2(1-\gamma)^3 N}} + \frac{(128K_2^2 + 32K_1 + 32)C^{\pi^*} S \log \frac{4SA}{\delta}}{(1-\gamma)^2 N}$$

$$+ 4K_2 \sqrt{\frac{8C^{\pi^*} S \log \frac{4SA}{\delta}}{N}} \sqrt{\frac{2}{\gamma^2(1-\gamma)^2 \max\{(1-\gamma), R\}}}$$

$$= \tilde{\mathcal{O}} \left( \frac{(C^{\pi^*}S + \frac{1}{\mu_{\min}})}{N(1-\gamma)^2} + \sqrt{\frac{(C^{\pi^*}S + \frac{1}{\mu_{\min}})}{N(1-\gamma)^3}} \right), \tag{38}$$

which completes the proof. $\qquad \square$

**Lemma 2.** *(Bound on Term A) With probability at least $1 - 2\delta$, it holds that*

$$V_{\mathcal{P}}^{\pi^*}(s) - V_{\tilde{\mathcal{P}}}^{\tilde{\pi}}(s)$$

$$\leq 4 \sqrt{\frac{16 K_2^2 C^{\pi^*} S \log \frac{4SA}{\delta}}{\gamma^2 (1-\gamma)^3 N}} + \frac{(128 K_2^2 + 32 K_1 + 32) C^{\pi^*} S \log \frac{4SA}{\delta}}{(1-\gamma)^2 N}$$

$$+ 4 K_2 \sqrt{\frac{8 C^{\pi^*} S \log \frac{4SA}{\delta}}{N}} \sqrt{\frac{2}{\gamma^2 (1-\gamma)^2 \max\{(1-\gamma), R\}}}, \tag{39}$$

*Proof.* To bound term $(A)$, note that

$$V_{\mathcal{P}}^{\pi^*}(s) - V_{\tilde{\mathcal{P}}}^{\tilde{\pi}}(s)$$

$$= r(s, \pi^*(s)) + \gamma \sigma_s^*(V_{\mathcal{P}}^{\pi^*}) - \max_a \{ r(s,a) + \gamma \tilde{\sigma}_s^a(V_{\tilde{\mathcal{P}}}^{\tilde{\pi}}) \}$$

$$\overset{(a)}{\leq} r(s, \pi^*(s)) + \gamma \sigma_s^*(V_{\mathcal{P}}^{\pi^*}) - r(s, \pi^*(s)) - \gamma \tilde{\sigma}_s^*(V_{\tilde{\mathcal{P}}}^{\tilde{\pi}})$$

$$= \gamma \sigma_s^*(V_{\mathcal{P}}^{\pi^*}) - \gamma \tilde{\sigma}_s^*(V_{\tilde{\mathcal{P}}}^{\tilde{\pi}})$$

$$= \gamma \sigma_s^*(V_{\mathcal{P}}^{\pi^*}) - \gamma \sigma_s^*(V_{\tilde{\mathcal{P}}}^{\tilde{\pi}}) + \gamma \sigma_s^*(V_{\tilde{\mathcal{P}}}^{\tilde{\pi}}) - \gamma \tilde{\sigma}_s^*(V_{\tilde{\mathcal{P}}}^{\tilde{\pi}})$$

$$\leq \gamma (\mathsf{P}_{\tilde{V}})_s^*(V_{\mathcal{P}}^{\pi^*} - V_{\tilde{\mathcal{P}}}^{\tilde{\pi}}) + \gamma \sigma_s^*(V_{\tilde{\mathcal{P}}}^{\tilde{\pi}}) - \gamma \tilde{\sigma}_s^*(V_{\tilde{\mathcal{P}}}^{\tilde{\pi}}), \tag{40}$$

where $(a)$ is from $\tilde{\pi} = \arg\max_\pi V_{\tilde{\mathcal{P}}}^{\pi}$, and the last inequality is from the fact $\mathsf{P}_{\tilde{V}} \in \mathcal{P}_s^*$ and hence $\sigma_s^*(V_{\mathcal{P}}^{\pi^*}) \leq (\mathsf{P}_{\tilde{V}})_s^* V_{\mathcal{P}}^{\pi^*}$.

Applying (40) recursively implies

$$V_{\mathcal{P}}^{\pi^*}(s) - V_{\tilde{\mathcal{P}}}^{\tilde{\pi}}(s) \leq \frac{1}{1-\gamma} \sum_x d_{\mathsf{P}_{\tilde{V}}}^*(x) |\sigma_x^*(V_{\tilde{\mathcal{P}}}^{\tilde{\pi}}) - \tilde{\sigma}_x^*(V_{\tilde{\mathcal{P}}}^{\tilde{\pi}})|. \tag{41}$$

Note that

$$\sum_x d_{\mathsf{P}_{\tilde{V}}}^*(x) |\sigma_x^*(V_{\tilde{\mathcal{P}}}^{\tilde{\pi}}) - \tilde{\sigma}_x^*(V_{\tilde{\mathcal{P}}}^{\tilde{\pi}})|$$

$$= \sum_x d_{\mathsf{P}_{\tilde{V}}}^*(x) |\sigma_x^*(V_{\tilde{\mathcal{P}}}^{\tilde{\pi}}) - \hat{\sigma}_x^*(V_{\tilde{\mathcal{P}}}^{\tilde{\pi}}) + \hat{\sigma}_x^*(V_{\tilde{\mathcal{P}}}^{\tilde{\pi}}) - \tilde{\sigma}_x^*(V_{\tilde{\mathcal{P}}}^{\tilde{\pi}})|$$

$$\leq \sum_x d_{\mathsf{P}_{\tilde{V}}}^*(x) |\sigma_x^*(V_{\tilde{\mathcal{P}}}^{\tilde{\pi}}) - \hat{\sigma}_x^*(V_{\tilde{\mathcal{P}}}^{\tilde{\pi}})| + \sum_x d_{\mathsf{P}_{\tilde{V}}}^*(x) |\hat{\sigma}_x^*(V_{\tilde{\mathcal{P}}}^{\tilde{\pi}}) - \tilde{\sigma}_x^*(V_{\tilde{\mathcal{P}}}^{\tilde{\pi}})|$$

$$\leq \sum_x d_{\mathsf{P}_{\tilde{V}}}^*(x) |\sigma_x^*(V_{\tilde{\mathcal{P}}}^{\tilde{\pi}}) - \hat{\sigma}_x^*(V_{\tilde{\mathcal{P}}}^{\tilde{\pi}})| + \frac{16 C^{\pi^*} S \log \frac{4SA}{\delta}}{(1-\gamma)N}, \tag{42}$$

where the last inequality is from

$$\|\hat{\sigma}^*(V_{\tilde{\mathcal{P}}}^{\tilde{\pi}}) - \tilde{\sigma}^*(V_{\tilde{\mathcal{P}}}^{\tilde{\pi}})\|$$

$$\overset{(a)}{=} \max_s \left\{ \left| \max_{0 \leq \lambda \leq \frac{1}{1-\gamma}} \left\{ \hat{\mathsf{P}}_s^*(V_{\tilde{\mathcal{P}}}^{\tilde{\pi}} - \lambda) - R\mathbf{Span}(V_{\tilde{\mathcal{P}}}^{\tilde{\pi}} - \lambda) \right\} \right. \right.$$

$$\left. \left. - \max_{0 \leq \lambda \leq \frac{1}{1-\gamma}} \left\{ \hat{\mathsf{P}}_s^*(V_{\tilde{\mathcal{P}}}^{\tilde{\pi}} - \lambda) - (\kappa_s^* + R)\mathbf{Span}(V_{\tilde{\mathcal{P}}}^{\tilde{\pi}} - \lambda) \right\} \right| \right\}$$

$$\overset{(b)}{\leq} \max_s \left\{ \left| \max_{0 \leq \lambda \leq \frac{1}{1-\gamma}} \left\{ \hat{\mathsf{P}}_s^*(V_{\tilde{\mathcal{P}}}^{\tilde{\pi}} - \lambda) + R\mathbf{Span}(V_{\tilde{\mathcal{P}}}^{\tilde{\pi}} - \lambda) - \hat{\mathsf{P}}_s^*(V_{\tilde{\mathcal{P}}}^{\tilde{\pi}} - \lambda) - (\kappa_s^* + R)\mathbf{Span}(V_{\tilde{\mathcal{P}}}^{\tilde{\pi}} - \lambda) \right\} \right| \right\}$$

$$\leq \max_s \left\{ \max_{0 \leq \lambda \leq \frac{1}{1-\gamma}} |\kappa_s^* \mathbf{Span}(V_{\tilde{\mathcal{P}}}^{\tilde{\pi}} - \lambda)| \right\}, \tag{43}$$

where $(a)$ is from the dual form of $\sigma_{\mathcal{P}}(V)$ and $(b)$ is from the fact that $|\max_x F(x) - \max_x G(x)| \leq \max_x |F(x) - G(x)|$.

Since $N(s,a) \geq \frac{N\mu(s,a)}{8 \log \frac{4SA}{\delta}}$, from Assumption 1, we have that

$$\frac{1}{N(s,\pi^*(s))} \leq \frac{8 \log \frac{4SA}{\delta}}{N\mu(s,\pi^*(s))} \leq \frac{8C^{\pi^*} \log \frac{4SA}{\delta}}{\min\{d_{\mathsf{P}_{\tilde{V}}}^*(s,\pi^*(s)), \frac{1}{S}\}} \leq \frac{8C^{\pi^*} \log \frac{4SA}{\delta}}{N} \left( S + \frac{1}{d_{\mathsf{P}_{\tilde{V}}}^*(s,\pi^*(s))} \right). \tag{44}$$

From the definition of $\kappa_s^a = \frac{1}{N(s,a)}$, it holds that $\kappa_s^* = \frac{1}{N(s,\pi^*(s))} \leq \frac{8 \log \frac{4SA}{\delta}}{N\mu(s,\pi^*(s))} \leq \frac{8C^{\pi^*} \log \frac{4SA}{\delta}}{N} \left( S + \frac{1}{d_{\mathsf{P}_{\tilde{V}}}^*(s,\pi^*(s))} \right)$ and hence

$$\sum_{s,a} d_{\mathsf{P}_{\tilde{V}}}^*(s,a) \mathbf{Span}(V_{\tilde{\mathcal{P}}}^{\tilde{\pi}} - \lambda)\kappa_s^* \leq \sum_{s,a} d_{\mathsf{P}_{\tilde{V}}}^*(s,a) \frac{8C^{\pi^*} \log \frac{4SA}{\delta}}{(1-\gamma)N} \left( S + \frac{1}{d_{\mathsf{P}_{\tilde{V}}}^*(s,\pi^*(s))} \right) \leq \frac{16C^{\pi^*} S \log \frac{4SA}{\delta}}{(1-\gamma)N}. \tag{45}$$

To further bound (42), we first note that

$$\frac{1}{1-\gamma} \sum_{s,a} d_{\mathsf{P}_{\tilde{V}}}^*(s,a)|\sigma_s^*(V_{\tilde{\mathcal{P}}}^{\tilde{\pi}}) - \hat{\sigma}_s^*(V_{\tilde{\mathcal{P}}}^{\tilde{\pi}})|$$

$$\leq \frac{1}{1-\gamma} \sum_{s,a} d_{\mathsf{P}_{\tilde{V}}}^*(s,a) \max_{0 \leq \lambda \leq \frac{1}{1-\gamma}} |\mathsf{P}_s^a[V_{\tilde{\mathcal{P}}}^{\tilde{\pi}}]_\lambda - \hat{\mathsf{P}}_s^a[V_{\tilde{\mathcal{P}}}^{\tilde{\pi}}]_\lambda|, \tag{46}$$

which is from the dual form of $\sigma_{\mathcal{P}}(V)$, i.e.,

$$\sigma_{\mathcal{P}_s^a}(V) = \max_{0 \leq \lambda \leq \frac{1}{1-\gamma}} \{\mathsf{P}_s^a[V]_\lambda - \mathbf{Span}([V]_\lambda)\}, \tag{47}$$

and $[V]_\lambda(s) = \min\{V(s), \lambda\}$. We then utilize Lemma 15 and (44), and it holds that

$$\frac{1}{1-\gamma} \sum_{s,a} d_{\mathsf{P}_{\tilde{V}}}^*(s,a)|\sigma_s^*(V_{\tilde{\mathcal{P}}}^{\tilde{\pi}}) - \hat{\sigma}_s^*(V_{\tilde{\mathcal{P}}}^{\tilde{\pi}})|$$

$$\leq \frac{1}{1-\gamma} \sum_{s,a} d_{\mathsf{P}_{\tilde{V}}}^*(s,a) \max_{0 \leq \lambda \leq \frac{1}{1-\gamma}} |\mathsf{P}_s^a[V_{\tilde{\mathcal{P}}}^{\tilde{\pi}}]_\lambda - \hat{\mathsf{P}}_s^a[V_{\tilde{\mathcal{P}}}^{\tilde{\pi}}]_\lambda|$$

$$\leq \frac{1}{1-\gamma} \sum_{s,a} d_{\mathsf{P}_{\tilde{V}}}^*(s,a) \left( \frac{K_1}{(1-\gamma)N(s,a)} + K_2 \sqrt{\frac{\mathbf{Var}_{\mathsf{P}_s^a}(V_{\tilde{\mathcal{P}}}^{\tilde{\pi}})}{N(s,a)}} \right)$$

$$\leq \frac{1}{1-\gamma} \sum_{s,a} d_{\mathsf{P}_{\tilde{V}}}^*(s,a) \left( \frac{8K_1 C^{\pi^*} \log \frac{4SA}{\delta}}{(1-\gamma)N} \left( S + \frac{1}{d_{\mathsf{P}_{\tilde{V}}}^*(s,a)} \right) + K_2 \sqrt{\frac{\mathbf{Var}_{\mathsf{P}_s^a}(V_{\tilde{\mathcal{P}}}^{\tilde{\pi}})}{N}} \right)$$

$$\leq \frac{16K_1 C^{\pi^*} S \log \frac{4SA}{\delta}}{N(1-\gamma)^2} + \frac{K_2}{1-\gamma} \sum_{s,a} d_{\mathsf{P}_{\tilde{V}}}^*(s,a) \sqrt{\frac{\mathbf{Var}_{\mathsf{P}_s^a}(V_{\tilde{\mathcal{P}}}^{\tilde{\pi}})}{N(s,a)}}$$

$$\leq \frac{16K_1 C^{\pi^*} S \log \frac{4SA}{\delta}}{N(1-\gamma)^2} + \underbrace{\frac{K_2}{1-\gamma} \sum_{s,a} d_{\mathsf{P}_{\tilde{V}}}^*(s,a) \sqrt{\frac{\mathbf{Var}_{(\mathsf{P}_{\tilde{V}})_s^a}(V_{\tilde{\mathcal{P}}}^{\tilde{\pi}})}{N(s,a)}}}_{A_1}$$

$$+ \underbrace{\frac{K_2}{1-\gamma} \sum_{s,a} d_{\mathsf{P}_{\tilde{V}}}^*(s,a) \sqrt{\frac{|\mathbf{Var}_{\mathsf{P}_s^a}(V_{\tilde{\mathcal{P}}}^{\tilde{\pi}}) - \mathbf{Var}_{(\mathsf{P}_{\tilde{V}})_s^a}(V_{\tilde{\mathcal{P}}}^{\tilde{\pi}})|}{N(s,a)}}}_{A_2}. \tag{48}$$

We then bound the two terms as follows.

**Bound on Term $A_1$**

We first claim the following inequality:

$$V_{\tilde{\mathcal{P}}}^{\tilde{\pi}} - \gamma \mathsf{P}_{\tilde{V}}^* V_{\tilde{\mathcal{P}}}^{\tilde{\pi}} + 2|\gamma\tilde{\sigma}^*(V_{\tilde{\mathcal{P}}}^{\tilde{\pi}}) - \gamma\sigma^*(V_{\tilde{\mathcal{P}}}^{\tilde{\pi}})| \geq 0. \tag{49}$$

To prove (49), we note that

$$
\begin{aligned}
V_{\tilde{\mathcal{P}}}^{\tilde{\pi}}(s) &= \max_a Q_{\tilde{\mathcal{P}}}^{\tilde{\pi}}(s,a) \\
&\geq Q_{\tilde{\mathcal{P}}}^{\tilde{\pi}}(s, \pi^*(s)) \\
&= \hat{r}(s, \pi^*(s)) + \gamma\tilde{\sigma}_s^*(V_{\tilde{\mathcal{P}}}^{\tilde{\pi}}) \\
&= \hat{r}(s, \pi^*(s)) + \gamma(\mathsf{P}_{\tilde{V}})_s^* V_{\tilde{\mathcal{P}}}^{\tilde{\pi}} - \gamma(\mathsf{P}_{\tilde{V}})_s^* V_{\tilde{\mathcal{P}}}^{\tilde{\pi}} + \gamma\tilde{\sigma}_s^*(V_{\tilde{\mathcal{P}}}^{\tilde{\pi}}) \\
&= \hat{r}(s, \pi^*(s)) + \gamma(\mathsf{P}_{\tilde{V}})_s^* V_{\tilde{\mathcal{P}}}^{\tilde{\pi}} + \gamma\tilde{\sigma}_s^*(V_{\tilde{\mathcal{P}}}^{\tilde{\pi}}) - \gamma\sigma_s^*(V_{\tilde{\mathcal{P}}}^{\tilde{\pi}}) \\
&\geq \hat{r}(s, \pi^*(s)) + \gamma(\mathsf{P}_{\tilde{V}})_s^* V_{\tilde{\mathcal{P}}}^{\tilde{\pi}} - 2|\gamma\tilde{\sigma}_s^*(V_{\tilde{\mathcal{P}}}^{\tilde{\pi}}) - \gamma\sigma_s^*(V_{\tilde{\mathcal{P}}}^{\tilde{\pi}})|, \tag{50}
\end{aligned}
$$

and hence for any $s \in \mathcal{S}$,

$$V_{\tilde{\mathcal{P}}}^{\tilde{\pi}}(s) - \gamma(\mathsf{P}_{\tilde{V}})_s^* V_{\tilde{\mathcal{P}}}^{\tilde{\pi}} + 2|\gamma\tilde{\sigma}_s^*(V_{\tilde{\mathcal{P}}}^{\tilde{\pi}}) - \gamma\sigma_s^*(V_{\tilde{\mathcal{P}}}^{\tilde{\pi}})| \geq \hat{r}(s, \pi^*(s)) \geq 0, \tag{51}$$

which proves (49).

Now with (49), we first note that

$$
\begin{aligned}
&(V_{\tilde{\mathcal{P}}}^{\tilde{\pi}} \circ V_{\tilde{\mathcal{P}}}^{\tilde{\pi}}) - (\gamma\mathsf{P}_{\tilde{V}}^* V_{\tilde{\mathcal{P}}}^{\tilde{\pi}}) \circ (\gamma\mathsf{P}_{\tilde{V}}^* V_{\tilde{\mathcal{P}}}^{\tilde{\pi}}) \\
&= (V_{\tilde{\mathcal{P}}}^{\tilde{\pi}} - \gamma\mathsf{P}_{\tilde{V}}^* V_{\tilde{\mathcal{P}}}^{\tilde{\pi}}) \circ (V_{\tilde{\mathcal{P}}}^{\tilde{\pi}} + \gamma\mathsf{P}_{\tilde{V}}^* V_{\tilde{\mathcal{P}}}^{\tilde{\pi}}) \\
&\leq (V_{\tilde{\mathcal{P}}}^{\tilde{\pi}} - \gamma\mathsf{P}_{\tilde{V}}^* V_{\tilde{\mathcal{P}}}^{\tilde{\pi}} + 2|\gamma\tilde{\sigma}^*(V_{\tilde{\mathcal{P}}}^{\tilde{\pi}}) - \gamma\sigma^*(V_{\tilde{\mathcal{P}}}^{\tilde{\pi}})|) \circ (V_{\tilde{\mathcal{P}}}^{\tilde{\pi}} + \gamma\mathsf{P}_{\tilde{V}}^* V_{\tilde{\mathcal{P}}}^{\tilde{\pi}}) \\
&\leq \frac{2}{1-\gamma}(V_{\tilde{\mathcal{P}}}^{\tilde{\pi}} - \gamma\mathsf{P}_{\tilde{V}}^* V_{\tilde{\mathcal{P}}}^{\tilde{\pi}} + 2|\gamma\tilde{\sigma}^*(V_{\tilde{\mathcal{P}}}^{\tilde{\pi}}) - \gamma\sigma^*(V_{\tilde{\mathcal{P}}}^{\tilde{\pi}})|), \tag{52}
\end{aligned}
$$

where the last inequality is due to the fact $\|V_{\tilde{\mathcal{P}}}^{\tilde{\pi}} + \gamma\mathsf{P}_{\tilde{V}}^* V_{\tilde{\mathcal{P}}}^{\tilde{\pi}}\| \leq \frac{2}{1-\gamma}$ and (49).

We first have that

$$
\begin{aligned}
&\sum_{s\in\mathcal{S}} d_{\mathsf{P}_{\tilde{V}}}^*(s) \mathbf{Var}_{(\mathsf{P}_{\tilde{V}})_s^*}(V_{\tilde{\mathcal{P}}}^{\tilde{\pi}}) \\
&= \langle d_{\mathsf{P}_{\tilde{V}}}^*, \mathsf{P}_{\tilde{V}}(V_{\tilde{\mathcal{P}}}^{\tilde{\pi}} \circ V_{\tilde{\mathcal{P}}}^{\tilde{\pi}}) - (\mathsf{P}_{\tilde{V}} V_{\tilde{\mathcal{P}}}^{\tilde{\pi}}) \circ (\mathsf{P}_{\tilde{V}} V_{\tilde{\mathcal{P}}}^{\tilde{\pi}}) \rangle \\
&\overset{(a)}{\leq} \left\langle d_{\mathsf{P}_{\tilde{V}}}^*, \mathsf{P}_{\tilde{V}}(V_{\tilde{\mathcal{P}}}^{\tilde{\pi}} \circ V_{\tilde{\mathcal{P}}}^{\tilde{\pi}}) - \frac{1}{\gamma^2}(V_{\tilde{\mathcal{P}}}^{\tilde{\pi}} \circ V_{\tilde{\mathcal{P}}}^{\tilde{\pi}}) + \frac{2}{\gamma^2(1-\gamma)}(V_{\tilde{\mathcal{P}}}^{\tilde{\pi}} - \gamma\mathsf{P}_{\tilde{V}} V_{\tilde{\mathcal{P}}}^{\tilde{\pi}} + 2|\gamma\tilde{\sigma}^*(V_{\tilde{\mathcal{P}}}^{\tilde{\pi}}) - \gamma\sigma^*(V_{\tilde{\mathcal{P}}}^{\tilde{\pi}})|) \right\rangle \\
&\overset{(b)}{\leq} \left\langle d_{\mathsf{P}_{\tilde{V}}}^*, \mathsf{P}_{\tilde{V}}(V_{\tilde{\mathcal{P}}}^{\tilde{\pi}} \circ V_{\tilde{\mathcal{P}}}^{\tilde{\pi}}) - \frac{1}{\gamma}(V_{\tilde{\mathcal{P}}}^{\tilde{\pi}} \circ V_{\tilde{\mathcal{P}}}^{\tilde{\pi}}) + \frac{2}{\gamma^2(1-\gamma)}(I - \gamma\mathsf{P}_{\tilde{V}})V_{\tilde{\mathcal{P}}}^{\tilde{\pi}} + \frac{4}{\gamma^2(1-\gamma)}|\gamma\tilde{\sigma}^*(V_{\tilde{\mathcal{P}}}^{\tilde{\pi}}) - \gamma\sigma^*(V_{\tilde{\mathcal{P}}}^{\tilde{\pi}})|) \right\rangle \\
&= \left\langle d_{\mathsf{P}_{\tilde{V}}}^*, \frac{1}{\gamma}(\gamma\mathsf{P}_{\tilde{V}} - I)(V_{\tilde{\mathcal{P}}}^{\tilde{\pi}} \circ V_{\tilde{\mathcal{P}}}^{\tilde{\pi}}) + \frac{2}{\gamma^2(1-\gamma)}(I - \gamma\mathsf{P}_{\tilde{V}})V_{\tilde{\mathcal{P}}}^{\tilde{\pi}} + \frac{4}{\gamma^2(1-\gamma)}|\gamma\tilde{\sigma}^*(V_{\tilde{\mathcal{P}}}^{\tilde{\pi}}) - \gamma\sigma^*(V_{\tilde{\mathcal{P}}}^{\tilde{\pi}})|) \right\rangle \\
&= (d_{\mathsf{P}_{\tilde{V}}}^*)^\top (I - \gamma\mathsf{P}_{\tilde{V}}) \left(-\frac{1}{\gamma}(V_{\tilde{\mathcal{P}}}^{\tilde{\pi}} \circ V_{\tilde{\mathcal{P}}}^{\tilde{\pi}}) + \frac{2}{\gamma^2(1-\gamma)}V_{\tilde{\mathcal{P}}}^{\tilde{\pi}}\right) + \frac{4}{\gamma^2(1-\gamma)}\langle d_{\mathsf{P}_{\tilde{V}}}^*, |\gamma\tilde{\sigma}^*(V_{\tilde{\mathcal{P}}}^{\tilde{\pi}}) - \gamma\sigma^*(V_{\tilde{\mathcal{P}}}^{\tilde{\pi}})|\rangle \\
&\overset{(c)}{=} (1-\gamma)\rho^\top \left(-\frac{1}{\gamma}(V_{\tilde{\mathcal{P}}}^{\tilde{\pi}} \circ V_{\tilde{\mathcal{P}}}^{\tilde{\pi}}) + \frac{2}{\gamma^2(1-\gamma)}V_{\tilde{\mathcal{P}}}^{\tilde{\pi}}\right) + \frac{4}{\gamma^2(1-\gamma)}\langle d_{\mathsf{P}_{\tilde{V}}}^*, |\gamma\tilde{\sigma}^*(V_{\tilde{\mathcal{P}}}^{\tilde{\pi}}) - \gamma\sigma^*(V_{\tilde{\mathcal{P}}}^{\tilde{\pi}})|\rangle \\
&\leq \frac{2}{\gamma^2}\rho^\top V_{\tilde{\mathcal{P}}}^{\tilde{\pi}} + \frac{4}{\gamma^2(1-\gamma)}\langle d_{\mathsf{P}_{\tilde{V}}}^*, |\gamma\tilde{\sigma}^*(V_{\tilde{\mathcal{P}}}^{\tilde{\pi}}) - \gamma\sigma^*(V_{\tilde{\mathcal{P}}}^{\tilde{\pi}})|\rangle \\
&\leq \frac{2}{\gamma^2(1-\gamma)} + \frac{4}{\gamma^2(1-\gamma)}\langle d_{\mathsf{P}_{\tilde{V}}}^*, |\gamma\tilde{\sigma}^*(V_{\tilde{\mathcal{P}}}^{\tilde{\pi}}) - \gamma\sigma^*(V_{\tilde{\mathcal{P}}}^{\tilde{\pi}})|\rangle, \tag{53}
\end{aligned}
$$

where $(a)$ is from (52), $(b)$ is due to $\gamma < 1$, $(c)$ is from the definition of visitation distribution.

Note that by Cauchy's inequality, $\sum_s d^*_{\mathsf{P}_{\tilde{V}}}(s)\sqrt{\mathbf{Var}_{(\mathsf{P}_{\tilde{V}})^*_s}(V^{\tilde{\pi}}_{\tilde{\mathcal{P}}})} \leq \sqrt{\sum_s d^*_{\mathsf{P}_{\tilde{V}}}(s)\mathbf{Var}_{(\mathsf{P}_{\tilde{V}})^*_s}(V^{\tilde{\pi}}_{\tilde{\mathcal{P}}})}$, hence

$$\sum_{s,a} d^*_{\mathsf{P}_{\tilde{V}}}(s,a)\sqrt{\frac{\mathbf{Var}_{(\mathsf{P}_{\tilde{V}})^a_s}(V^{\tilde{\pi}}_{\tilde{\mathcal{P}}})}{N(s,a)}}$$

$$\leq \sum_{s,a} d^*_{\mathsf{P}_{\tilde{V}}}(s,a)\sqrt{\frac{8C^{\pi^*}\left(S + \frac{1}{d^*_{\mathsf{P}_{\tilde{V}}}(s,a)}\right)\log\frac{4SA}{\delta}\mathbf{Var}_{(\mathsf{P}_{\tilde{V}})^a_s}(V^{\tilde{\pi}}_{\tilde{\mathcal{P}}})}{N}}$$

$$= \sqrt{\frac{8C^{\pi^*}\log\frac{4SA}{\delta}}{N}}\sum_{s,a} d^*_{\mathsf{P}_{\tilde{V}}}(s,a)\sqrt{\left(S + \frac{1}{d^*_{\mathsf{P}_{\tilde{V}}}(s,a)}\right)\mathbf{Var}_{(\mathsf{P}_{\tilde{V}})^a_s}(V^{\tilde{\pi}}_{\tilde{\mathcal{P}}})}$$

$$\leq \sqrt{\frac{8C^{\pi^*}\log\frac{4SA}{\delta}}{N}}\sum_{s,a} d^*_{\mathsf{P}_{\tilde{V}}}(s,a)\sqrt{\mathbf{Var}_{(\mathsf{P}_{\tilde{V}})^a_s}(V^{\tilde{\pi}}_{\tilde{\mathcal{P}}})}\left(\sqrt{S} + \frac{1}{\sqrt{d^*_{\mathsf{P}_{\tilde{V}}}(s,a)}}\right)$$

$$\leq \sqrt{\frac{8C^{\pi^*}\log\frac{4SA}{\delta}}{N}}\sqrt{\sum_{s,a} d^*_{\mathsf{P}_{\tilde{V}}}(s,a)S\mathbf{Var}_{(\mathsf{P}_{\tilde{V}})^a_s}(V^{\tilde{\pi}}_{\tilde{\mathcal{P}}})} + \sqrt{\sum_{s,a} d^*_{\mathsf{P}_{\tilde{V}}}(s,a)\mathbf{Var}_{(\mathsf{P}_{\tilde{V}})^a_s}(V^{\tilde{\pi}}_{\tilde{\mathcal{P}}})}$$

$$\leq \sqrt{\frac{32C^{\pi^*}S\log\frac{4SA}{\delta}}{N}}\sqrt{\sum_{s,a} d^*_{\mathsf{P}_{\tilde{V}}}(s,a)\mathbf{Var}_{(\mathsf{P}_{\tilde{V}})^a_s}(V^{\tilde{\pi}}_{\tilde{\mathcal{P}}})}$$

$$\leq \sqrt{\frac{32C^{\pi^*}S\log\frac{4SA}{\delta}}{N}}\sqrt{\frac{2}{\gamma^2(1-\gamma)} + \frac{4}{\gamma^2(1-\gamma)}\langle d^*_{\mathsf{P}_{\tilde{V}}}, |\gamma\tilde{\sigma}^*(V^{\tilde{\pi}}_{\tilde{\mathcal{P}}}) - \gamma\sigma^*(V^{\tilde{\pi}}_{\tilde{\mathcal{P}}})|\rangle}$$

$$\leq \sqrt{\frac{64C^{\pi^*}S\log\frac{4SA}{\delta}}{\gamma^2(1-\gamma)N}} + \sqrt{\frac{128C^{\pi^*}S\log\frac{4SA}{\delta}}{\gamma(1-\gamma)N}}\sqrt{\langle d^*_{\mathsf{P}_{\tilde{V}}}, |\tilde{\sigma}^*(V^{\tilde{\pi}}_{\tilde{\mathcal{P}}}) - \sigma^*(V^{\tilde{\pi}}_{\tilde{\mathcal{P}}})|\rangle}$$

$$\leq \sqrt{\frac{64C^{\pi^*}S\log\frac{4SA}{\delta}}{\gamma^2(1-\gamma)N}} + \frac{64K_2C^{\pi^*}S\log\frac{4SA}{\delta}}{\gamma(1-\gamma)N} + \frac{1}{2K_2}\langle d^*_{\mathsf{P}_{\tilde{V}}}, |\tilde{\sigma}^*(V^{\tilde{\pi}}_{\tilde{\mathcal{P}}}) - \sigma^*(V^{\tilde{\pi}}_{\tilde{\mathcal{P}}})|\rangle$$

$$\leq \sqrt{\frac{64C^{\pi^*}S\log\frac{4SA}{\delta}}{\gamma^2(1-\gamma)N}} + \frac{64K_2C^{\pi^*}S\log\frac{4SA}{\delta}}{\gamma(1-\gamma)N} + \frac{1}{2K_2}\langle d^*_{\mathsf{P}_{\tilde{V}}}, |\tilde{\sigma}^*(V^{\tilde{\pi}}_{\tilde{\mathcal{P}}}) - \hat{\sigma}^*(V^{\tilde{\pi}}_{\tilde{\mathcal{P}}})|\rangle$$

$$\quad + \frac{1}{2K_2}\langle d^*_{\mathsf{P}_{\tilde{V}}}, |\sigma^*(V^{\tilde{\pi}}_{\tilde{\mathcal{P}}}) - \hat{\sigma}^*(V^{\tilde{\pi}}_{\tilde{\mathcal{P}}})|\rangle$$

$$\leq \sqrt{\frac{64C^{\pi^*}S\log\frac{4SA}{\delta}}{\gamma^2(1-\gamma)N}} + \frac{64C^{\pi^*}SK_2\log\frac{4SA}{\delta}}{\gamma(1-\gamma)N} + \frac{1}{2K_2}\langle d^*_{\mathsf{P}_{\tilde{V}}}, |\sigma^*(V^{\tilde{\pi}}_{\tilde{\mathcal{P}}}) - \hat{\sigma}^*(V^{\tilde{\pi}}_{\tilde{\mathcal{P}}})|\rangle$$

$$\quad + \frac{16C^{\pi^*}S\log\frac{4SA}{\delta}}{(1-\gamma)K_2N}, \tag{54}$$

where the first inequality is from (44) and the last inequality follows similarly as (42).

Hence, Term $A_1$ can be bounded as

$$A_1 \leq \sqrt{\frac{64K_2^2C^{\pi^*}S\log\frac{4SA}{\delta}}{\gamma^2(1-\gamma)^3N}} + \frac{64K_2^2C^{\pi^*}S\log\frac{4SA}{\delta}}{\gamma(1-\gamma)^2N\mu_{\min}} + \frac{1}{2(1-\gamma)}\langle d^*_{\mathsf{P}_{\tilde{V}}}, |\sigma^*(V^{\tilde{\pi}}_{\tilde{\mathcal{P}}}) - \hat{\sigma}^*(V^{\tilde{\pi}}_{\tilde{\mathcal{P}}})|\rangle$$

$$\quad + \frac{16C^{\pi^*}S\log\frac{4SA}{\delta}}{(1-\gamma)^2N}. \tag{55}$$

**Bound on Term $A_2$**

From Lemma 10, it is straightforward to see that for any transition kernel $q^a_s \in \mathcal{P}^a_s$,

$$|\mathbf{Var}_{\mathsf{P}^a_s}(V^{\tilde{\pi}}_{\tilde{\mathcal{P}}}) - \mathbf{Var}_{q^a_s}(V^{\tilde{\pi}}_{\tilde{\mathcal{P}}})|$$

$$= |\mathbf{Var}_{\mathsf{P}_s^a}(V_{\tilde{\mathcal{P}}}^{\tilde{\pi}} - \min_s V_{\tilde{\mathcal{P}}}^{\tilde{\pi}}(s)) - \mathbf{Var}_{q_s^a}(V_{\tilde{\mathcal{P}}}^{\tilde{\pi}} - \min_s V_{\tilde{\mathcal{P}}}^{\tilde{\pi}}(s))|$$

$$\leq \|\mathsf{P}_s^a - q_s^a\|_1 \|V_{\tilde{\mathcal{P}}}^{\tilde{\pi}} - \min_s V_{\tilde{\mathcal{P}}}^{\tilde{\pi}}(s)\|^2$$

$$\leq 2R(\mathbf{Span}(V_{\tilde{\mathcal{P}}}^{\tilde{\pi}}))^2$$

$$\leq \frac{2}{\gamma^2 \max\{(1-\gamma), R\}}. \tag{56}$$

Hence

$$\sum_{s,a} d_{\mathsf{P}_{\tilde{V}}}^*(s,a) \sqrt{\frac{|\mathbf{Var}_{\mathsf{P}_s^a}(V_{\tilde{\mathcal{P}}}^{\tilde{\pi}}) - \mathbf{Var}_{(\mathsf{P}_{\tilde{V}})_s^a}(V_{\tilde{\mathcal{P}}}^{\tilde{\pi}})|}{N(s,a)}}$$

$$\leq \sum_{s,a} d_{\mathsf{P}_{\tilde{V}}}^*(s,a) \sqrt{\frac{32 C^{\pi^*} S \log \frac{4SA}{\delta} |\mathbf{Var}_{\mathsf{P}_s^a}(V_{\tilde{\mathcal{P}}}^{\tilde{\pi}}) - \mathbf{Var}_{(\mathsf{P}_{\tilde{V}})_s^a}(V_{\tilde{\mathcal{P}}}^{\tilde{\pi}})|}{N}}$$

$$= \sqrt{\frac{32 C^{\pi^*} S \log \frac{4SA}{\delta}}{N}} \sum_{s,a} d_{\mathsf{P}_{\tilde{V}}}^*(s,a) \sqrt{|\mathbf{Var}_{\mathsf{P}_s^a}(V_{\tilde{\mathcal{P}}}^{\tilde{\pi}}) - \mathbf{Var}_{(\mathsf{P}_{\tilde{V}})_s^a}(V_{\tilde{\mathcal{P}}}^{\tilde{\pi}})|}$$

$$\leq \sqrt{\frac{32 C^{\pi^*} S \log \frac{4SA}{\delta}}{N}} \sum_{s,a} d_{\mathsf{P}_{\tilde{V}}}^*(s,a) \sqrt{\frac{2}{\gamma^2 \max\{(1-\gamma), R\}}}$$

$$\leq \sqrt{\frac{32 C^{\pi^*} S \log \frac{4SA}{\delta}}{N}} \sqrt{\frac{2}{\gamma^2 \max\{(1-\gamma), R\}}}. \tag{57}$$

Thus Term $A_2$ can be bounded as

$$A_2 \leq K_2 \sqrt{\frac{32 C^{\pi^*} S \log \frac{4SA}{\delta}}{N}} \sqrt{\frac{2}{\gamma^2 (1-\gamma)^2 \max\{(1-\gamma), R\}}}. \tag{58}$$

Combine the bounds we obtained for terms $A_1$ and $A_2$ in (55) and (58), we have that

$$\frac{1}{1-\gamma} \sum_{s,a} d_{\mathsf{P}_{\tilde{V}}}^*(s,a) |\sigma_s^*(V_{\tilde{\mathcal{P}}}^{\tilde{\pi}}) - \hat{\sigma}_s^*(V_{\tilde{\mathcal{P}}}^{\tilde{\pi}})|$$

$$\leq \sqrt{\frac{64 K_2^2 C^{\pi^*} S \log \frac{4SA}{\delta}}{\gamma^2 (1-\gamma)^3 N}} + \frac{64 K_2^2 C^{\pi^*} S \log \frac{4SA}{\delta}}{\gamma (1-\gamma)^2 N} + \frac{1}{2(1-\gamma)} \sum_{s,a} d_{\mathsf{P}_{\tilde{V}}}^*(s,a) |\sigma_s^*(V_{\tilde{\mathcal{P}}}^{\tilde{\pi}}) - \hat{\sigma}_s^*(V_{\tilde{\mathcal{P}}}^{\tilde{\pi}})|$$

$$+ \frac{16 C^{\pi^*} S \log \frac{4SA}{\delta}}{(1-\gamma)^2 N} + K_2 \sqrt{\frac{32 C^{\pi^*} S \log \frac{4SA}{\delta}}{N}} \sqrt{\frac{2}{\gamma^2 (1-\gamma)^2 \max\{(1-\gamma), R\}}} + \frac{16 K_1 C^{\pi^*} S \log \frac{4SA}{\delta}}{N(1-\gamma)^2}, \tag{59}$$

which further implies that

$$\frac{1}{1-\gamma} \sum_{s,a} d_{\mathsf{P}_{\tilde{V}}}^*(s,a) |\sigma_s^*(V_{\tilde{\mathcal{P}}}^{\tilde{\pi}}) - \hat{\sigma}_s^*(V_{\tilde{\mathcal{P}}}^{\tilde{\pi}})|$$

$$\leq 4\sqrt{\frac{16 K_2^2 C^{\pi^*} S \log \frac{4SA}{\delta}}{\gamma^2 (1-\gamma)^3 N}} + \frac{64 K_2^2 C^{\pi^*} S \log \frac{4SA}{\delta}}{\gamma (1-\gamma)^2 N} + \frac{(32 K_1 \log \frac{4SA}{\delta} + 16 \log \frac{4SA}{\delta}) C^{\pi^*} S}{(1-\gamma)^2 N}$$

$$+ 4K_2 \sqrt{\frac{8 C^{\pi^*} S \log \frac{4SA}{\delta}}{N}} \sqrt{\frac{2}{\gamma^2 (1-\gamma)^2 \max\{(1-\gamma), R\}}}. \tag{60}$$

Then, combine (41), (42) and the inequality above, we have that

$$V_{\mathcal{P}}^{\pi^*}(s) - V_{\tilde{\mathcal{P}}}^{\tilde{\pi}}(s)$$

$$\leq \frac{1}{1-\gamma} \sum_x d^*_{\mathsf{P}_{\tilde{V}}}(x) |\sigma^*_x(V^{\tilde{\pi}}_{\tilde{\mathcal{P}}}) - \tilde{\sigma}^*_x(V^{\tilde{\pi}}_{\tilde{\mathcal{P}}})|$$

$$\leq \frac{1}{1-\gamma} \sum_x d^*_{\mathsf{P}_{\tilde{V}}}(x) |\sigma^*_x(V^{\tilde{\pi}}_{\tilde{\mathcal{P}}}) - \hat{\sigma}^*_x(V^{\tilde{\pi}}_{\tilde{\mathcal{P}}})| + \frac{8 C^{\pi^*} S \log \frac{4SA}{\delta}}{(1-\gamma)^2 N}$$

$$\leq 4\sqrt{\frac{16 K_2^2 C^{\pi^*} S \log \frac{4SA}{\delta}}{\gamma^2 (1-\gamma)^3 N}} + \frac{(128 K_2^2 + 32 K_1 + 32) C^{\pi^*} S \log \frac{4SA}{\delta}}{(1-\gamma)^2 N}$$

$$+ 4 K_2 \sqrt{\frac{8 C^{\pi^*} S \log \frac{4SA}{\delta}}{N}} \sqrt{\frac{2}{\gamma^2 (1-\gamma)^2 \max\{(1-\gamma), R\}}}, \tag{61}$$

which completes the proof.

$\square$

**Lemma 3.** *(Bound on Term B) If* $N \geq \frac{196 c_1 \log \frac{4SAN}{\delta} \log \frac{4SA}{\delta}}{(1-\gamma)^2 N \mu_{\min}}$, *then with probability at least* $1 - 2\delta$, *it holds that*

$$B \leq (32 + 4\sqrt{8}) \sqrt{\frac{16 c_1 \log \frac{4SAN}{\delta} \log \frac{4SA}{\delta}}{(1-\gamma)^2 N \mu_{\min}}} \sqrt{\frac{1}{\max\{R, 1-\gamma\}}}$$

$$+ \frac{2(8 K_1 + 16) \log \frac{4SA}{\delta}}{N \mu_{\min} (1-\gamma)^2}. \tag{62}$$

*Proof.* From the robust Bellman equation, it holds that

$$V^{\tilde{\pi}}_{\tilde{\mathcal{P}}}(s) - V^{\tilde{\pi}}_{\mathcal{P}}(s)$$
$$= \hat{r}(s, \tilde{\pi}(s)) + \gamma \tilde{\sigma}^{\tilde{\pi}}_s(V^{\tilde{\pi}}_{\tilde{\mathcal{P}}}) - \hat{r}(s, \tilde{\pi}(s)) - \gamma \sigma^{\tilde{\pi}}_s(V^{\tilde{\pi}}_{\mathcal{P}})$$
$$= \gamma(\tilde{\sigma}^{\tilde{\pi}}_s(V^{\tilde{\pi}}_{\tilde{\mathcal{P}}}) - \sigma^{\tilde{\pi}}_s(V^{\tilde{\pi}}_{\mathcal{P}}))$$
$$= \gamma(\tilde{\sigma}^{\tilde{\pi}}_s(V^{\tilde{\pi}}_{\tilde{\mathcal{P}}}) - (\mathsf{P}_V)^{\tilde{\pi}(s)}_s V^{\tilde{\pi}}_{\tilde{\mathcal{P}}} + (\mathsf{P}_V)^{\tilde{\pi}(s)}_s V^{\tilde{\pi}}_{\tilde{\mathcal{P}}} - \sigma^{\tilde{\pi}}_s(V^{\tilde{\pi}}_{\mathcal{P}}))$$
$$= \gamma(\mathsf{P}_V)^{\tilde{\pi}(s)}_s(V^{\tilde{\pi}}_{\tilde{\mathcal{P}}} - V^{\tilde{\pi}}_{\mathcal{P}}) + \gamma(\tilde{\sigma}^{\tilde{\pi}}_s(V^{\tilde{\pi}}_{\tilde{\mathcal{P}}}) - (\mathsf{P}_V)^{\tilde{\pi}(s)}_s V^{\tilde{\pi}}_{\tilde{\mathcal{P}}})$$
$$= \frac{\gamma}{1-\gamma} \sum_x d^{\tilde{\pi}}_{\mathsf{P}_V}(x)(\tilde{\sigma}^{\tilde{\pi}}_x(V^{\tilde{\pi}}_{\tilde{\mathcal{P}}}) - (\mathsf{P}_V)^{\tilde{\pi}(x)}_x V^{\tilde{\pi}}_{\tilde{\mathcal{P}}})$$
$$\leq \frac{\gamma}{1-\gamma} \sum_x d^{\tilde{\pi}}_{\mathsf{P}_V}(x)(\tilde{\sigma}^{\tilde{\pi}}_x(V^{\tilde{\pi}}_{\tilde{\mathcal{P}}}) - (\mathsf{P}_{\tilde{V}})^{\tilde{\pi}(x)}_x V^{\tilde{\pi}}_{\tilde{\mathcal{P}}}), \tag{63}$$

where $(\mathsf{P}_V)^{\tilde{\pi}(s)}_s = \arg\min_{q \in \mathcal{P}^{\tilde{\pi}(s)}_s} q V^{\tilde{\pi}}_{\tilde{\mathcal{P}}}$, and $(\mathsf{P}_{\tilde{V}})^{\tilde{\pi}(s)}_s = \arg\min_{q \in \mathcal{P}^{\tilde{\pi}(s)}_s} q V^{\tilde{\pi}}_{\tilde{\mathcal{P}}}$, and the last inequality is from $(\mathsf{P}_V)^{\tilde{\pi}(s)}_s \in \mathcal{P}^{\pi^*(s)}_s$ and $(\mathsf{P}_{\tilde{V}})^{\tilde{\pi}(s)}_s V^{\tilde{\pi}}_{\tilde{\mathcal{P}}} = \sigma_{\mathcal{P}^{\tilde{\pi}(s)}_s}(V^{\tilde{\pi}}_{\tilde{\mathcal{P}}}) \leq (\mathsf{P}_V)^{\tilde{\pi}(s)}_s V^{\tilde{\pi}}_{\tilde{\mathcal{P}}}$.

On the other hand,

$$V^{\tilde{\pi}}_{\tilde{\mathcal{P}}}(s) - V^{\tilde{\pi}}_{\mathcal{P}}(s)$$
$$= \hat{r}(s, \tilde{\pi}(s)) + \gamma \tilde{\sigma}^{\tilde{\pi}}_s(V^{\tilde{\pi}}_{\tilde{\mathcal{P}}}) - \hat{r}(s, \tilde{\pi}(s)) - \gamma \sigma^{\tilde{\pi}}_s(V^{\tilde{\pi}}_{\mathcal{P}})$$
$$= \gamma(\tilde{\sigma}^{\tilde{\pi}}_s(V^{\tilde{\pi}}_{\tilde{\mathcal{P}}}) - \sigma^{\tilde{\pi}}_s(V^{\tilde{\pi}}_{\mathcal{P}}))$$
$$= \gamma(\tilde{\sigma}^{\tilde{\pi}}_s(V^{\tilde{\pi}}_{\tilde{\mathcal{P}}}) - (\mathsf{P}_{\tilde{V}})^{\tilde{\pi}(s)}_s V^{\tilde{\pi}}_{\tilde{\mathcal{P}}} + (\mathsf{P}_{\tilde{V}})^{\tilde{\pi}(s)}_s V^{\tilde{\pi}}_{\tilde{\mathcal{P}}} - \sigma^{\tilde{\pi}}_s(V^{\tilde{\pi}}_{\mathcal{P}}))$$
$$\overset{(a)}{\geq} \gamma(\tilde{\sigma}^{\tilde{\pi}}_s(V^{\tilde{\pi}}_{\tilde{\mathcal{P}}}) - (\mathsf{P}_{\tilde{V}})^{\tilde{\pi}(s)}_s V^{\tilde{\pi}}_{\tilde{\mathcal{P}}} + (\mathsf{P}_{\tilde{V}})^{\tilde{\pi}(s)}_s V^{\tilde{\pi}}_{\tilde{\mathcal{P}}} - (\mathsf{P}_{\tilde{V}})^{\tilde{\pi}(s)}_s V^{\tilde{\pi}}_{\mathcal{P}})$$
$$= \gamma(\mathsf{P}_{\tilde{V}})^{\tilde{\pi}(s)}_s(V^{\tilde{\pi}}_{\tilde{\mathcal{P}}} - V^{\tilde{\pi}}_{\mathcal{P}}) + \gamma(\tilde{\sigma}^{\tilde{\pi}}_s(V^{\tilde{\pi}}_{\tilde{\mathcal{P}}}) - (\mathsf{P}_{\tilde{V}})^{\tilde{\pi}(s)}_s V^{\tilde{\pi}}_{\tilde{\mathcal{P}}})$$
$$= \frac{\gamma}{1-\gamma} \sum_x d^{\tilde{\pi}}_{\mathsf{P}_{\tilde{V}}}(x)(\tilde{\sigma}^{\tilde{\pi}}_x(V^{\tilde{\pi}}_{\tilde{\mathcal{P}}}) - (\mathsf{P}_{\tilde{V}})^{\tilde{\pi}(x)}_x V^{\tilde{\pi}}_{\tilde{\mathcal{P}}}), \tag{64}$$

where $(a)$ is from the fact that $\mathsf{P}_{\tilde{V}} \in \mathcal{P}$ and hence $\sigma^{\tilde{\pi}}_s(V^{\tilde{\pi}}_{\mathcal{P}}) \leq (\mathsf{P}_{\tilde{V}})^{\tilde{\pi}(s)}_s V^{\tilde{\pi}}_{\mathcal{P}}$.

Hence we have that

$$\|V_{\tilde{\mathcal{P}}}^{\tilde{\pi}}(s) - V_{\mathcal{P}}^{\tilde{\pi}}(s)\| \leq \max\left\{ \underbrace{\left| \frac{\gamma}{1-\gamma} \sum_x d_{\mathsf{P}_V}^{\tilde{\pi}}(x)(\tilde{\sigma}_x^{\tilde{\pi}}(V_{\tilde{\mathcal{P}}}^{\tilde{\pi}}) - (\mathsf{P}_{\tilde{V}})_x^{\tilde{\pi}(x)} V_{\tilde{\mathcal{P}}}^{\tilde{\pi}}) \right|}_{I}, \right.$$
$$\left. \underbrace{\left| \frac{\gamma}{1-\gamma} \sum_x d_{\mathsf{P}_{\tilde{V}}}^{\tilde{\pi}}(x)(\tilde{\sigma}_x^{\tilde{\pi}}(V_{\tilde{\mathcal{P}}}^{\tilde{\pi}}) - (\mathsf{P}_{\tilde{V}})_x^{\tilde{\pi}(x)} V_{\tilde{\mathcal{P}}}^{\tilde{\pi}}) \right|}_{II} \right\}. \tag{65}$$

We then bound terms $(I)$ and $(II)$. Note that the only difference between the two terms are the visitation distributions, i.e, $d_{\mathsf{P}_V}^{\tilde{\pi}}$ and $d_{\mathsf{P}_{\tilde{V}}}^{\tilde{\pi}}$.

**Bound on Term** $(I)$

We first note that

$$\begin{aligned}
&|\tilde{\sigma}_s^{\tilde{\pi}}(V_{\tilde{\mathcal{P}}}^{\tilde{\pi}}) - (\mathsf{P}_{\tilde{V}})_s^{\tilde{\pi}(s)} V_{\tilde{\mathcal{P}}}^{\tilde{\pi}}| \\
&= |\tilde{\sigma}_s^{\tilde{\pi}}(V_{\tilde{\mathcal{P}}}^{\tilde{\pi}}) - \sigma_s^{\tilde{\pi}(s)}(V_{\tilde{\mathcal{P}}}^{\tilde{\pi}})| \\
&\leq \underbrace{|\tilde{\sigma}_s^{\tilde{\pi}}(V_{\tilde{\mathcal{P}}}^{\tilde{\pi}}) - \hat{\sigma}_s^{\tilde{\pi}}(V_{\tilde{\mathcal{P}}}^{\tilde{\pi}})|}_{(B_1)} + \underbrace{|\hat{\sigma}_s^{\tilde{\pi}}(V_{\tilde{\mathcal{P}}}^{\tilde{\pi}}) - \sigma_s^{\tilde{\pi}}(V_{\tilde{\mathcal{P}}}^{\tilde{\pi}})|}_{(B_2)},
\end{aligned} \tag{66}$$

hence it is sufficient to bound $B_1, B_2$.

To bound $B_1$, note that

$$\begin{aligned}
&|\tilde{\sigma}_s^{\tilde{\pi}}(V_{\tilde{\mathcal{P}}}^{\tilde{\pi}}) - \hat{\sigma}_s^{\tilde{\pi}}(V_{\tilde{\mathcal{P}}}^{\tilde{\pi}})| \\
&\overset{(a)}{=} |\max_\lambda\{\hat{\mathsf{P}}_s^{\tilde{\pi}(s)}(V_{\tilde{\mathcal{P}}}^{\tilde{\pi}} - \lambda) - (R + \kappa_s^{\tilde{\pi}(s)})\mathbf{Span}(V_{\tilde{\mathcal{P}}}^{\tilde{\pi}} - \lambda)\} - \max_\lambda\{\hat{\mathsf{P}}_s^{\tilde{\pi}(s)}(V_{\tilde{\mathcal{P}}}^{\tilde{\pi}} - \lambda) - R\mathbf{Span}(V_{\tilde{\mathcal{P}}}^{\tilde{\pi}} - \lambda)\}| \\
&\leq \max_\lambda\{\kappa_s^{\tilde{\pi}(s)}\mathbf{Span}(V_{\tilde{\mathcal{P}}}^{\tilde{\pi}} - \lambda)\} \\
&\leq \frac{\kappa_s^{\tilde{\pi}(s)}}{1-\gamma},
\end{aligned} \tag{67}$$

where $(a)$ is from the dual form of the support function [13] and the last inequality is because $\mathbf{Span}(V) \leq \frac{1}{1-\gamma}$ for any $0 \leq V \leq \frac{1}{1-\gamma}$.

We then bound $B_2$. Similarly from the dual form, it holds that

$$|\hat{\sigma}_s^{\tilde{\pi}}(V_{\tilde{\mathcal{P}}}^{\tilde{\pi}}) - \sigma_s^{\tilde{\pi}(s)}(V_{\tilde{\mathcal{P}}}^{\tilde{\pi}})| \leq \max_\lambda\{|(\hat{\mathsf{P}}_s^{\tilde{\pi}(s)} - \mathsf{P}_s^{\tilde{\pi}(s)})(V_{\tilde{\mathcal{P}}}^{\tilde{\pi}} - \lambda)\}. \tag{68}$$

We then apply Lemma 15, it holds that

$$\begin{aligned}
&\max_\lambda\{|(\hat{\mathsf{P}}_s^{\tilde{\pi}(s)} - \mathsf{P}_s^{\tilde{\pi}(s)})(V_{\tilde{\mathcal{P}}}^{\tilde{\pi}} - \lambda)\} \\
&\leq \frac{K_1}{N(s,\tilde{\pi}(s))(1-\gamma)} + \sqrt{\frac{c_1 \log(\frac{4SAN^2}{\delta})\mathbf{Var}_{\mathsf{P}_s^{\tilde{\pi}(s)}}(V_{\tilde{\mathcal{P}}}^{\tilde{\pi}})}{N(s,\tilde{\pi}(s))}}
\end{aligned} \tag{69}$$

with probability at least $1 - \delta$ and $K_1 = \left(2 + c_2 \log\left(\frac{4SAN^2}{\delta}\right) + \sqrt{2c_1 \log\left(\frac{4SAN^2}{\delta}\right)}\right)$.

Note that for any $s, a$,

$$\begin{aligned}
&\mathbf{Var}_{\mathsf{P}_s^a}(V_{\tilde{\mathcal{P}}}^{\tilde{\pi}}) \\
&= \mathbf{Var}_{(\mathsf{P}_V)_s^a}(V_{\tilde{\mathcal{P}}}^{\tilde{\pi}}) + (\mathbf{Var}_{(\mathsf{P}_V)_s^a}(V_{\tilde{\mathcal{P}}}^{\tilde{\pi}}) - \mathbf{Var}_{(\mathsf{P}_V)_s^a}(V_{\tilde{\mathcal{P}}}^{\tilde{\pi}})) + (\mathbf{Var}_{\mathsf{P}_s^a}(V_{\tilde{\mathcal{P}}}^{\tilde{\pi}}) - \mathbf{Var}_{(\mathsf{P}_V)_s^a}(V_{\tilde{\mathcal{P}}}^{\tilde{\pi}})). \tag{70}
\end{aligned}$$

Hence

$$\sqrt{\mathbf{Var}_{\mathsf{P}_s^a}(V_{\tilde{\mathcal{P}}}^{\tilde{\pi}})} \leq \sqrt{\mathbf{Var}_{(\mathsf{P}_V)_s^a}(V_{\tilde{\mathcal{P}}}^{\tilde{\pi}})} + \sqrt{|\mathbf{Var}_{(\mathsf{P}_V)_s^a}(V_{\tilde{\mathcal{P}}}^{\tilde{\pi}}) - \mathbf{Var}_{(\mathsf{P}_V)_s^a}(V_{\tilde{\mathcal{P}}}^{\tilde{\pi}})|}$$

$$+ \sqrt{|\mathbf{Var}_{\mathsf{P}_s^a}(V_{\tilde{\mathcal{P}}}^{\tilde{\pi}}) - \mathbf{Var}_{(\mathsf{P}_V)_s^a}(V_{\tilde{\mathcal{P}}}^{\tilde{\pi}})|}. \tag{71}$$

Firstly, Lemma 7 of [38] and Lemma 10 imply that

$$\sum_x d_{\mathsf{P}_V}^{\tilde{\pi}}(x)\sqrt{\mathbf{Var}_{(\mathsf{P}_V)_x^a}(V_{\tilde{\mathcal{P}}}^{\tilde{\pi}})} \leq \sqrt{\frac{8\mathbf{Span}(V_{\tilde{\mathcal{P}}}^{\tilde{\pi}})}{\gamma^2}} \leq \sqrt{\frac{64}{\max\{R, 1-\gamma\}}}. \tag{72}$$

To bound $\sqrt{|\mathbf{Var}_{(\mathsf{P}_V)_s^a}(V_{\tilde{\mathcal{P}}}^{\tilde{\pi}}) - \mathbf{Var}_{(\mathsf{P}_V)_s^a}(V_{\tilde{\mathcal{P}}}^{\tilde{\pi}})|}$, note that it holds that $|\mathbf{Var}_{(\mathsf{P}_V)_s^a}(V_{\tilde{\mathcal{P}}}^{\tilde{\pi}}) - \mathbf{Var}_{(\mathsf{P}_V)_s^a}(V_{\tilde{\mathcal{P}}}^{\tilde{\pi}})| \leq \|V_{\tilde{\mathcal{P}}}^{\tilde{\pi}} - V_{\tilde{\mathcal{P}}}^{\tilde{\pi}}\|^2$, thus

$$\sum_x d_{\mathsf{P}_V}^{\tilde{\pi}}(x)\sqrt{|\mathbf{Var}_{(\mathsf{P}_V)_x^a}(V_{\tilde{\mathcal{P}}}^{\tilde{\pi}}) - \mathbf{Var}_{(\mathsf{P}_V)_x^a}(V_{\tilde{\mathcal{P}}}^{\tilde{\pi}})|} \leq \|V_{\tilde{\mathcal{P}}}^{\tilde{\pi}} - V_{\tilde{\mathcal{P}}}^{\tilde{\pi}}\|. \tag{73}$$

And the last term can be bounded as

$$|\mathbf{Var}_{\mathsf{P}_s^a}(V_{\tilde{\mathcal{P}}}^{\tilde{\pi}}) - \mathbf{Var}_{(\mathsf{P}_V)_s^a}(V_{\tilde{\mathcal{P}}}^{\tilde{\pi}})| \leq \frac{2}{\gamma^2 \max\{R, 1-\gamma\}} \tag{74}$$

using (56), and hence

$$\sum_x d_{\mathsf{P}_V}^{\tilde{\pi}}(x)\sqrt{|\mathbf{Var}_{\mathsf{P}_x^a}(V_{\tilde{\mathcal{P}}}^{\tilde{\pi}}) - \mathbf{Var}_{(\mathsf{P}_V)_x^a}(V_{\tilde{\mathcal{P}}}^{\tilde{\pi}})|} \leq \sqrt{\frac{8}{\max\{R, 1-\gamma\}}}. \tag{75}$$

Hence by combining (72),(73) and (75) we have that

$$\sum_x d_{\mathsf{P}_V}^{\tilde{\pi}}(x)\sqrt{\frac{\mathbf{Var}_{\mathsf{P}_s^{\tilde{\pi}(s)}}(V_{\tilde{\mathcal{P}}}^{\tilde{\pi}})}{N(s, \tilde{\pi}(s))}}$$

$$\leq \sum_x d_{\mathsf{P}_V}^{\tilde{\pi}}(x)\sqrt{\frac{8\log\frac{4SA}{\delta}\mathbf{Var}_{\mathsf{P}_s^{\tilde{\pi}(s)}}(V_{\tilde{\mathcal{P}}}^{\tilde{\pi}})}{N\mu_{\min}}}$$

$$\leq \sqrt{\frac{8\log\frac{4SA}{\delta}}{N\mu_{\min}}}\left(\|V_{\tilde{\mathcal{P}}}^{\tilde{\pi}} - V_{\tilde{\mathcal{P}}}^{\tilde{\pi}}\| + (8 + \sqrt{8})\sqrt{\frac{1}{\max\{R, 1-\gamma\}}}\right) \tag{76}$$

Thus plug the inequality above and the bound of $B_1$ in (63), we have that

$$I \leq \frac{\gamma}{1-\gamma}\sum_x d_{\mathsf{P}_V}^{\tilde{\pi}}(x)(\tilde{\sigma}_s^{\tilde{\pi}}(V_{\tilde{\mathcal{P}}}^{\tilde{\pi}}) - (\mathsf{P}_{\tilde{V}})_s^{\tilde{\pi}(s)}V_{\tilde{\mathcal{P}}}^{\tilde{\pi}})$$

$$\leq \sqrt{\frac{16c_1\log\frac{4SAN}{\delta}\log\frac{4SA}{\delta}}{N\mu_{\min}(1-\gamma)^2}}\left(\|V_{\tilde{\mathcal{P}}}^{\tilde{\pi}} - V_{\tilde{\mathcal{P}}}^{\tilde{\pi}}\| + (8 + \sqrt{8})\sqrt{\frac{1}{\max\{R, 1-\gamma\}}}\right)$$

$$+ \frac{4K_1\log\frac{4SA}{\delta}}{N\mu_{\min}(1-\gamma)^2} + \frac{8\log\frac{4SA}{\delta}}{(1-\gamma)^2 N\mu_{\min}} \tag{77}$$

**Bound on Term $II$**

Similarly to the analysis in bounding the Term (I), we have that

$$|\tilde{\sigma}_s^{\tilde{\pi}}(V_{\tilde{\mathcal{P}}}^{\tilde{\pi}}) - (\mathsf{P}_{\tilde{V}})_s^{\tilde{\pi}(s)}V_{\tilde{\mathcal{P}}}^{\tilde{\pi}}| \leq \underbrace{|\tilde{\sigma}_s^{\tilde{\pi}}(V_{\tilde{\mathcal{P}}}^{\tilde{\pi}}) - \hat{\sigma}_s^{\tilde{\pi}}(V_{\tilde{\mathcal{P}}}^{\tilde{\pi}})|}_{(B_1)} + \underbrace{|\hat{\sigma}_s^{\tilde{\pi}}(V_{\tilde{\mathcal{P}}}^{\tilde{\pi}}) - \sigma_s^{\tilde{\pi}}(V_{\tilde{\mathcal{P}}}^{\tilde{\pi}})|}_{(B_2)}, \tag{78}$$

where $B_1$ can be bounded as

$$|\tilde{\sigma}_s^{\tilde{\pi}}(V_{\tilde{\mathcal{P}}}^{\tilde{\pi}}) - \hat{\sigma}_s^{\tilde{\pi}}(V_{\tilde{\mathcal{P}}}^{\tilde{\pi}})| \leq \frac{\kappa_s^{\tilde{\pi}(s)}}{1-\gamma}. \tag{79}$$

The bound on $B_2$ similarly follows as in Term $(I)$. Note that

$$|\hat{\sigma}_s^{\tilde{\pi}}(V_{\tilde{\mathcal{P}}}^{\tilde{\pi}}) - \sigma_s^{\tilde{\pi}(s)}(V_{\tilde{\mathcal{P}}}^{\tilde{\pi}})| \le \frac{K_1}{N(s, \tilde{\pi}(s))(1-\gamma)} + \sqrt{\frac{c_1 \log(\frac{4SAN^2}{\delta}) \mathbf{Var}_{\mathsf{P}_s^{\tilde{\pi}(s)}}(V_{\tilde{\mathcal{P}}}^{\tilde{\pi}})}{N(s, \tilde{\pi}(s))}} \tag{80}$$

with probability at least $1 - \delta$ and $K_1 = \left( 2 + c_2 \log\left( \frac{4SAN^2}{\delta} \right) + \sqrt{2c_1 \log\left( \frac{4SAN^2}{\delta} \right)} \right).$

Note that for any $s, a$,

$$\mathbf{Var}_{\mathsf{P}_s^a}(V_{\tilde{\mathcal{P}}}^{\tilde{\pi}}) = \mathbf{Var}_{(\mathsf{P}_{\tilde{V}})_s^a}(V_{\tilde{\mathcal{P}}}^{\tilde{\pi}}) + (\mathbf{Var}_{\mathsf{P}_s^a}(V_{\tilde{\mathcal{P}}}^{\tilde{\pi}}) - \mathbf{Var}_{(\mathsf{P}_{\tilde{V}})_s^a}(V_{\tilde{\mathcal{P}}}^{\tilde{\pi}})). \tag{81}$$

Hence

$$\sqrt{\mathbf{Var}_{\mathsf{P}_s^a}(V_{\tilde{\mathcal{P}}}^{\tilde{\pi}})} \le \sqrt{\mathbf{Var}_{(\mathsf{P}_{\tilde{V}})_s^a}(V_{\tilde{\mathcal{P}}}^{\tilde{\pi}})} + \sqrt{\mathbf{Var}_{\mathsf{P}_s^a}(V_{\tilde{\mathcal{P}}}^{\tilde{\pi}}) - \mathbf{Var}_{(\mathsf{P}_{\tilde{V}})_s^a}(V_{\tilde{\mathcal{P}}}^{\tilde{\pi}})}. \tag{82}$$

Similarly, Lemma 13 of [38] and Lemma 10 imply that

$$\sum_s d_{\mathsf{P}_{\tilde{V}}}^{\tilde{\pi}}(s) \sqrt{\mathbf{Var}_{(\mathsf{P}_{\tilde{V}})_s^a}(V_{\tilde{\mathcal{P}}}^{\tilde{\pi}})} \le \sqrt{\frac{8\mathbf{Span}(V_{\tilde{\mathcal{P}}}^{\tilde{\pi}})}{\gamma^2}} \le \sqrt{\frac{64}{\max\{R, 1-\gamma\}}}. \tag{83}$$

And to bound $\sqrt{\mathbf{Var}_{\mathsf{P}_s^a}(V_{\tilde{\mathcal{P}}}^{\tilde{\pi}}) - \mathbf{Var}_{(\mathsf{P}_{\tilde{V}})_s^a}(V_{\tilde{\mathcal{P}}}^{\tilde{\pi}})}$, note that

$$|\mathbf{Var}_{\mathsf{P}_s^a}(V_{\tilde{\mathcal{P}}}^{\tilde{\pi}}) - \mathbf{Var}_{(\mathsf{P}_{\tilde{V}})_s^a}(V_{\tilde{\mathcal{P}}}^{\tilde{\pi}})| \le \frac{2}{\gamma^2 \max\{R, 1-\gamma\}} \tag{84}$$

from (56), and hence

$$\sum_s d_{\mathsf{P}_{\tilde{V}}}^{\tilde{\pi}}(s) \sqrt{\mathbf{Var}_{\mathsf{P}_s^a}(V_{\tilde{\mathcal{P}}}^{\tilde{\pi}}) - \mathbf{Var}_{(\mathsf{P}_{\tilde{V}})_s^a}(V_{\tilde{\mathcal{P}}}^{\tilde{\pi}})} \le \sqrt{\frac{8}{\max\{R, 1-\gamma\}}}. \tag{85}$$

Hence by combining (83) and (85) we have that

$$\begin{aligned}
&\sum_s d_{\mathsf{P}_{\tilde{V}}}^{\tilde{\pi}}(s) \sqrt{\frac{\mathbf{Var}_{\mathsf{P}_s^{\tilde{\pi}(s)}}(V_{\tilde{\mathcal{P}}}^{\tilde{\pi}})}{N(s, \tilde{\pi}(s))}} \\
&\le \sum_s d_{\mathsf{P}_{\tilde{V}}}^{\tilde{\pi}}(s) \sqrt{\frac{8 \log\frac{4SA}{\delta} \mathbf{Var}_{\mathsf{P}_s^{\tilde{\pi}(s)}}(V_{\tilde{\mathcal{P}}}^{\tilde{\pi}})}{N\mu_{\min}}} \\
&\le \sqrt{\frac{8 \log\frac{4SA}{\delta}}{N\mu_{\min}}} \left( (8 + \sqrt{8}) \sqrt{\frac{1}{\max\{R, 1-\gamma\}}} \right)
\end{aligned} \tag{86}$$

Thus plug the inequality above and the bound of $B_1$, we have that

$$\begin{aligned}
II &\le \frac{\gamma}{1-\gamma} \sum_s d_{\mathsf{P}_V}^{\tilde{\pi}}(s)(\tilde{\sigma}_s^{\tilde{\pi}}(V_{\tilde{\mathcal{P}}}^{\tilde{\pi}}) - (\mathsf{P}_{\tilde{V}})_s^{\tilde{\pi}(s)} V_{\tilde{\mathcal{P}}}^{\tilde{\pi}}) \\
&\le \sqrt{\frac{16c_1 \log\frac{4SAN}{\delta} \log\frac{4SA}{\delta}}{N\mu_{\min}(1-\gamma)^2}} \left( (8+\sqrt{8}) \sqrt{\frac{1}{\max\{R, 1-\gamma\}}} \right) \\
&\quad + \frac{4K_1 \log\frac{4SA}{\delta}}{N\mu_{\min}(1-\gamma)^2} + \frac{8 \log\frac{4SA}{\delta}}{(1-\gamma)^2 N\mu_{\min}}
\end{aligned} \tag{87}$$

Then we combine the bounds of terms $(I)$ and $(II)$ together, and we have that

$$\begin{aligned}
&\|V_{\tilde{\mathcal{P}}}^{\tilde{\pi}}(s) - V_{\mathcal{P}}^{\tilde{\pi}}(s)\| \\
&\le \sqrt{\frac{16c_1 \log\frac{4SAN}{\delta} \log\frac{4SA}{\delta}}{N\mu_{\min}(1-\gamma)^2}} \left( \|V_{\tilde{\mathcal{P}}}^{\tilde{\pi}} - V_{\mathcal{P}}^{\tilde{\pi}}\| + (16 + 2\sqrt{8}) \sqrt{\frac{1}{\max\{R, 1-\gamma\}}} \right)
\end{aligned}$$

$$+ \frac{8K_1 \log \frac{4SA}{\delta}}{N\mu_{\min}(1-\gamma)^2} + \frac{16 \log \frac{4SA}{\delta}}{(1-\gamma)^2 N\mu_{\min}} \tag{88}$$

Due to the fact that $N \geq \frac{196c_1 \log \frac{4SAN}{\delta} \log \frac{4SA}{\delta}}{(1-\gamma)^2 N\mu_{\min}}$,

$$\|V_{\tilde{\mathcal{P}}}^{\tilde{\pi}} - V_{\tilde{\mathcal{P}}}^{\tilde{\pi}}\| \leq (32 + 4\sqrt{8}) \sqrt{\frac{16c_1 \log \frac{4SAN}{\delta} \log \frac{4SA}{\delta}}{(1-\gamma)^2 N\mu_{\min}}} \sqrt{\frac{1}{\max\{R, 1-\gamma\}}}$$

$$+ \frac{2(8K_1 + 16) \log \frac{4SA}{\delta}}{N\mu_{\min}(1-\gamma)^2}, \tag{89}$$

and hence completes the proof. □

## C   Proofs of Section 4.2

**Theorem 7.** *With probability at least $1 - 4\delta$, the output policy $\tilde{\pi}$ of algorithm 1 satisfies*

$$V_{\mathcal{P}}^{\pi^*}(\rho) - V_{\mathcal{P}}^{\tilde{\pi}}(\rho) \leq 40 \sqrt{\frac{(1+R) \log \frac{24SAN}{\delta} \log \frac{4SA}{\delta} C^{\pi^*} S}{N(1-\gamma)^4}} + \frac{\gamma K_2 S \sqrt{1+R} \log \frac{8SA}{\delta}}{(1-\gamma)^2 N\mu_{\min}}. \tag{90}$$

*Proof.* Using the similar decomposition implies that

$$V_{\mathcal{P}}^{\pi^*}(s) - V_{\mathcal{P}}^{\tilde{\pi}}(s) = \underbrace{V_{\mathcal{P}}^{\pi^*}(s) - V_{\tilde{\mathcal{P}}}^{\tilde{\pi}}(s)}_{(A)} + \underbrace{V_{\tilde{\mathcal{P}}}^{\tilde{\pi}}(s) - V_{\mathcal{P}}^{\tilde{\pi}}(s)}_{(B)}. \tag{91}$$

The proof is then completed by combing the following two lemmas. □

T

**Lemma 4.** *(Bound on Term A) With probability at least $1 - 4\delta$, it holds that*

$$V_{\mathcal{P}}^{\pi^*}(s) - V_{\tilde{\mathcal{P}}}^{\tilde{\pi}}(s) \leq 40 \sqrt{\frac{(1+R)C^{\pi^*} S \log \frac{24SAN}{\delta} \log \frac{4SA}{\delta}}{N(1-\gamma)^4}} \tag{92}$$

*Proof.* To bound term $(A)$, note that

$$V_{\mathcal{P}}^{\pi^*}(s) - V_{\tilde{\mathcal{P}}}^{\tilde{\pi}}(s)$$

$$= r(s, \pi^*(s)) + \gamma \sigma_s^*(V_{\mathcal{P}}^{\pi^*}) - \max_a \{r(s,a) + \gamma \tilde{\sigma}_s^a(V_{\tilde{\mathcal{P}}}^{\tilde{\pi}})\}$$

$$\overset{(a)}{\leq} r(s, \pi^*(s)) + \gamma \sigma_s^*(V_{\mathcal{P}}^{\pi^*}) - r(s, \pi^*(s)) - \tilde{\sigma}_s^*(V_{\tilde{\mathcal{P}}}^{\tilde{\pi}})$$

$$= \gamma \sigma_s^*(V_{\mathcal{P}}^{\pi^*}) - \gamma \tilde{\sigma}_s^*(V_{\tilde{\mathcal{P}}}^{\tilde{\pi}})$$

$$= \gamma \sigma_s^*(V_{\mathcal{P}}^{\pi^*}) - \gamma \sigma_s^*(V_{\tilde{\mathcal{P}}}^{\tilde{\pi}}) + \gamma \sigma_s^*(V_{\tilde{\mathcal{P}}}^{\tilde{\pi}}) - \gamma \tilde{\sigma}_s^*(V_{\tilde{\mathcal{P}}}^{\tilde{\pi}})$$

$$\leq \gamma (\mathsf{P}_{\tilde{V}})_s^*(V_{\mathcal{P}}^{\pi^*} - V_{\tilde{\mathcal{P}}}^{\tilde{\pi}}) + \gamma \sigma_s^*(V_{\tilde{\mathcal{P}}}^{\tilde{\pi}}) - \gamma \tilde{\sigma}_s^*(V_{\tilde{\mathcal{P}}}^{\tilde{\pi}}), \tag{93}$$

where $(a)$ is from $\tilde{\pi} = \arg\max_\pi V_{\tilde{\mathcal{P}}}^\pi$, and the last inequality is from the fact $\mathsf{P}_{\tilde{V}} \in \mathcal{P}_s^*$ and hence $\sigma_s^*(V_{\mathcal{P}}^{\pi^*}) \leq (\mathsf{P}_{\tilde{V}})_s^* V_{\mathcal{P}}^{\pi^*}$.

Applying (93) recursively implies

$$V_{\mathcal{P}}^{\pi^*}(s) - V_{\tilde{\mathcal{P}}}^{\tilde{\pi}}(s) \leq \frac{1}{1-\gamma} \sum_x d_{\mathsf{P}_{\tilde{V}}}^*(x)|\sigma_x^*(V_{\tilde{\mathcal{P}}}^{\tilde{\pi}}) - \tilde{\sigma}_x^*(V_{\tilde{\mathcal{P}}}^{\tilde{\pi}})|. \tag{94}$$

Note that

$$\sum_x d_{\mathsf{P}_{\tilde{V}}}^*(x)|\sigma_x^*(V_{\tilde{\mathcal{P}}}^{\tilde{\pi}}) - \tilde{\sigma}_x^*(V_{\tilde{\mathcal{P}}}^{\tilde{\pi}})|$$

$$= \sum_x d^*_{\mathsf{P}_{\tilde{V}}}(x)|\sigma^*_x(V^{\tilde{\pi}}_{\tilde{\mathcal{P}}}) - \hat{\sigma}^*_x(V^{\tilde{\pi}}_{\tilde{\mathcal{P}}}) + \hat{\sigma}^*_x(V^{\tilde{\pi}}_{\tilde{\mathcal{P}}}) - \tilde{\sigma}^*_x(V^{\tilde{\pi}}_{\tilde{\mathcal{P}}})|$$

$$\leq \sum_x d^*_{\mathsf{P}_{\tilde{V}}}(x)|\sigma^*_x(V^{\tilde{\pi}}_{\tilde{\mathcal{P}}}) - \hat{\sigma}^*_x(V^{\tilde{\pi}}_{\tilde{\mathcal{P}}})| + \sum_x d^*_{\mathsf{P}_{\tilde{V}}}(x)|\hat{\sigma}^*_x(V^{\tilde{\pi}}_{\tilde{\mathcal{P}}}) - \tilde{\sigma}^*_x(V^{\tilde{\pi}}_{\tilde{\mathcal{P}}})|. \tag{95}$$

The first term in (95) can be bounded as follows. Recall the dual form of the support function w.r.t. the uncertainty set $\mathcal{P} = \{q \in \Delta(\mathcal{S}) : \chi^2(q||p) \leq R\}$ as follows:

$$\sigma_{\mathcal{P}}(V) = \max_{\alpha \in [0,V]} \left\{ pV_\alpha - \sqrt{R\mathbf{Var}_p(V_\alpha)} \right\}, \tag{96}$$

where $V_\alpha(s) = \min\{V(s), \alpha\}$.

Applying the dual form further implies that

$$|\sigma^a_s(V^{\tilde{\pi}}_{\tilde{\mathcal{P}}}) - \hat{\sigma}^a_s(V^{\tilde{\pi}}_{\tilde{\mathcal{P}}})|$$

$$\leq \max_{0 \leq \alpha \leq \frac{1}{1-\gamma}} \left| (\mathsf{P}^a_s - \hat{\mathsf{P}}^a_s)(V^{\tilde{\pi}}_{\tilde{\mathcal{P}}})_\alpha - \left( \sqrt{R\mathbf{Var}_{\mathsf{P}^a_s}((V^{\tilde{\pi}}_{\tilde{\mathcal{P}}})_\alpha)} - \sqrt{R\mathbf{Var}_{\hat{\mathsf{P}}^a_s}((V^{\tilde{\pi}}_{\tilde{\mathcal{P}}})_\alpha)} \right) \right|$$

$$\leq \max_{0 \leq \alpha \leq \frac{1}{1-\gamma}} \left| (\mathsf{P}^a_s - \hat{\mathsf{P}}^a_s)(V^{\tilde{\pi}}_{\tilde{\mathcal{P}}})_\alpha \right| + \max_{0 \leq \alpha \leq \frac{1}{1-\gamma}} \left| \left( \sqrt{R\mathbf{Var}_{\mathsf{P}^a_s}((V^{\tilde{\pi}}_{\tilde{\mathcal{P}}})_\alpha)} - \sqrt{R\mathbf{Var}_{\hat{\mathsf{P}}^a_s}((V^{\tilde{\pi}}_{\tilde{\mathcal{P}}})_\alpha)} \right) \right|$$

$$\leq 2\sqrt{\frac{\log \frac{2SAN}{\delta}}{N(s,a)(1-\gamma)^2}} + \max_{0 \leq \alpha \leq \frac{1}{1-\gamma}} \left| \left( \sqrt{R\mathbf{Var}_{\mathsf{P}^a_s}((V^{\tilde{\pi}}_{\tilde{\mathcal{P}}})_\alpha)} - \sqrt{R\mathbf{Var}_{\hat{\mathsf{P}}^a_s}((V^{\tilde{\pi}}_{\tilde{\mathcal{P}}})_\alpha)} \right) \right| + \frac{R}{(1-\gamma)N}, \tag{97}$$

where the last inequality directly follows from the Hoeffding's inequality [38] and $\frac{1}{N}$-net technique used in Lemma 15.

As for the second term in (97), we utilize Lemma 7 and the technique of $\epsilon$-net, which implies that

$$|\sigma^a_s((V^{\tilde{\pi}}_{\tilde{\mathcal{P}}})_\alpha) - \hat{\sigma}^a_s(V^{\tilde{\pi}}_{\tilde{\mathcal{P}}})|$$

$$\leq 2\sqrt{\frac{\log \frac{2SAN}{\delta}}{N(s,a)(1-\gamma)^2}} + 2\sqrt{\frac{2R\log \frac{24SAN}{\delta}}{N(s,a)(1-\gamma)^2}}$$

$$\leq 4\sqrt{\frac{2(1+R)\log \frac{24SAN}{\delta}}{N(s,a)(1-\gamma)^2}} + \frac{R}{(1-\gamma)N}. \tag{98}$$

This hence bounds the first term in (95). For the second term in (95), similarly apply the dual form and we have that

$$|\hat{\sigma}^a_s(V^{\tilde{\pi}}_{\tilde{\mathcal{P}}}) - \tilde{\sigma}^a_s(V^{\tilde{\pi}}_{\tilde{\mathcal{P}}})|$$

$$\leq \max_{0 \leq \alpha \leq \frac{1}{1-\gamma}} \left| \left( \sqrt{R\mathbf{Var}_{\hat{\mathsf{P}}^a_s}((V^{\tilde{\pi}}_{\tilde{\mathcal{P}}})_\alpha)} - \sqrt{(R+\kappa^a_s)\mathbf{Var}_{\hat{\mathsf{P}}^a_s}((V^{\tilde{\pi}}_{\tilde{\mathcal{P}}})_\alpha)} \right) \right|$$

$$\leq \frac{\kappa^a_s}{\sqrt{R} + \sqrt{R+\kappa^a_s}} \max_{0 \leq \alpha \leq \frac{1}{1-\gamma}} \sqrt{\mathbf{Var}_{\hat{\mathsf{P}}^a_s}((V^{\tilde{\pi}}_{\tilde{\mathcal{P}}})_\alpha)}$$

$$\leq \sqrt{\frac{(1+R)\log \frac{24SAN}{\delta}}{N(s,a)(1-\gamma)^2}}, \tag{99}$$

where the last inequality is from the fact that $\mathbf{Var}(V_\alpha) \leq \frac{1}{(1-\gamma)^2}$ for any $V \leq \frac{1}{1-\gamma}$ and any $\alpha \leq \frac{1}{1-\gamma}$.

We now plug (98) and (99) to (95) and (94), and we have that

$$V^{\pi^*}_{\mathcal{P}}(s) - V^{\tilde{\pi}}_{\tilde{\mathcal{P}}}(s) \leq \frac{1}{1-\gamma} \sum_x d^*_{\mathsf{P}_{\tilde{V}}}(x)|\sigma^*_x(V^{\tilde{\pi}}_{\tilde{\mathcal{P}}}) - \tilde{\sigma}^*_x(V^{\tilde{\pi}}_{\tilde{\mathcal{P}}})|$$

$$\leq \frac{1}{1-\gamma} \sum_x d^*_{\mathsf{P}_{\tilde{V}}}(x) \left( \sqrt{\frac{(1+R)\log \frac{24SAN}{\delta}}{N(x,\pi^*(x))(1-\gamma)^2}} + 4\sqrt{\frac{2(1+R)\log \frac{24SAN}{\delta}}{N(x,\pi^*(x))(1-\gamma)^2}} + \frac{2R}{(1-\gamma)N} \right)$$

$$\leq 40\sqrt{\frac{(1+R)C^{\pi^*}S\log\frac{24SAN}{\delta}\log\frac{4SA}{\delta}}{N(1-\gamma)^4}} \tag{100}$$

where the last inequality is from (44). This hence completes the proof. $\square$

**Lemma 5.** *(Bound on Term B) With probability at least $1 - 4\delta$, it holds that*

$$V_{\tilde{\mathcal{P}}}^{\tilde{\pi}}(s) - V_{\mathcal{P}}^{\tilde{\pi}}(s) \leq \frac{\gamma K_2 S\sqrt{1+R}\log\frac{8SA}{\delta}}{(1-\gamma)^2 N\mu_{\min}}. \tag{101}$$

*Proof.* Similarly to the proof of Lemma 4, note that

$$\begin{aligned}
&V_{\tilde{\mathcal{P}}}^{\tilde{\pi}}(s) - V_{\mathcal{P}}^{\tilde{\pi}}(s) \\
&= r(s,\tilde{\pi}(s)) + \gamma\tilde{\sigma}_s^{\tilde{\pi}(s)}(V_{\tilde{\mathcal{P}}}^{\tilde{\pi}}) - r(s,\tilde{\pi}(s)) - \gamma\sigma_s^{\tilde{\pi}(s)}(V_{\mathcal{P}}^{\tilde{\pi}}) \\
&= \gamma\tilde{\sigma}_s^{\tilde{\pi}(s)}(V_{\tilde{\mathcal{P}}}^{\tilde{\pi}}) - \gamma\sigma_s^{\tilde{\pi}(s)}(V_{\mathcal{P}}^{\tilde{\pi}}) \\
&= \gamma\tilde{\sigma}_s^{\tilde{\pi}(s)}(V_{\tilde{\mathcal{P}}}^{\tilde{\pi}}) - \gamma\tilde{\sigma}_s^{\tilde{\pi}(s)}(V_{\mathcal{P}}^{\tilde{\pi}}) + \gamma\tilde{\sigma}_s^{\tilde{\pi}(s)}(V_{\mathcal{P}}^{\tilde{\pi}}) - \gamma\sigma_s^{\tilde{\pi}(s)}(V_{\mathcal{P}}^{\tilde{\pi}}) \\
&\leq \gamma(\tilde{\mathsf{P}}_V)_s^{\tilde{\pi}(s)}(V_{\tilde{\mathcal{P}}}^{\tilde{\pi}} - V_{\mathcal{P}}^{\tilde{\pi}}) + \gamma\tilde{\sigma}_s^{\tilde{\pi}(s)}(V_{\mathcal{P}}^{\tilde{\pi}}) - \gamma\sigma_s^{\tilde{\pi}(s)}(V_{\mathcal{P}}^{\tilde{\pi}}) \\
&\leq \frac{\gamma}{1-\gamma}\sum_s d_V^{\tilde{\pi}}(s)\left(\tilde{\sigma}_s^{\tilde{\pi}(s)}(V_{\mathcal{P}}^{\tilde{\pi}}) - \sigma_s^{\tilde{\pi}(s)}(V_{\mathcal{P}}^{\tilde{\pi}})\right)
\end{aligned} \tag{102}$$

where $(\tilde{\mathsf{P}}_V)_s^{\tilde{\pi}(s)} = \arg\min_{q\in\tilde{\mathcal{P}}_s^{\tilde{\pi}(s)}} qV_{\mathcal{P}}^{\tilde{\pi}}$. We then bound the term $\tilde{\sigma}_s^{\tilde{\pi}(s)}(V_{\mathcal{P}}^{\tilde{\pi}}) - \sigma_s^{\tilde{\pi}(s)}(V_{\mathcal{P}}^{\tilde{\pi}})$.

From Lemma 6, it holds that

$$\begin{aligned}
&\tilde{\sigma}_s^{\tilde{\pi}(s)}(V_{\mathcal{P}}^{\tilde{\pi}}) - \sigma_s^{\tilde{\pi}(s)}(V_{\mathcal{P}}^{\tilde{\pi}}) \\
&\leq \frac{\gamma}{1-\gamma}\sum_s d_V^{\tilde{\pi}}(s)\frac{K_2 S\sqrt{1+R}}{(1-\gamma)N(s,a)} \\
&= \frac{\gamma K_2 S\sqrt{1+R}\log\frac{8SA}{\delta}}{(1-\gamma)^2 N\mu_{\min}},
\end{aligned} \tag{103}$$

which completes the proof. $\square$

**Lemma 6.** *With probability at least $1 - \delta$, it holds that for any $s \in \mathcal{S}$,*

$$-\sigma_s^{\tilde{\pi}(s)}(V_{\mathcal{P}}^{\tilde{\pi}}) + \tilde{\sigma}_s^{\tilde{\pi}(s)}(V_{\mathcal{P}}^{\tilde{\pi}}) \leq \frac{K_2 S\sqrt{1+R}}{(1-\gamma)N(s,\tilde{\pi}(s))} \tag{104}$$

*Proof.* We first consider a vector $V$ that is independent from $\hat{\mathsf{P}}$.

$$\begin{aligned}
&\tilde{\sigma}_s^{\tilde{\pi}(s)}(V) - \sigma_s^{\tilde{\pi}(s)}(V) \\
&= \max_\alpha\{\hat{\mathsf{P}}_s^{\tilde{\pi}(s)}(V_\alpha) - \sqrt{(R+\kappa_s^{\tilde{\pi}(s)})\mathbf{Var}_{\hat{\mathsf{P}}_s^{\tilde{\pi}(s)}}(V_\alpha)}\} - \max_\alpha\{\mathsf{P}_s^{\tilde{\pi}(s)}(V_\alpha) - \sqrt{R\mathbf{Var}_{\mathsf{P}_s^{\tilde{\pi}(s)}}(V_\alpha)}\} \\
&\leq \hat{\mathsf{P}}_s^{\tilde{\pi}(s)}(V_{\alpha^*}) - \sqrt{(R+\kappa_s^{\tilde{\pi}(s)})\mathbf{Var}_{\hat{\mathsf{P}}_s^{\tilde{\pi}(s)}}(V_{\alpha^*})} - \mathsf{P}_s^{\tilde{\pi}(s)}(V_{\alpha^*}) + \sqrt{R\mathbf{Var}_{\mathsf{P}_s^{\tilde{\pi}(s)}}(V_{\alpha^*})},
\end{aligned} \tag{105}$$

where $\alpha^* \triangleq \arg\max_\alpha\{\hat{\mathsf{P}}_s^{\tilde{\pi}(s)}(V_\alpha) - \sqrt{(R+\kappa_s^{\tilde{\pi}(s)})\mathbf{Var}_{\hat{\mathsf{P}}_s^{\tilde{\pi}(s)}}(V_\alpha)}\}$, and the last inequality is from the fact that $-\max f \leq -f(x)$ for any $x$.

We then construct an $\epsilon_1$-net over $[0, \frac{1}{1-\gamma}]$, such that there exists $\beta$ with $\|V_{\alpha^*} - V_\beta\| \leq \epsilon_1$. Then we further have that

$$\begin{aligned}
&\tilde{\sigma}_s^{\tilde{\pi}(s)}(V) - \sigma_s^{\tilde{\pi}(s)}(V) \\
&\leq \hat{\mathsf{P}}_s^{\tilde{\pi}(s)}(V_{\alpha^*}) - \sqrt{(R+\kappa_s^{\tilde{\pi}(s)})\mathbf{Var}_{\hat{\mathsf{P}}_s^{\tilde{\pi}(s)}}(V_{\alpha^*})} - \mathsf{P}_s^{\tilde{\pi}(s)}(V_{\alpha^*}) + \sqrt{R\mathbf{Var}_{\mathsf{P}_s^{\tilde{\pi}(s)}}(V_{\alpha^*})} \\
&\leq \frac{c_1}{(1-\gamma)N(s,\tilde{\pi}(s))} + \sqrt{\frac{c_2\mathbf{Var}_{\hat{\mathsf{P}}_s^{\tilde{\pi}(s)}}(V_\beta)}{N(s,\tilde{\pi}(s))}} + \frac{\epsilon_1}{(1-\gamma)}
\end{aligned}$$

$$- \sqrt{(R + \kappa_s^{\tilde{\pi}(s)})\mathbf{Var}_{\hat{\mathsf{P}}_s^{\tilde{\pi}(s)}}(V_{\alpha^*})} + \sqrt{R\mathbf{Var}_{\mathsf{P}_s^{\tilde{\pi}(s)}}(V_{\alpha^*})}, \tag{106}$$

where we use the Bernstein inequality and the technique of $\epsilon_1$-net. To bound the last two terms, we note that

$$- \sqrt{(R + \kappa_s^{\tilde{\pi}(s)})\mathbf{Var}_{\hat{\mathsf{P}}_s^{\tilde{\pi}(s)}}(V_{\alpha^*})} + \sqrt{R\mathbf{Var}_{\mathsf{P}_s^{\tilde{\pi}(s)}}(V_{\alpha^*})}$$

$$\stackrel{(a)}{=} - \sqrt{(R + \kappa_s^{\tilde{\pi}(s)})\mathbf{Var}_{\hat{\mathsf{P}}_s^{\tilde{\pi}(s)}}(V_\beta)} + \sqrt{R\mathbf{Var}_{\hat{\mathsf{P}}_s^{\tilde{\pi}(s)}}(V_\beta)} - \sqrt{R\mathbf{Var}_{\hat{\mathsf{P}}_s^{\tilde{\pi}(s)}}(V_\beta)} + \sqrt{R\mathbf{Var}_{\mathsf{P}_s^{\tilde{\pi}(s)}}(V_\beta)} + \sqrt{\frac{\epsilon_1}{(1-\gamma)}}$$

$$\stackrel{(b)}{=} - \sqrt{(R + \kappa_s^{\tilde{\pi}(s)})\mathbf{Var}_{\hat{\mathsf{P}}_s^{\tilde{\pi}(s)}}(V_\beta)} + \sqrt{R\mathbf{Var}_{\hat{\mathsf{P}}_s^{\tilde{\pi}(s)}}(V_\beta)} + \sqrt{\frac{\epsilon_1}{(1-\gamma)}}$$

$$+ \sqrt{R}\left(\sqrt{\frac{C_1\mathbf{Var}_{\hat{\mathsf{P}}_s^{\tilde{\pi}(s)}}(V_\beta)}{N(s,\tilde{\pi}(s))}} + \frac{C_2}{(1-\gamma)N(s,\tilde{\pi}(s))}\right), \tag{107}$$

where we again use the $\epsilon_1$-net technique in $(a)$, and apply Lemma 7 in $(b)$. Combining (106) and (107) implies that

$$\tilde{\sigma}_s^{\tilde{\pi}(s)}(V) - \sigma_s^{\tilde{\pi}(s)}(V)$$

$$\leq \frac{c_1}{(1-\gamma)N(s,\tilde{\pi}(s))} + \sqrt{\frac{c_2\mathbf{Var}_{\hat{\mathsf{P}}_s^{\tilde{\pi}(s)}}(V_\beta)}{N(s,\tilde{\pi}(s))}} + \frac{\epsilon_1}{(1-\gamma)}$$

$$- \sqrt{(R + \kappa_s^{\tilde{\pi}(s)})\mathbf{Var}_{\hat{\mathsf{P}}_s^{\tilde{\pi}(s)}}(V_\beta)} + \sqrt{R\mathbf{Var}_{\hat{\mathsf{P}}_s^{\tilde{\pi}(s)}}(V_\beta)} + \sqrt{\frac{\epsilon_1}{(1-\gamma)}}$$

$$+ \sqrt{R}\left(\sqrt{\frac{C_1\mathbf{Var}_{\hat{\mathsf{P}}_s^{\tilde{\pi}(s)}}(V_\beta)}{N(s,\tilde{\pi}(s))}} + \frac{C_2}{(1-\gamma)N(s,\tilde{\pi}(s))}\right)$$

$$= \frac{c_1}{(1-\gamma)N(s,\tilde{\pi}(s))} + \frac{\epsilon_1}{(1-\gamma)} + \sqrt{\frac{\epsilon_1}{(1-\gamma)}} + \frac{C_2\sqrt{R}}{(1-\gamma)N(s,\tilde{\pi}(s))}$$

$$- \sqrt{(R + \kappa_s^{\tilde{\pi}(s)})\mathbf{Var}_{\hat{\mathsf{P}}_s^{\tilde{\pi}(s)}}(V_\beta)} + \sqrt{R\mathbf{Var}_{\hat{\mathsf{P}}_s^{\tilde{\pi}(s)}}(V_\beta)} + \sqrt{\frac{RC_1\mathbf{Var}_{\hat{\mathsf{P}}_s^{\tilde{\pi}(s)}}(V_\beta)}{N(s,\tilde{\pi}(s))}} + \sqrt{\frac{c_2\mathbf{Var}_{\hat{\mathsf{P}}_s^{\tilde{\pi}(s)}}(V_\beta)}{N(s,\tilde{\pi}(s))}}$$

$$= \frac{c_1}{(1-\gamma)N(s,\tilde{\pi}(s))} + \frac{\epsilon_1}{(1-\gamma)} + \sqrt{\frac{\epsilon_1}{(1-\gamma)}} + \frac{C_2\sqrt{R}}{(1-\gamma)N(s,\tilde{\pi}(s))}$$

$$+ \sqrt{\mathbf{Var}_{\hat{\mathsf{P}}_s^{\tilde{\pi}(s)}}(V_\beta)}\left(\sqrt{\frac{c_2}{N(s,\tilde{\pi}(s))}} + \sqrt{\frac{RC_1}{N(s,\tilde{\pi}(s))}} + \sqrt{R} - \sqrt{R + \kappa_s^{\tilde{\pi}(s)}}\right). \tag{108}$$

Clearly, if we set $C = \sqrt{c_2} + \sqrt{RC_1}$ and

$$\kappa_s^{\tilde{\pi}(s)} = \frac{C}{N(s,\tilde{\pi}(s))} + 2\sqrt{\frac{CR}{N(s,\tilde{\pi}(s))}} = \tilde{\mathcal{O}}\left(\sqrt{\frac{1}{N(s,\tilde{\pi}(s))}}\right), \tag{109}$$

then $\left(\sqrt{\frac{c_2}{N(s,\tilde{\pi}(s))}} + \sqrt{\frac{RC_1}{N(s,\tilde{\pi}(s))}} + \sqrt{R} - \sqrt{R + \kappa_s^{\tilde{\pi}(s)}}\right) \leq 0$, and hence

$$\tilde{\sigma}_s^{\tilde{\pi}(s)}(V) - \sigma_s^{\tilde{\pi}(s)}(V) \leq \frac{c_1}{(1-\gamma)N(s,\tilde{\pi}(s))} + \frac{\epsilon_1}{(1-\gamma)} + \sqrt{\frac{\epsilon_1}{(1-\gamma)}} + \frac{C_2\sqrt{R}}{(1-\gamma)N(s,\tilde{\pi}(s))}$$

$$\leq \tilde{\mathcal{O}}\left(\frac{\sqrt{1+R}}{(1-\gamma)N(s,\tilde{\pi}(s))}\right), \tag{110}$$

by setting $\epsilon_1 \leq \tilde{\mathcal{O}}(\frac{1}{N})$.

Now to show the claim for $V_{\tilde{\mathcal{P}}}^{\tilde{\pi}}$, we construct an $\epsilon_2$-net over $[0, \frac{1}{1-\gamma}]^S$. By applying the similar trick, we complete the proof. $\square$

**Lemma 7.** *For any vector $V \in [0, \frac{1}{1-\gamma}]$ that is independent with $\mathsf{P}$, with probability at least $1 - \delta$, it holds that*

$$\left| \sqrt{\mathbf{Var}_{\mathsf{P}_s^a}(V)} - \sqrt{\mathbf{Var}_{\hat{\mathsf{P}}_s^a}(V)} \right| \leq \sqrt{\frac{C_1 \mathbf{Var}_{\hat{\mathsf{P}}_s^a}(V)}{N(s,a)}} + \frac{C_2}{(1-\gamma)N(s,a)}. \tag{111}$$

*Proof.* This result can be derived from the Bernstein inequality for $U$-statistics [3, 32], by noting that the sample standard deviation $\sqrt{\frac{n}{n-1} \mathbf{Var}_{\hat{\mathsf{P}}_s^a}(V)}$ is an $U$-statistics and

$$\left| \sqrt{\mathbf{Var}_{\hat{\mathsf{P}}_s^a}(V)} - \sqrt{\frac{n}{n-1} \mathbf{Var}_{\hat{\mathsf{P}}_s^a}(V)} \right| \leq \sqrt{\frac{\mathbf{Var}_{\hat{\mathsf{P}}_s^a}(V)}{n-1}}. \tag{112}$$

$\square$

## D   Proofs of Section 4.3

**Theorem 8.** *If $N \geq \frac{8 \log \frac{1}{\delta}}{\mu_{\min}}$, then there exists some universal constants $C_1, C_2$, such that with probability at least $1 - 4\delta$, it holds that*

$$V_{\mathcal{P}}^{\pi^*}(\rho) - V_{\mathcal{P}}^{\tilde{\pi}}(\rho) \leq \frac{8}{NR(1-\gamma)^2} + \frac{2C_2\sqrt{C^{\pi^*}} \log \frac{2(1+R)N^3 S}{(1-\gamma)\delta}}{R(1-\gamma)^2} \sqrt{\frac{S}{N\mathsf{P}_{\min}}}. \tag{113}$$

*Proof.* Note that

$$V_{\mathcal{P}}^{\pi^*} - V_{\mathcal{P}}^{\tilde{\pi}} = \underbrace{V_{\mathcal{P}}^{\pi^*} - V_{\tilde{\mathcal{P}}}^{\tilde{\pi}}}_{\Delta_1} + \underbrace{V_{\tilde{\mathcal{P}}}^{\tilde{\pi}} - V_{\mathcal{P}}^{\tilde{\pi}}}_{\Delta_2}. \tag{114}$$

The proof completes by combining the bounds on the two terms, shown in the following two lemmas. $\square$

**Lemma 8.** *There exists a constant $C_2$, such that with probability at least $1 - \delta$, it holds that*

$$\rho^\top \Delta_1 \leq \frac{4}{NR(1-\gamma)^2} + \frac{2C_2\sqrt{C^{\pi^*}} \log \frac{2(1+R)N^3 S}{(1-\gamma)\delta}}{R(1-\gamma)^2} \sqrt{\frac{S}{N\mathsf{P}_{\min}}}. \tag{115}$$

*Proof.* From the definition,

$$V_{\tilde{\mathcal{P}}}^{\tilde{\pi}}(s) = \max_a Q_{\tilde{\mathcal{P}}}^{\tilde{\pi}}(s,a) \geq Q_{\tilde{\mathcal{P}}}^{\tilde{\pi}}(s, \pi^*(s)) = \hat{r}(s, \pi^*(s)) + \gamma \sigma_{\tilde{\mathcal{P}}_s^{\pi^*}}(V_{\tilde{\mathcal{P}}}^{\tilde{\pi}}), \tag{116}$$

hence

$$\begin{aligned}
\Delta_1(s) &= V_{\mathcal{P}}^{\pi^*}(s) - V_{\tilde{\mathcal{P}}}^{\tilde{\pi}}(s) \\
&= \hat{r}(s, \pi^*(s)) + \gamma \sigma_{\mathcal{P}_s^{\pi^*}}(V_{\mathcal{P}}^*) - V_{\tilde{\mathcal{P}}}^{\tilde{\pi}}(s) \\
&\leq \gamma \sigma_{\mathcal{P}_s^{\pi^*}}(V_{\mathcal{P}}^{\pi^*}) - \gamma \sigma_{\tilde{\mathcal{P}}_s^{\pi^*}}(V_{\tilde{\mathcal{P}}}^{\tilde{\pi}}) \\
&= \gamma \sigma_{\mathcal{P}_s^{\pi^*}}(V_{\mathcal{P}}^*) - \gamma \sigma_{\mathcal{P}_s^{\pi^*}}(V_{\tilde{\mathcal{P}}}^{\tilde{\pi}}) + \gamma \sigma_{\mathcal{P}_s^{\pi^*}}(V_{\tilde{\mathcal{P}}}^{\tilde{\pi}}) - \gamma \sigma_{\tilde{\mathcal{P}}_s^{\pi^*}}(V_{\tilde{\mathcal{P}}}^{\tilde{\pi}}) \\
&\overset{(a)}{\leq} \gamma q_s^{\pi^*}(V_{\mathcal{P}}^{\pi^*} - V_{\tilde{\mathcal{P}}}^{\tilde{\pi}}) + c(s), \tag{117}
\end{aligned}$$

where $q_s^a$ is the worst-case transition kernel of $V_{\tilde{\mathcal{P}}}^{\tilde{\pi}}$ in $\mathcal{P}_s^a$, and $(a)$ is from $\sigma_{\mathcal{P}_s^{\pi^*}}(V_{\mathcal{P}}^{\pi^*}) - \sigma_{\mathcal{P}_s^{\pi^*}}(V_{\tilde{\mathcal{P}}}^{\tilde{\pi}}) = \sigma_{\mathcal{P}_s^{\pi^*}}(V_{\mathcal{P}}^{\pi^*}) - (q_s^{\pi^*})(V_{\tilde{\mathcal{P}}}^{\tilde{\pi}}) \leq (q_s^{\pi^*})(V_{\mathcal{P}}^{\pi^*} - V_{\tilde{\mathcal{P}}}^{\tilde{\pi}})$, and $c(s) \triangleq \gamma \sigma_{\mathcal{P}_s^{\pi^*}}(V_{\tilde{\mathcal{P}}}^{\tilde{\pi}}) - \gamma \sigma_{\tilde{\mathcal{P}}_s^{\pi^*}}(V_{\tilde{\mathcal{P}}}^{\tilde{\pi}})$.

Recursively applying (117) further implies that

$$\rho^\top \Delta_1 \leq \frac{1}{1-\gamma} \langle d_q^{\pi^*}, c \rangle. \tag{118}$$

We moreover rewrite $c$ as

$$c(s) = \underbrace{\gamma\sigma_{\mathcal{P}_s^{\pi^*}}(V_{\tilde{\mathcal{P}}}^{\tilde{\pi}}) - \gamma\sigma_{\hat{\mathcal{P}}_s^{\pi^*}}(V_{\tilde{\mathcal{P}}}^{\tilde{\pi}})}_{\Delta_{1,1}} + \underbrace{\gamma\sigma_{\hat{\mathcal{P}}_s^{\pi^*}}(V_{\tilde{\mathcal{P}}}^{\tilde{\pi}}) - \gamma\sigma_{\tilde{\mathcal{P}}_s^{\pi^*}}(V_{\tilde{\mathcal{P}}}^{\tilde{\pi}})}_{\Delta_{1,2}}. \tag{119}$$

Moreover, we introduce two sets

$$\mathcal{S}_1 = \{s : \max_{\mathsf{Q}\in\mathcal{P}} d_{\mathsf{Q}}^{\pi^*}(s, \pi^*(s)) = 0\}, \tag{120}$$

$$\mathcal{S}_2 = (\mathcal{S}_1)^c. \tag{121}$$

For $s \in \mathcal{S}_1$, (216) of [36] implies that $d_q^{\pi^*}(s) = 0$.

We then focus on $s \in \mathcal{S}_2$. It has been shown in [36] that $\mu(s, \pi^*(s)) > 0$ and $N(s, \pi^*(s)) \geq \frac{N\min\{\frac{1}{S}, d_q^{\pi^*}(s)\}}{12C^{\pi^*}}$.

Thus Lemma 17 first implies that

$$|\sigma_{\mathcal{P}_s^{\pi^*}}(V_{\tilde{\mathcal{P}}}^{\tilde{\pi}}) - \sigma_{\hat{\mathcal{P}}_s^{\pi^*}}(V_{\tilde{\mathcal{P}}}^{\tilde{\pi}})| \leq \frac{4}{NR(1-\gamma)} + \frac{C_1}{R(1-\gamma)}\sqrt{\frac{\log\frac{2(1+R)N^3S}{(1-\gamma)\delta}}{N(s, \pi^*(s))\min_x \hat{\mathsf{P}}_{s,x}^{\pi^*}}}$$

$$\leq \frac{4}{NR(1-\gamma)} + \frac{C_1}{R(1-\gamma)}\sqrt{\frac{12C^{\pi^*}\log\frac{2(1+R)N^3S}{(1-\gamma)\delta}}{\min_x \hat{\mathsf{P}}_{s,x}^{\pi^*}N\min\{\frac{1}{S}, d_q^{\pi^*}(s)\}}}; \tag{122}$$

To bound $\Delta_{1,2}$, note that

$$\Delta_{1,2}(s) = \gamma\sigma_{\hat{\mathcal{P}}_s^{\pi^*}}(V_{\tilde{\mathcal{P}}}^{\tilde{\pi}})(s) - \gamma\sigma_{\tilde{\mathcal{P}}_s^{\pi^*}}(V_{\tilde{\mathcal{P}}}^{\tilde{\pi}})(s)$$

$$\overset{(a)}{=} \max_{0\leq\lambda}\left\{-\lambda\log\left(\hat{\mathsf{P}}_s^{\pi^*}\mathbf{exp}\left(\frac{-V_{\tilde{\mathcal{P}}}^{\tilde{\pi}}}{\lambda}\right)\right) - \lambda R\right\}$$

$$- \max_{0\leq\lambda}\left\{-\lambda\log\left(\hat{\mathsf{P}}_s^{\pi^*}\mathbf{exp}\left(\frac{-V_{\tilde{\mathcal{P}}}^{\tilde{\pi}}}{\lambda}\right)\right) - \lambda(R + \kappa_s^{\pi^*(s)})\right\}$$

$$\overset{(b)}{=} \max_{0\leq\lambda\leq\frac{1}{(1-\gamma)(R+\kappa_s^{\pi^*(s)})}}\left\{-\lambda\log\left(\hat{\mathsf{P}}_s^{\pi^*}\mathbf{exp}\left(\frac{-V_{\tilde{\mathcal{P}}}^{\tilde{\pi}}}{\lambda}\right)\right) - \lambda R\right\}$$

$$- \max_{0\leq\lambda\leq\frac{1}{(1-\gamma)(R+\kappa_s^{\pi^*(s)})}}\left\{-\lambda\log\left(\hat{\mathsf{P}}_s^{\pi^*}\mathbf{exp}\left(\frac{-V_{\tilde{\mathcal{P}}}^{\tilde{\pi}}}{\lambda}\right)\right) - \lambda(R + \kappa_s^{\pi^*(s)})\right\}$$

$$\overset{(c)}{\leq} \max_{0\leq\lambda\leq\frac{1}{(1-\gamma)(R+\kappa_s^{\pi^*(s)})}}\{\lambda\kappa_s^{\pi^*(s)}\}$$

$$\leq \frac{\kappa_s^{\pi^*(s)}}{(1-\gamma)(R + \kappa_s^{\pi^*(s)})}$$

$$\leq \frac{\kappa_s^{\pi^*(s)}}{(1-\gamma)R}. \tag{123}$$

where $(a)$ is from the dual solution of KL-divergence, and $(b)$ is due to the fact that the optimal solutions to both dual forms satisfy $\lambda^* \leq \frac{1}{(1-\gamma)(R+\kappa_s^{\pi^*(s)})}$ [65], and $(c)$ is due to $\max f - \max g \leq \max|f - g|$.

Then we plug in the definition of $\kappa_s^{\pi^*(s)} = C_1\sqrt{\frac{\log\frac{2(1+R)N^3S}{(1-\gamma)\delta}}{N(s,\pi^*(s))\min_x \hat{\mathsf{P}}_{s,x}^{\pi^*}}}$ and using (202) of [36], we have that

$$\Delta_{1,2}(s) \leq \frac{C_1}{(1-\gamma)R}\sqrt{\frac{8C^{\pi^*}\log^2\frac{NS}{\delta}\log\frac{2(1+R)N^3S}{(1-\gamma)\delta}}{N\min\{\frac{1}{S}, d_q^{\pi^*}(s)\}\mathsf{P}_{\min}}}. \tag{124}$$

Combine (122) and (124), we have that

$$c(s) \le \frac{C_1}{(1-\gamma)R}\sqrt{\frac{8C^{\pi^*}\log\frac{NS}{\delta}\log\frac{2(1+R)N^3S}{(1-\gamma)\delta}}{N\min\{\frac{1}{S},d_q^{\pi^*}(s)\}\mathsf{P}_{\min}}} + \frac{C_1}{R(1-\gamma)}\sqrt{\frac{12C^{\pi^*}\log\frac{2(1+R)N^3S}{(1-\gamma)\delta}}{\min_x\hat{\mathsf{P}}_{s,x}^{\pi^*}N\min\{\frac{1}{S},d_q^{\pi^*}(s)\}}}$$
$$+ \frac{4}{NR(1-\gamma)}. \tag{125}$$

Thus we have that

$$\langle d_q^{\pi^*}, c\rangle = \sum_s d_q^{\pi^*}(s)c(s)$$

$$\le \sum_s d_q^{\pi^*}(s)\left(\frac{C_2}{(1-\gamma)R}\sqrt{\frac{C^{\pi^*}\log\frac{NS}{\delta}\log\frac{2(1+R)N^3S}{(1-\gamma)\delta}}{N\mathsf{P}_{\min}\min\{\frac{1}{S},d_q^{\pi^*}(s)\}}} + \frac{4}{NR(1-\gamma)}\right)$$

$$\overset{(a)}{\le} \frac{4}{NR(1-\gamma)} + \frac{2C_2\sqrt{C^{\pi^*}\log^2\frac{2(1+R)N^3S}{(1-\gamma)\delta}}}{R(1-\gamma)}\sqrt{\sum_s \frac{d_q^{\pi^*}(s)}{\min\{\frac{1}{S},d_q^{\pi^*}(s)\}}}\sqrt{\frac{1}{N\mathsf{P}_{\min}}}$$

$$\overset{(b)}{\le} \frac{4}{NR(1-\gamma)} + \frac{2C_2\sqrt{C^{\pi^*}\log^2\frac{2(1+R)N^3S}{(1-\gamma)\delta}}}{R(1-\gamma)}\sqrt{\frac{S}{N\mathsf{P}_{\min}}}, \tag{126}$$

where $(a)$ is from Cauchy inequality, and $(b)$ is from (220) of [36], which hence completes the proof. $\qquad\square$

We then bound the term $\Delta_2$.

**Lemma 9.**

$$\rho^\top\Delta_2 \le \frac{4}{NR(1-\gamma)^2}. \tag{127}$$

*Proof.* First, note that

$$\Delta_2(s) = V_{\tilde{\mathcal{P}}}^{\tilde{\pi}}(s) - V_{\mathcal{P}}^{\tilde{\pi}}(s)$$
$$= \gamma(\sigma_{\tilde{\mathcal{P}}_s^{\tilde{\pi}}}(V_{\tilde{\mathcal{P}}}^{\tilde{\pi}}) - \sigma_{\mathcal{P}_s^{\tilde{\pi}}}(V_{\mathcal{P}}^{\tilde{\pi}}))$$
$$= \gamma(\sigma_{\tilde{\mathcal{P}}_s^{\tilde{\pi}}}(V_{\tilde{\mathcal{P}}}^{\tilde{\pi}}) - \sigma_{\mathcal{P}_s^{\tilde{\pi}}}(V_{\tilde{\mathcal{P}}}^{\tilde{\pi}}) + \sigma_{\mathcal{P}_s^{\tilde{\pi}}}(V_{\tilde{\mathcal{P}}}^{\tilde{\pi}}) - \sigma_{\mathcal{P}_s^{\tilde{\pi}}}(V_{\mathcal{P}}^{\tilde{\pi}}))$$
$$= \gamma(\sigma_{\mathcal{P}_s^{\tilde{\pi}}}(V_{\tilde{\mathcal{P}}}^{\tilde{\pi}}) - \sigma_{\mathcal{P}_s^{\tilde{\pi}}}(V_{\mathcal{P}}^{\tilde{\pi}})) + \gamma(\sigma_{\tilde{\mathcal{P}}_s^{\tilde{\pi}}}(V_{\tilde{\mathcal{P}}}^{\tilde{\pi}}) - \sigma_{\mathcal{P}_s^{\tilde{\pi}}}(V_{\tilde{\mathcal{P}}}^{\tilde{\pi}}))$$
$$\overset{(a)}{\le} \gamma q_s^{\tilde{\pi}}(V_{\tilde{\mathcal{P}}}^{\tilde{\pi}} - V_{\mathcal{P}}^{\tilde{\pi}}) + b(s), \tag{128}$$

where $q_s^{\tilde{\pi}}$ is the worst-case transition kernel of $V_{\mathcal{P}}^{\tilde{\pi}}$ in $\mathcal{P}_s^{\tilde{\pi}}$, and $b(s) \triangleq \gamma(\sigma_{\tilde{\mathcal{P}}_s^{\tilde{\pi}}}(V_{\tilde{\mathcal{P}}}^{\tilde{\pi}}) - \sigma_{\mathcal{P}_s^{\tilde{\pi}}}(V_{\tilde{\mathcal{P}}}^{\tilde{\pi}}))$.

Recursively applying (128) implies

$$\rho^\top\Delta_2 \le \frac{1}{1-\gamma}\langle d_q^{\tilde{\pi}}, b\rangle. \tag{129}$$

We further introduce two sets as follows.

$$\mathcal{S}_1 = \{s : \mu(s,\tilde{\pi}(s)) = 0\}, \tag{130}$$
$$\mathcal{S}_2 = \{s : \mu(s,\tilde{\pi}(s)) > 0\}. \tag{131}$$

For $s \in \mathcal{S}_1$, $\tilde{\mathcal{P}}_s^{\tilde{\pi}} = \Delta(\mathcal{S})$, hence

$$b(s) = \gamma(\sigma_{\tilde{\mathcal{P}}_s^{\tilde{\pi}}}(V_{\tilde{\mathcal{P}}}^{\tilde{\pi}}) - \sigma_{\mathcal{P}_s^{\tilde{\pi}}}(V_{\tilde{\mathcal{P}}}^{\tilde{\pi}})) \le 0. \tag{132}$$

For $s \in \mathcal{S}_2$, we have that

$$b(s) = \gamma(\sigma_{\tilde{\mathcal{P}}_s^{\tilde{\pi}}}(V_{\tilde{\mathcal{P}}}^{\tilde{\pi}}) - \sigma_{\mathcal{P}_s^{\tilde{\pi}}}(V_{\tilde{\mathcal{P}}}^{\tilde{\pi}}))$$

$$= \gamma(\sigma_{\tilde{\mathcal{P}}^{\tilde{\pi}}_s}(V^{\tilde{\pi}}_{\tilde{\mathcal{P}}}) - \sigma_{\hat{\mathcal{P}}^{\tilde{\pi}}_s}(V^{\tilde{\pi}}_{\tilde{\mathcal{P}}}) + \sigma_{\hat{\mathcal{P}}^{\tilde{\pi}}_s}(V^{\tilde{\pi}}_{\tilde{\mathcal{P}}}) - \sigma_{\mathcal{P}^{\tilde{\pi}}_s}(V^{\tilde{\pi}}_{\tilde{\mathcal{P}}})). \tag{133}$$

Hence invoke Lemma 17, we have that for $s \in \mathcal{S}_2$, with probability at least $1 - \delta$,

$$\sigma_{\hat{\mathcal{P}}^{\tilde{\pi}}_s}(V^{\tilde{\pi}}_{\tilde{\mathcal{P}}}) - \sigma_{\mathcal{P}^{\tilde{\pi}}_s}(V^{\tilde{\pi}}_{\tilde{\mathcal{P}}}) \le \min\left\{ \frac{1}{1-\gamma}, \frac{4}{NR(1-\gamma)} + \frac{C_1}{R(1-\gamma)}\sqrt{\frac{\log\frac{2(1+R)N^3 S}{(1-\gamma)\delta}}{N(s,\tilde{\pi}(s))\min_x \hat{\mathsf{P}}^{\tilde{\pi}}_{s,x}}} \right\}. \tag{134}$$

To further bound the RHS of (134), we first note that Lemma 8 of [36] states that if $N\mu(s,a) \ge 8\log\frac{1}{\delta}$, then with probability $1 - \delta$, for any $(s,a)$ pair,

$$N(s,a) \ge \frac{N\mu(s,a)}{8\log\frac{4}{\delta}}. \tag{135}$$

This moreover implies that with probability $1 - \delta$, for $s \in \mathcal{S}_2$,

$$N(s,\tilde{\pi}(s)) \ge \frac{N\mu(s,\tilde{\pi}(s))}{8\log\frac{4}{\delta}}. \tag{136}$$

On the other hand, (202) of [36] states that with probability at least $1 - \delta$,

$$\frac{\min_x \mathsf{P}^{\tilde{\pi}}_{s,x}}{8\log(NS/\delta)} \le \min_x \hat{\mathsf{P}}^{\tilde{\pi}}_{s,x} \le e^2 \min_x \mathsf{P}^{\tilde{\pi}}_{s,x}. \tag{137}$$

Hence by plugging (136) and (137) in (134) we have that

$$\sigma_{\hat{\mathcal{P}}^{\tilde{\pi}}_s}(V^{\tilde{\pi}}_{\tilde{\mathcal{P}}}) - \sigma_{\mathcal{P}^{\tilde{\pi}}_s}(V^{\tilde{\pi}}_{\tilde{\mathcal{P}}}) \le \frac{4}{NR(1-\gamma)} + \frac{C_1}{R(1-\gamma)}\sqrt{\frac{8\log\frac{NS}{\delta}\log\frac{2(1+R)N^3 S}{(1-\gamma)\delta}}{N(s,\tilde{\pi}(s))\mathsf{P}_{\min}}}. \tag{138}$$

On the other hand, similarly to (123), it holds that

$$\sigma_{\hat{\mathcal{P}}^{\tilde{\pi}}_s}(V^{\tilde{\pi}}_{\tilde{\mathcal{P}}}) - \sigma_{\tilde{\mathcal{P}}^{\tilde{\pi}}_s}(V^{\tilde{\pi}}_{\tilde{\mathcal{P}}})$$

$$= \max_{0 \le \lambda \le \frac{1}{(1-\gamma)R}}\left\{ -\lambda\log\left(\hat{\mathsf{P}}^{\tilde{\pi}}_s \mathbf{exp}\left(\frac{-V^{\tilde{\pi}}_{\tilde{\mathcal{P}}}}{\lambda}\right)\right) - \lambda R \right\}$$

$$\quad - \max_{0 \le \lambda \le \frac{1}{(1-\gamma)(R+\kappa^{\tilde{\pi}(s)}_s)}}\left\{ -\lambda\log\left(\hat{\mathsf{P}}^{\tilde{\pi}}_s \mathbf{exp}\left(\frac{-V^{\tilde{\pi}}_{\tilde{\mathcal{P}}}}{\lambda}\right)\right) - \lambda(R + \kappa^{\tilde{\pi}(s)}_s) \right\}$$

$$\overset{(a)}{\ge} \frac{1}{(1-\gamma)R}\kappa^{\tilde{\pi}(s)}_s, \tag{139}$$

where the last inequality is from the fact that $\max F - \max G \ge F(x) - G(x), \forall x$.

Combining with(134) further implies that

$$\sigma_{\hat{\mathcal{P}}^{\tilde{\pi}}_s}(V^{\tilde{\pi}}_{\tilde{\mathcal{P}}}) - \sigma_{\mathcal{P}^{\tilde{\pi}}_s}(V^{\tilde{\pi}}_{\tilde{\mathcal{P}}})$$

$$\le \frac{4}{NR(1-\gamma)} + \frac{C_1}{R(1-\gamma)}\sqrt{\frac{\log\frac{2(1+R)N^3 S}{(1-\gamma)\delta}}{N(s,\tilde{\pi}(s))\min_x \hat{\mathsf{P}}^{\tilde{\pi}}_{s,x}}}$$

$$\le \frac{4}{NR(1-\gamma)} + \sigma_{\hat{\mathcal{P}}^{\tilde{\pi}}_s}(V^{\tilde{\pi}}_{\tilde{\mathcal{P}}}) - \sigma_{\tilde{\mathcal{P}}^{\tilde{\pi}}_s}(V^{\tilde{\pi}}_{\tilde{\mathcal{P}}}), \tag{140}$$

by combining (138) and (139). Thus

$$\sigma_{\tilde{\mathcal{P}}^{\tilde{\pi}}_s}(V^{\tilde{\pi}}_{\tilde{\mathcal{P}}}) - \sigma_{\mathcal{P}^{\tilde{\pi}}_s}(V^{\tilde{\pi}}_{\tilde{\mathcal{P}}}) \le \frac{4}{NR(1-\gamma)} \tag{141}$$

Hence combine with (129) and (132), we further have that

$$\rho^\top \Delta_2 \le \frac{4}{NR(1-\gamma)^2}. \tag{142}$$

$\square$

# E  Auxiliary Lemmas

**Lemma 10.** *It holds that*

$$\textbf{Span}(V_{\tilde{\mathcal{P}}}^{\tilde{\pi}}) \leq \frac{1}{\gamma \max\{R, 1 - \gamma\}}. \tag{143}$$

*Proof.* Note that when $N > \frac{8 \log \frac{4SA}{\delta}}{(1-R)\mu_{\min}}$ and the fact that $N(s, a) \geq \frac{8 \log \frac{4SA}{\delta}}{N\mu_{\min}}$, it holds that

$$
\begin{aligned}
R + \kappa_s^{\tilde{\pi}(s)} &\leq R + \frac{1}{N(s, \tilde{\pi}(s))} \\
&\leq R + \frac{8 \log \frac{4SA}{\delta}}{N\mu_{\min}} \\
&< 1,
\end{aligned} \tag{144}
$$

Denote $s^* = \arg\min_s V_{\tilde{\mathcal{P}}}^{\tilde{\pi}}(s)$. Then it holds that

$$V_{\tilde{\mathcal{P}}}^{\tilde{\pi}}(s^*) = \{\hat{r}(s^*, \tilde{\pi}(s)) + \gamma \sigma_{\tilde{\mathcal{P}}_s^{\tilde{\pi}(s)}}(V_{\tilde{\mathcal{P}}}^{\tilde{\pi}})\}. \tag{145}$$

We denote the optimal action $\tilde{\pi}(s)$ by $a$ in the following proof. Note that there exists a vector $q_s^a \in \mathbb{R}^S$, such that $\hat{\mathsf{P}}_s^a \geq q_s^a \geq 0$, and $\sum_{s'} q_s^a(s') = 1 - R - \kappa_s^a$. This is doable because $\sum_{s'} \hat{\mathsf{P}}_s^a(s') = 1$ and $R + \kappa_s^a \leq 1$. Hence it implies that the transition kernel $q_s^a + (R + \kappa_s^a)\mathbf{1}_{s_*} \in \tilde{\mathcal{P}}_s^a$, since $\|q_s^a + (R + \kappa_s^a)\mathbf{1}_{s_*} - \hat{\mathsf{P}}_s^a\| \leq \|q_s^a - \hat{\mathsf{P}}_s^a\| + (R + \kappa_s^a) \leq 2(R + \kappa_s^a)$.

Hence

$$
\begin{aligned}
\sigma_{\tilde{\mathcal{P}}_s^a}(V_{\tilde{\mathcal{P}}}^{\tilde{\pi}}) &\leq (q_s^a + (R + \kappa_s^a)\mathbf{1}_{s_*})V_{\tilde{\mathcal{P}}}^{\tilde{\pi}} \\
&\leq (R + \kappa_s^a)V_{\tilde{\mathcal{P}}}^{\tilde{\pi}}(s_*) + q_s^a V_{\tilde{\mathcal{P}}}^{\tilde{\pi}} \\
&\leq (R + \kappa_s^a)V_{\tilde{\mathcal{P}}}^{\tilde{\pi}}(s_*) + \|q_s^a\|_1 \|V_{\tilde{\mathcal{P}}}^{\tilde{\pi}}\| \\
&= (R + \kappa_s^a)V_{\tilde{\mathcal{P}}}^{\tilde{\pi}}(s_*) + (1 - R - \kappa_s^a)V_{\tilde{\mathcal{P}}}^{\tilde{\pi}}(s^*),
\end{aligned} \tag{146}
$$

where the last equation is from $\sum_{s'} q_s^a(s') = 1 - R - \kappa_s^a$ and $\|V_{\tilde{\mathcal{P}}}^{\tilde{\pi}}\| = \max_s V_{\tilde{\mathcal{P}}}^{\tilde{\pi}} = V_{\tilde{\mathcal{P}}}^{\tilde{\pi}}(s^*)$.

Plug this inequality in (145), and it holds that

$$
\begin{aligned}
V_{\tilde{\mathcal{P}}}^{\tilde{\pi}}(s^*) = V_{\max} &\leq \hat{r}(s^*, a) + \gamma(R + \kappa_s^a)V_{\tilde{\mathcal{P}}}^{\tilde{\pi}}(s_*) + \gamma(1 - R - \kappa_s^a)V_{\tilde{\mathcal{P}}}^{\tilde{\pi}}(s^*) \\
&\leq 1 + \gamma(R + \kappa_s^a)V_{\min} + \gamma(1 - R - \kappa_s^a)V_{\max}.
\end{aligned} \tag{147}
$$

Thus

$$
\begin{aligned}
V_{\max} &\leq \frac{1 + \gamma(R + \kappa_s^a)V_{\min}}{1 - \gamma(1 - R - \kappa_s^a)} \\
&= \frac{1 + \gamma(R + \kappa_s^a)V_{\min}}{1 - \gamma + \gamma R + \gamma \kappa_s^a} \\
&\leq \frac{1}{1 - \gamma + \gamma(R + \kappa_s^a)} + V_{\min},
\end{aligned} \tag{148}
$$

which implies that

$$\textbf{Span}(V_{\tilde{\mathcal{P}}}^{\tilde{\pi}}) \leq \frac{1}{1 - \gamma + \gamma(R + \kappa_s^a)} \leq \frac{1}{\gamma \max\{R, 1 - \gamma\}}. \tag{149}$$

$\square$

**Lemma 11.** *Recall the set $\mathcal{S}^0 \triangleq \{s \in \mathcal{S} : N(s) = 0\}$. Then*

*(1). For any policy $\pi$ and $s \in \mathcal{S}^0$, $V_{\tilde{\mathcal{P}}}^{\pi}(s) = 0$;*

*(2). There exists a deterministic robust optimal policy $\tilde{\pi}$, such that for any $s \notin \mathcal{S}^0$, $N(s, \tilde{\pi}(s)) > 0$.*

*Proof.* **Proof of (1).**

For any $s \in \mathcal{S}^0$, it holds that $N(s, a) = 0$ for any $a \in \mathcal{A}$. Hence $\hat{r}(s, a) = 0$ and $\tilde{\mathcal{P}}_s^a = \Delta(\mathcal{S})$.

Then for any policy $\pi$ and $a \in \mathcal{A}$, it holds that

$$Q_{\tilde{\mathcal{P}}}^\pi(s, a) = \hat{r}(s, a) + \gamma \sigma_{\tilde{\mathcal{P}}_s^a}(V_{\tilde{\mathcal{P}}}^\pi) \leq \gamma V_{\tilde{\mathcal{P}}}^\pi(s). \tag{150}$$

Thus

$$V_{\tilde{\mathcal{P}}}^\pi(s) = \sum_a \pi(a|s) Q_{\tilde{\mathcal{P}}}^\pi(s, a) \leq \gamma V_{\tilde{\mathcal{P}}}^\pi(s), \tag{151}$$

which implies $V_{\tilde{\mathcal{P}}}^\pi(s) = 0$ together with the fact that $V_{\tilde{\mathcal{P}}}^\pi \geq 0$.

**Proof of (2).**

We prove Claim (2) by contradiction. Assume that for any optimal policy $\tilde{\pi}$, there exists $s \notin \mathcal{S}^0$ such that $N(s, \tilde{\pi}(s)) = 0$. We then consider a fixed pair $(\tilde{\pi}, s)$.

$N(s, \tilde{\pi}(s)) = 0$ further implies $\hat{r}(s, \tilde{\pi}(s)) = 0$, $\tilde{\mathcal{P}}_s^{\tilde{\pi}(s)} = \Delta(\mathcal{S})$, and

$$V_{\tilde{\mathcal{P}}}^{\tilde{\pi}}(s) = \max_a Q_{\tilde{\mathcal{P}}}^{\tilde{\pi}}(s, a) = Q_{\tilde{\mathcal{P}}}^{\tilde{\pi}}(s, \tilde{\pi}(s)) = \hat{r}(s, \tilde{\pi}(s)) + \gamma \sigma_{\tilde{\mathcal{P}}_s^{\tilde{\pi}(s)}}(V_{\tilde{\mathcal{P}}}^{\tilde{\pi}}) \leq \gamma V_{\tilde{\mathcal{P}}}^{\tilde{\pi}}(s), \tag{152}$$

where the last inequality is from $\tilde{\mathcal{P}}_s^{\tilde{\pi}(s)} = \Delta(\mathcal{S})$, $\hat{r}(s, \tilde{\pi}(s)) = 0$, and $\sigma_{\tilde{\mathcal{P}}_s^{\tilde{\pi}(s)}}(V_{\tilde{\mathcal{P}}}^{\tilde{\pi}}) \leq \mathbf{1}_s V_{\tilde{\mathcal{P}}}^{\tilde{\pi}} = V_{\tilde{\mathcal{P}}}^{\tilde{\pi}}(s)$. This further implies that $V_{\tilde{\mathcal{P}}}^{\tilde{\pi}}(s) = 0$ because $V_{\tilde{\mathcal{P}}}^{\tilde{\pi}} \geq 0$.

On the other hand, since $s \notin \mathcal{S}^0$, there exists another action $b \neq \tilde{\pi}(s)$ such that $N(s, b) > 0$, and hence $\hat{r}(s, b) = r(s, b)$. We consider the following two cases.

(I). If $r(s, b) > 0$, then

$$Q_{\tilde{\mathcal{P}}}^{\tilde{\pi}}(s, b) = \hat{r}(s, b) + \gamma \sigma_{\tilde{\mathcal{P}}_s^b}(V_{\tilde{\mathcal{P}}}^{\tilde{\pi}}) > 0 = Q_{\tilde{\mathcal{P}}}^{\tilde{\pi}}(s, \tilde{\pi}(s)), \tag{153}$$

which is contradict to $V_{\tilde{\mathcal{P}}}^{\tilde{\pi}}(s) = \max_a Q_{\tilde{\mathcal{P}}}^{\tilde{\pi}}(s, a) = Q_{\tilde{\mathcal{P}}}^{\tilde{\pi}}(s, \tilde{\pi}(s))$.

(II). If $r(s, b) = 0$, Lemma 12 then implies the modified policy $f_b^s(\tilde{\pi})$ is also optimal, and satisfies $N(x, f_b^s(\tilde{\pi})(x)) = N(x, \tilde{\pi}(x))$ for any $x \neq s$, and $N(s, f_b^s(\tilde{\pi})(s)) > 0$.

Then consider the modified policy $f_b^s(\tilde{\pi})$.

If there still exists $s' \notin \mathcal{S}^0$ such that $N(s', f_b^s(\tilde{\pi})(s')) = 0$, then similarly, there exists another action $b' \neq f_b^s(\tilde{\pi})(s')$ such that $N(s', b') > 0$. Then whether $r(s', b') > 0$, which falls into Case (I) and leads to a contradiction, or applying Lemma 12 again implies another optimal policy $f_{b'}^{s'}(f_b^s(\tilde{\pi}))$, such that $N(s, f_{b'}^{s'}(f_b^s(\tilde{\pi}))(x)) = N(s, f_b^s(\tilde{\pi})(x)) > 0$ for $x \notin \{s, s'\}$, $N(s, f_{b'}^{s'}(f_b^s(\tilde{\pi}))(s)) = N(s, f_b^s(\tilde{\pi})(s)) > 0$ and $N(s', f_{b'}^{s'}(f_b^s(\tilde{\pi}))(s')) > 0$.

Repeating this procedure recursively further implies there exists an optimal policy $\pi$, such that $N(s, \pi(s)) > 0$ for any $s \notin \mathcal{S}^0$, which is a contraction to our assumption.

Therefore it completes the proof. □

**Lemma 12.** *For a robust optimal policy $\tilde{\pi}$, if there exists a state $s \notin \mathcal{S}^0$ and an action $b$ such that $N(s, \tilde{\pi}(s)) = 0$, $r(s, b) = 0$ and $N(s, b) > 0$, define a modified policy $f_b^s(\tilde{\pi})$ as*

$$f_b^s(\tilde{\pi})(s) = b, \tag{154}$$
$$f_b^s(\tilde{\pi})(x) = \tilde{\pi}(x), \text{ for } x \neq s. \tag{155}$$

*Then the modified policy $f_b^s(\tilde{\pi})$ is also optimal, and satisfies $N(s, f_b^s(\tilde{\pi})(s)) > 0$, $N(x, f_b^s(\tilde{\pi})(x)) = N(x, \tilde{\pi}(x)), \forall x \neq s$.*

*Proof.* Recall that $\tilde{\mathcal{P}}_s^{\tilde{\pi}(s)} = \Delta(\mathcal{S})$ and $\tilde{\mathcal{P}}_s^b \subseteq \Delta(\mathcal{S})$, we have that

$$V_{\tilde{\mathcal{P}}}^{f_b^s(\tilde{\pi})} \geq V_{\tilde{\mathcal{P}}_s^b}^{f_b^s(\tilde{\pi})}, \tag{156}$$

where $\tilde{\mathcal{P}}_s^b$ is a modified uncertainty set defined as

$$(\tilde{\mathcal{P}}_s^b)_s^b = \Delta(\mathcal{S}), \tag{157}$$

$$(\tilde{\mathcal{P}}_s^b)_x^a = \tilde{\mathcal{P}}_x^a, \text{ for } (x,a) \neq (s,b). \tag{158}$$

Now we have that

$$V_{\tilde{\mathcal{P}}_s^b}^{f_b^s(\tilde{\pi})}(s) = Q_{\tilde{\mathcal{P}}_s^b}^{f_b^s(\tilde{\pi})}(s,b) = r(s,b) + \gamma\sigma_{(\tilde{\mathcal{P}}_s^b)_s^b}(V_{\tilde{\mathcal{P}}_s^b}^{f_b^s(\tilde{\pi})}) \leq \gamma V_{\tilde{\mathcal{P}}_s^b}^{f_b^s(\tilde{\pi})}(s), \tag{159}$$

which further implies $V_{\tilde{\mathcal{P}}_s^b}^{f_b^s(\tilde{\pi})}(s) = 0$. Note that in eq. (152), we have shown $V_{\tilde{\mathcal{P}}}^{\tilde{\pi}}(s) = 0$, hence $V_{\tilde{\mathcal{P}}_s^b}^{f_b^s(\tilde{\pi})}(s) = V_{\tilde{\mathcal{P}}}^{\tilde{\pi}}(s) = 0$.

Now consider the two robust Bellman operator $\mathbf{T}_b^s V(x) = \sum_a f_b^s(\tilde{\pi})(a|x)(\hat{r}(x,a) + \gamma\sigma_{(\tilde{\mathcal{P}}_s^b)_x^a}(V))$ and $\mathbf{T}V(x) = \hat{r}(x, \tilde{\pi}(x)) + \gamma\sigma_{\tilde{\mathcal{P}}_x^{\tilde{\pi}(x)}}(V)$. It is known that $V_{\tilde{\mathcal{P}}_s^b}^{f_b^s(\tilde{\pi})}$ is the unique fixed point of the robust Bellman operator $\mathbf{T}_b^s$ and $V_{\tilde{\mathcal{P}}}^{\tilde{\pi}}$ is the unique fixed point of $\mathbf{T}$.

When $x \neq s$,

$$\begin{aligned}
\mathbf{T}_b^s V_{\tilde{\mathcal{P}}}^{\tilde{\pi}}(x) &= \sum_a f_b^s(\tilde{\pi})(a|x)(\hat{r}(x,a) + \gamma\sigma_{(\tilde{\mathcal{P}}_s^b)_x^a}(V_{\tilde{\mathcal{P}}}^{\tilde{\pi}})) \\
&\overset{(a)}{=} \sum_a \tilde{\pi}(a|x)(\hat{r}(x,a) + \gamma\sigma_{(\tilde{\mathcal{P}}_s^b)_x^a}(V_{\tilde{\mathcal{P}}}^{\tilde{\pi}})) \\
&= \hat{r}(x, \tilde{\pi}(x)) + \gamma\sigma_{(\tilde{\mathcal{P}}_s^b)_x^{\tilde{\pi}(x)}}(V_{\tilde{\mathcal{P}}}^{\tilde{\pi}}) \\
&\overset{(b)}{=} \hat{r}(x, \tilde{\pi}(x)) + \gamma\sigma_{\tilde{\mathcal{P}}_x^{\tilde{\pi}(x)}}(V_{\tilde{\mathcal{P}}}^{\tilde{\pi}}) \\
&= \mathbf{T}V_{\tilde{\mathcal{P}}}^{\tilde{\pi}}(x) = V_{\tilde{\mathcal{P}}}^{\tilde{\pi}}(x), \tag{160}
\end{aligned}$$

where $(a)$ is from $f_b^s(\tilde{\pi})(x) = \tilde{\pi}(x)$ when $x \neq s$, $(b)$ is from $(\tilde{\mathcal{P}}_s^b)_x^{\tilde{\pi}(x)} = \tilde{\mathcal{P}}_x^{\tilde{\pi}(x)}$.

And for $s$, it holds that

$$\begin{aligned}
\mathbf{T}_b^s V_{\tilde{\mathcal{P}}}^{\tilde{\pi}}(s) &= \hat{r}(s,b) + \gamma\sigma_{(\tilde{\mathcal{P}}_s^b)_s^b}(V_{\tilde{\mathcal{P}}}^{\tilde{\pi}}) \\
&\overset{(a)}{=} \hat{r}(s, \tilde{\pi}(s)) + \gamma\sigma_{\Delta(\mathcal{S})}(V_{\tilde{\mathcal{P}}}^{\tilde{\pi}}) \\
&\overset{(b)}{=} \hat{r}(s, \tilde{\pi}(s)) + \gamma\sigma_{\tilde{\mathcal{P}}_s^{\tilde{\pi}(s)}}(V_{\tilde{\mathcal{P}}}^{\tilde{\pi}}) \\
&= \mathbf{T}V_{\tilde{\mathcal{P}}}^{\tilde{\pi}}(s) \\
&= V_{\tilde{\mathcal{P}}}^{\tilde{\pi}}(s), \tag{161}
\end{aligned}$$

where $(a)$ is from $(\tilde{\mathcal{P}}_s^b)_s^b = \Delta(\mathcal{S})$ and $\hat{r}(s,b) = r(s,b) = 0 = \hat{r}(s, \tilde{\pi}(s))$, and $(b)$ follows from the fact $\tilde{\mathcal{P}}_s^{\tilde{\pi}(s)} = \Delta(\mathcal{S})$.

eq. (160) and eq. (161) further imply that $V_{\tilde{\mathcal{P}}}^{\tilde{\pi}}$ is also a fixed point of $\mathbf{T}_b^s$. Hence it must be identical to $V_{\tilde{\mathcal{P}}_s^b}^{f_b^s(\tilde{\pi})}$, i.e., $V_{\tilde{\mathcal{P}}_s^b}^{f_b^s(\tilde{\pi})} = V_{\tilde{\mathcal{P}}}^{\tilde{\pi}}$.

Combine with eq. (156), we have

$$V_{\tilde{\mathcal{P}}}^{f_b^s(\tilde{\pi})} \geq V_{\tilde{\mathcal{P}}_s^b}^{f_b^s(\tilde{\pi})} = V_{\tilde{\mathcal{P}}}^{\tilde{\pi}}, \tag{162}$$

which implies that $f_b^s(\tilde{\pi})$ is also optimal with $N(s, f_b^s(\tilde{\pi})(s)) = N(s,b) > 0$. And since $f_b^s(\tilde{\pi})(x) = \tilde{\pi}(x)$ for $x \neq s$, then $N(x, f_b^s(\tilde{\pi})(x)) = N(x, \tilde{\pi}(x))$. This thus completes the proof. $\square$

**Lemma 13** (Lemma 4, [19]). *For any $\delta$, with probability $1 - \delta$, $\max\{12N(s,a), 8\log\frac{NS}{\delta}\} \geq N\mu(s,a), \forall s,a.$*

**Lemma 14** (Lemma 9, [19]). *For any $(s,a)$ pair with $N(s,a) > 0$, if $V$ is an vector independent of $\hat{\mathsf{P}}_s^a$ obeying $\|V\| \le \frac{1}{1-\gamma}$, then with probability at least $1 - \delta$,*

$$|(\hat{\mathsf{P}}_s^a - \mathsf{P}_s^a)V| \le \sqrt{\frac{48 \boldsymbol{Var}_{\hat{\mathsf{P}}_s^a}(V) \log \frac{4N}{\delta}}{N(s,a)}} + \frac{48 \log \frac{4N}{\delta}}{(1-\gamma)N(s,a)}, \tag{163}$$

$$\boldsymbol{Var}_{\hat{\mathsf{P}}_s^a}(V) \le 2\boldsymbol{Var}_{\mathsf{P}_s^a}(V) + \frac{5 \log \frac{4N}{\delta}}{3(1-\gamma)^2 N(s,a)}. \tag{164}$$

**Lemma 15.** *For the total variation uncertain set and for any fixed $s, a$ and $\delta$, with probability at least $1 - \delta$, it holds that*

$$|\sigma_s^a(V_{\tilde{\mathcal{P}}}^{\tilde{\pi}}) - \hat{\sigma}_s^a(V_{\tilde{\mathcal{P}}}^{\tilde{\pi}})|$$

$$\le \frac{1}{N(s,a)(1-\gamma)}\left(2 + c_2 \log\left(\frac{4SAN^2}{\delta}\right) + \sqrt{2c_1 \log\left(\frac{4SAN^2}{\delta}\right)}\right) + \sqrt{\frac{c_1 \log\left(\frac{4SAN^2}{\delta}\right)\boldsymbol{Var}_{\mathsf{P}_s^a}(V_{\tilde{\mathcal{P}}}^{\tilde{\pi}})}{N(s,a)}}, \tag{165}$$

*and*

$$\boldsymbol{Var}_{\hat{\mathsf{P}}_s^a}(V_{\tilde{\mathcal{P}}}^{\tilde{\pi}}) \le 2\boldsymbol{Var}_{\mathsf{P}_s^a}(V_{\tilde{\mathcal{P}}}^{\tilde{\pi}}) + \frac{41 \log \frac{2N}{(1-\gamma)\delta}}{(1-\gamma)^2 N(s,a)}. \tag{166}$$

*Proof.* **Step 1: Construct an auxiliary robust MDP.** For the state $s \in \mathcal{S}$ and any constant $u \in [0,1]$, we construct the following MDP $M^{s,u} = (\mathcal{S}, \mathcal{A}, r^{s,u}, \mathsf{P}^{s,u})$ with transition kernel $\mathsf{P}^{s,u}$ and reward $r^{s,u}$:

$$(\mathsf{P}^{s,u})_x^a = \mathbf{1}_s, \text{ if } x = s; \tag{167}$$

$$(\mathsf{P}^{s,u})_x^a = \hat{\mathsf{P}}_x^a, \text{ if } x \ne s. \tag{168}$$

and

$$r^{s,u}(x,a) = u, \text{ if } x = s; \tag{169}$$

$$r^{s,u}(x,a) = r(x,a), \text{ if } x \ne s. \tag{170}$$

Centered at $M^{s,u}$, we further construct the following robust MDP $\tilde{M}^{s,u} = (\mathcal{S}, \mathcal{A}, \tilde{\mathcal{P}}^{s,u}, r^{s,u})$:

$$(\tilde{\mathcal{P}}^{s,u})_x^a = \{(\mathsf{P}^{s,u})_x^a\}, \text{ if } x = s; \tag{171}$$

$$(\tilde{\mathcal{P}}^{s,u})_x^a = \tilde{\mathcal{P}}_x^a, \text{ if } x \ne s. \tag{172}$$

$$\tag{173}$$

The robust Bellman operator associated with this robust MDP is hence

$$\tilde{\mathbf{T}}^{s,u}V(x) = \max_a\{r^{s,u}(x,a) + \gamma\sigma_{(\tilde{\mathcal{P}}^{s,u})_x^a}(V)\}, \tag{174}$$

and we denote the robust value function w.r.t. it by $V^{s,u}$.

**Step 2: Prove $V_{\tilde{\mathcal{P}}}^{\tilde{\pi}}$ is a robust value function w.r.t. $\tilde{M}^{s,u^*}$ for some $u^* \in [0,1]$.** We claim that if we set $u^* = (1-\gamma)V_{\tilde{\mathcal{P}}}^{\tilde{\pi}}(s) \in [0,1]$, then the robust value function w.r.t. $\tilde{M}^{s,u^*}$ is equal to $V_{\tilde{\mathcal{P}}}^{\tilde{\pi}}$.

To prove the claim, recall that $V_{\tilde{\mathcal{P}}}^{\tilde{\pi}}$ is the unique fixed point of the optimal robust Bellman operator

$$\tilde{\mathbf{T}}V(x) = \max_a\{r(x,a) + \gamma\sigma_{(\tilde{\mathcal{P}})_x^a}(V)\}. \tag{175}$$

For the state $s$, note that

$$\begin{aligned}
\tilde{\mathbf{T}}^{s,u}V_{\tilde{\mathcal{P}}}^{\tilde{\pi}}(s) &= \max_a\{r^{s,u^*}(s,a) + \gamma\sigma_{(\tilde{\mathcal{P}}^{s,u})_x^a}(V_{\tilde{\mathcal{P}}}^{\tilde{\pi}})\}\\
&= \max_a\{u^* + \gamma(\mathsf{P}^{s,u})_s^a(V_{\tilde{\mathcal{P}}}^{\tilde{\pi}})\}\\
&= \max_a\{(1-\gamma)V_{\tilde{\mathcal{P}}}^{\tilde{\pi}}(s) + \gamma V_{\tilde{\mathcal{P}}}^{\tilde{\pi}}(s)\}
\end{aligned}$$

$$= V_{\tilde{\mathcal{P}}}^{\tilde{\pi}}(s), \tag{176}$$

which is due to the construction of $\tilde{M}^{s,u^*}$; And for states $x \neq s$, we have that

$$\begin{aligned}
\tilde{\mathbf{T}}^{s,u} V_{\tilde{\mathcal{P}}}^{\tilde{\pi}}(x) &= \max_a \{ r^{s,u^*}(x,a) + \gamma \sigma_{(\tilde{\mathcal{P}}^{s,u})_x^a}(V_{\tilde{\mathcal{P}}}^{\tilde{\pi}}) \} \\
&= \max_a \{ r(x,a) + \gamma \sigma_{\tilde{\mathcal{P}}_x^a}(V_{\tilde{\mathcal{P}}}^{\tilde{\pi}}) \} \\
&= \tilde{\mathbf{T}}(V_{\tilde{\mathcal{P}}}^{\tilde{\pi}})(x) \\
&= V_{\tilde{\mathcal{P}}}^{\tilde{\pi}}(x), \tag{177}
\end{aligned}$$

which is from the construction of $\tilde{M}^{s,u^*}$ and the fact that $V_{\tilde{\mathcal{P}}}^{\tilde{\pi}}$ is the fixed point of $\tilde{\mathbf{T}}$.

These two equations hence imply that $V_{\tilde{\mathcal{P}}}^{\tilde{\pi}}$ is a fixed point of the robust Bellman operator $\mathbf{T}^{s,u^*}$, which further proves the claim.

**Step 3: Decouple the dependence of $V_{\tilde{\mathcal{P}}}^{\tilde{\pi}}$ on $\hat{\mathsf{P}}$ by constructing an $\frac{1}{N}$-net.**

Define $\mathcal{U} = \{ \frac{i}{N} : i = 0, 1, ..., N \}$. Clearly, $\mathcal{U}$ is a $\frac{1}{N}$-net [43] of the interval $[0,1]$, i.e., for any $u \in [0,1]$, there exists $u_i \in \mathcal{U}$, such that $|u - u_i| \leq \frac{1}{N}$.

Clearly, each $u \in \mathcal{U}$ is a constant independent with $\hat{\mathsf{P}}_s^a$, hence applying Lemma 14 implies that for any $u \in \mathcal{U}$,

$$|\mathsf{P}_s^a V^{s,u} - \hat{\mathsf{P}}_s^a V^{s,u}| \leq \sqrt{\frac{c_1 \log(\frac{4N}{\delta}) \mathbf{Var}_{\mathsf{P}_s^a}(V^{s,u})}{N(s,a)}} + \frac{c_2 \log(\frac{4N}{\delta})}{(1-\gamma)N(s,a)}. \tag{178}$$

Take the union bound over any $(s, a, u) \in \mathcal{S} \times \mathcal{A} \times \mathcal{U}$, and we have the following inequality holds with probability at least $1 - \delta$:

$$|\mathsf{P}_s^a V^{s,u} - \hat{\mathsf{P}}_s^a V^{s,u}| \leq \sqrt{\frac{c_1 \log(\frac{4SAN^2}{\delta}) \mathbf{Var}_{\mathsf{P}_s^a}(V^{s,u})}{N(s,a)}} + \frac{c_2 \log(\frac{4SAN^2}{\delta})}{(1-\gamma)N(s,a)}. \tag{179}$$

**Step 4: Approximate $|\mathsf{P}_s^a V_{\tilde{\mathcal{P}}}^{\tilde{\pi}} - \hat{\mathsf{P}}_s^a V_{\tilde{\mathcal{P}}}^{\tilde{\pi}}|$ using the $\frac{1}{N}$-net.**

As we showed above, there exists $u \in \mathcal{U}$ such that $|u - u^*| \leq \frac{1}{N}$. Moreover, note that

$$\begin{aligned}
\|V^{s,u} - V^{s,u^*}\| &= \|\tilde{\mathbf{T}}^{s,u} V^{s,u} - \tilde{\mathbf{T}}^{s,u^*} V^{s,u^*}\| \\
&\leq \|\tilde{\mathbf{T}}^{s,u} V^{s,u} - \tilde{\mathbf{T}}^{s,u} V^{s,u^*}\| + \|\tilde{\mathbf{T}}^{s,u} V^{s,u^*} - \tilde{\mathbf{T}}^{s,u^*} V^{s,u^*}\| \\
&\leq \gamma \|V^{s,u} - V^{s,u^*}\| + \|r^{s,u} - r^{s,u^*}\| + \gamma \|\sigma_{(\tilde{\mathcal{P}}^{s,u})}(V^{s,u^*}) - \sigma_{(\tilde{\mathcal{P}}^{s,u^*})}(V^{s,u^*})\| \\
&\leq \gamma \|V^{s,u} - V^{s,u^*}\| + \frac{1}{N}, \tag{180}
\end{aligned}$$

where the last inequality is from $\sigma_{(\tilde{\mathcal{P}}^{s,u})_x^a}(V^{s,u^*}) = \sigma_{(\tilde{\mathcal{P}}^{s,u^*})_s^a}(V^{s,u^*})$ for any $x, a$. Hence we have that

$$\|V^{s,u} - V^{s,u^*}\| \leq \frac{1}{(1-\gamma)N}. \tag{181}$$

We further have that

$$|\mathsf{P}_s^a V_{\tilde{\mathcal{P}}}^{\tilde{\pi}} - \hat{\mathsf{P}}_s^a V_{\tilde{\mathcal{P}}}^{\tilde{\pi}}|$$
$$\leq |\mathsf{P}_s^a V^{s,u} - \hat{\mathsf{P}}_s^a V^{s,u}| + |(\mathsf{P}_s^a - \hat{\mathsf{P}}_s^a)(V^{s,u} - V^{s,u^*})|$$
$$\leq \frac{2}{(1-\gamma)N} + \frac{c_2 \log(\frac{4SAN^2}{\delta})}{(1-\gamma)N(s,a)} + \sqrt{\frac{c_1 \log(\frac{4SAN^2}{\delta}) \mathbf{Var}_{\mathsf{P}_s^a}(V^{s,u})}{N(s,a)}}$$
$$= \frac{2}{(1-\gamma)N} + \frac{c_2 \log(\frac{4SAN^2}{\delta})}{(1-\gamma)N(s,a)} + \sqrt{\frac{c_1 \log(\frac{4SAN^2}{\delta})(\mathbf{Var}_{\mathsf{P}_s^a}(V^{s,u^*}) + (\mathbf{Var}_{\mathsf{P}_s^a}(V^{s,u}) - \mathbf{Var}_{\mathsf{P}_s^a}(V^{s,u^*})))}{N(s,a)}}$$

$$\leq \frac{2}{(1-\gamma)N} + \frac{c_2 \log(\frac{4SAN^2}{\delta})}{(1-\gamma)N(s,a)} + \sqrt{\frac{c_1 \log(\frac{4SAN^2}{\delta})\mathbf{Var}_{\mathsf{P}_s^a}(V^{s,u^*})}{N(s,a)}}$$

$$+ \sqrt{\frac{c_1 \log(\frac{4SAN^2}{\delta})|\mathbf{Var}_{\mathsf{P}_s^a}(V^{s,u}) - \mathbf{Var}_{\mathsf{P}_s^a}(V^{s,u^*})|}{N(s,a)}}$$

$$\leq \frac{2}{(1-\gamma)N} + \frac{c_2 \log(\frac{4SAN^2}{\delta})}{(1-\gamma)N(s,a)} + \sqrt{\frac{c_1 \log(\frac{4SAN^2}{\delta})\mathbf{Var}_{\mathsf{P}_s^a}(V^{s,u^*})}{N(s,a)}}$$

$$+ \sqrt{\frac{2c_1 \log(\frac{4SAN^2}{\delta})}{N(s,a)(1-\gamma)^2}}, \tag{182}$$

where the last inequality is from

$$|\mathbf{Var}_q(V_1) - \mathbf{Var}_q(V_2)|$$
$$= |q(V_1 \circ V_1) - (qV_1) \circ (qV_1) - q(V_2 \circ V_2) + (qV_2) \circ (qV_2)|$$
$$\leq |q(V_1 \circ V_1 - V_2 \circ V_2)| + |(qV_1 + qV_2)q(V_1 - V_2)|$$
$$\leq 2\|V_1 + V_2\|\|V_1 - V_2\|$$
$$\leq \frac{2\|V_1 - V_2\|}{1-\gamma}. \tag{183}$$

Thus (182) can be further bounded as

$$|\mathsf{P}_s^a V_{\tilde{\mathcal{P}}}^{\tilde{\pi}} - \hat{\mathsf{P}}_s^a V_{\tilde{\mathcal{P}}}^{\tilde{\pi}}|$$

$$\leq \frac{2}{(1-\gamma)N} + \frac{c_2 \log(\frac{4SAN^2}{\delta})}{(1-\gamma)N(s,a)} + \sqrt{\frac{c_1 \log(\frac{4SAN^2}{\delta})\mathbf{Var}_{\mathsf{P}_s^a}(V^{s,u^*})}{N(s,a)}}$$

$$+ \sqrt{\frac{2c_1 \log(\frac{4SAN^2}{\delta})}{N(s,a)(1-\gamma)^2}}$$

$$\leq \frac{1}{N(s,a)(1-\gamma)} \left( 2 + c_2 \log\left(\frac{4SAN^2}{\delta}\right) + \sqrt{2c_1 \log(\frac{4SAN^2}{\delta})} \right)$$

$$+ \sqrt{\frac{c_1 \log\left(\frac{4SAN^2}{\delta}\right)\mathbf{Var}_{\mathsf{P}_s^a}(V^{s,u^*})}{N(s,a)}}. \tag{184}$$

We note that this inequality exactly matches (127) in [38], hence the following part directly follows, which is omitted here. $\qquad\square$

**Lemma 16.** *(Lemma 14 of [36]) For any $(s,a)$ satisfying $N(s,a) > 0$, and any vector $V \in \mathbb{R}^{|\mathcal{S}|}$ independent of $\hat{\mathsf{P}}_s^a$ obeying $\|V\| \leq \frac{1}{1-\gamma}$, with probability at least $1 - \delta$, it holds that*

$$|\sigma_{\mathcal{P}_s^a}(V) - \sigma_{\hat{\mathcal{P}}_s^a}(V)| \leq \frac{C_1}{R(1-\gamma)}\sqrt{\frac{\log \frac{NS}{\delta}}{N(s,a)\min_x \hat{\mathsf{P}}_{s,x}^a}}. \tag{185}$$

**Lemma 17.** *For the KL-divergence uncertainty set, and for any $(s,a)$ satisfying $N(s,a) > 0$, with probability at least $1 - \delta$, it holds that*

$$|\sigma_{\mathcal{P}_s^a}(V_{\tilde{\mathcal{P}}}^{\tilde{\pi}}) - \sigma_{\hat{\mathcal{P}}_s^a}(V_{\tilde{\mathcal{P}}}^{\tilde{\pi}})| \leq \min\left\{ \frac{1}{1-\gamma}, \frac{4}{NR(1-\gamma)} + \frac{C_1}{R(1-\gamma)}\sqrt{\frac{\log \frac{2(1+R)N^3 S}{(1-\gamma)\delta}}{N(s,a)\min_x \hat{\mathsf{P}}_{s,x}^a}} \right\}. \tag{186}$$

*Proof.* **Step 1.** We first construct a robust MDP with state-absorbing empirical nominal transition kernels. More specifically, for each state $s$ and a constant $u \geq 0$, define the following transition kernels for any $a \in \mathcal{A}, x \in \mathcal{S}$:

$$(\mathsf{P}^{s,u})_{s,x}^a = \mathbf{1}_{s=x}, \tag{187}$$

$$(\mathsf{P}^{s,u})^a_{s,x} = \hat{\mathsf{P}}^a_{s,x}, \text{ if } x \neq s. \tag{188}$$

Moreover, define the modified reward function:

$$r^{s,u}(s,a) = u, \tag{189}$$
$$r^{s,u}(x,a) = r(x,a), \text{ if } x \neq s. \tag{190}$$

We then define the following uncertainty set $(\mathcal{P}^{s,u})^a_x$ for any $a \in \mathcal{A}, x \in \mathcal{S}$ as follows:

$$(\mathcal{P}^{s,u})^a_s = \{\mathbf{1}_s\}, \tag{191}$$
$$(\mathcal{P}^{s,u})^a_x = \{q \in \Delta(\mathcal{S}) : D(q\|(\mathsf{P}^{s,u})^a_x) \leq R + \kappa^a_x\}, \text{ if } x \neq s. \tag{192}$$

The auxiliary robust MDP is then defined as $M^{s,u} = (\mathcal{P}^{s,u} = \bigotimes_{s,a}(\mathcal{P}^{s,u})^a_s, r^{s,u})$.

**Step 2.** We next show that if we set $u^* = (1-\gamma)V^{\tilde{\pi}}_{\tilde{\mathcal{P}}}(s)$, then the optimal robust value function of $M^{s,u^*}$ is identical to $V^{\tilde{\pi}}_{\tilde{\mathcal{P}}}$.

For any state $x \neq s$ and action $b$, it can be verified from the definitions that $(\mathcal{P}^{s,u})^b_x = \tilde{\mathcal{P}}^b_x$, and hence

$$\max_b\{r^{s,u^*}(x,b) + \gamma\sigma_{(\mathcal{P}^{s,u^*})^b_x}(V^{\tilde{\pi}}_{\tilde{\mathcal{P}}})\} = \max_b\{r(x,b) + \gamma\sigma_{\tilde{\mathcal{P}}^b_x}(V^{\tilde{\pi}}_{\tilde{\mathcal{P}}})\} = V^{\tilde{\pi}}_{\tilde{\mathcal{P}}}(x). \tag{193}$$

For state $s$, we have that

$$\max_b\{r^{s,u^*}(s,b) + \gamma\sigma_{(\mathcal{P}^{s,u^*})^b_s}(V^{\tilde{\pi}}_{\tilde{\mathcal{P}}})\} = \max_b\{u^* + \gamma V^{\tilde{\pi}}_{\tilde{\mathcal{P}}}(s)\} = V^{\tilde{\pi}}_{\tilde{\mathcal{P}}}(s). \tag{194}$$

Thus we verified that $V^{\tilde{\pi}}_{\tilde{\mathcal{P}}}$ is the fixed point of the robust Bellman operator of $M^{s,u}$, and hence identical to the optimal robust value function of $M^{s,u}$.

**Step 3.** Define the set $\mathcal{U} \triangleq \{\frac{i}{N} : 1 \leq i \leq N\}$. The set $\mathcal{U}$ is then a $\frac{1}{N}$-net of the interval $[0,1]$. Since it is clear that $u^* \leq 1$, there exists $u_0 \in \mathcal{U}$, such that $|u_0 - u^*| \leq \frac{1}{N}$.

On the other hand, for any $u \in \mathcal{U}$, since $u \leq 1$, the optimal robust value function $\|V^{s,u}\| \leq \frac{1}{1-\gamma}$; Moreover, from the construction, the uncertainty set $\mathcal{P}^{s,u}$ is independent of $\hat{\mathsf{P}}^a_s$, hence invoking Lemma 16 implies

$$|\sigma_{\mathcal{P}^a_s}(V^{s,u}) - \sigma_{\hat{\mathcal{P}}^a_s}(V^{s,u})| \leq \frac{C_1}{R(1-\gamma)}\sqrt{\frac{\log\frac{NS}{\delta}}{N(s,a)\min_x\hat{\mathsf{P}}^a_{s,x}}}, \tag{195}$$

with probability at least $1 - \delta$. $\qquad\square$

**Step 4.** We further show the following claim:

$$\|V^{s,u_0} - V^{s,u^*}\| \leq \frac{1}{N(1-\gamma)}. \tag{196}$$

To show (196), for $s$, we have that

$$|V^{s,u_0}(s) - V^{s,u^*}(s)| \overset{(a)}{\leq} \max_b|(u_0 - u^*) + \gamma(\sigma_{(\mathcal{P}^{s,u_0})^b_s}(V^{s,u_0}) - \sigma_{(\mathcal{P}^{s,u^*})^b_s}(V^{s,u^*}))|$$

$$\overset{(b)}{\leq} \frac{1}{N} + \gamma|V^{s,u_0}(s) - V^{s,u^*}(s)|$$

$$\leq \frac{1}{N} + \gamma\|V^{s,u_0}(s) - V^{s,u^*}(s)\|, \tag{197}$$

where $(a)$ is from $|\max f - \max g| \leq \max|f - g|$, and $(b)$ is from the fact that $(\mathcal{P}^{s,u})^b_s = \{\mathbf{1}_s\}$ for any $u$ and $b$.

And for state $x \neq s$, we have that

$$|V^{s,u_0}(x) - V^{s,u^*}(x)| \leq \max_b|r^{s,u_0}(x,b) - r^{s,u^*}(x,b) + \gamma(\sigma_{(\mathcal{P}^{s,u_0})^b_x}(V^{s,u_0}) - \sigma_{(\mathcal{P}^{s,u^*})^b_x}(V^{s,u^*}))|$$

$$\overset{(a)}{\leq} \gamma\|V^{s,u_0} - V^{s,u^*}\|, \tag{198}$$

where $(a)$ is from $(\mathcal{P}^{s,u})^b_x = \tilde{\mathcal{P}}^b_x$ and $r^{s,u_0}(x,b) = r^{s,u^*}(x,b)$ for any $b \in \mathcal{A}$ and $x \neq s$.

Hence together we proved (196), which is identical to (240) of [36]. The remaining proof hence follows exactly the same as [36].

