# OpenReview forum: "A Unified Principle of Pessimism for Offline Reinforcement Learning under Model Mismatch"
_NeurIPS.cc/2024/Conference — NeurIPS 2024 poster_

### Official Review · Reviewer_3D3w · 2024-07-11

**Soundness:** 3
**Presentation:** 3
**Contribution:** 3
**Rating:** 6
**Confidence:** 3

**Summary:**

This paper proposes an algorithm that tackles the offline RL task (for tabular state and action space) under two key challenges (i) the mismatch between the environment dynamics used to generate the dataset and the environment used to run the learned policy (for evaluation), and (ii) the bias in the data-collection procedure where for certain (s, a) pairs there are only a few samples in the dataset. In contrast to prior approaches that tackle the two problems separately, this paper proposes a unified solution to both problems under the distributionally robust MDP framework.

Intuitively, distributionally robust MDP aims to learn a policy/value function with the best possible expected return given the worst possible environment dynamics in a set of distributions. Specifically, the set contains all distributions that are close (i.e., smaller than R) under some metrics (this paper considered total variation, chi-squared divergence, and KL divergence). This paper also takes into account the scarcity of data for certain (s, a) pairs by changing the fixed bound R to an adaptive bound R + K_{s}^{a}, where K_{s}^{a} depends on the visit count of (s, a) in the offline dataset.

With this new formulation, the authors derive high-probability lower bounds of the optimality of the learned value functions, under three divergences: total variation, chi-squared divergence, and KL divergence). The bounds appear to be tighter than that in prior work.

**Strengths:**

This paper proposes a theoretically useful offline RL algorithm that can handle model misspecification errors and data scarcity in a unified perspective. Specifically, it turns the task into a canonical algorithm for distributionally robust MDP. This can significantly simplify the analysis and inspire future work to find better connections between the two challenges in offline RL.

The paper contains a thorough introduction and comparison with prior work, which makes it much easier to see the contributions of this paper.

**Weaknesses:**

One major weakness of the proposed algorithm is its practicality. The proposed algorithm relies on tabular state and action representations and involves solving robust Bellman equations which could be prohibitively hard in practice. Also, to compute certain statistics, we need to record the state-action visitation count, which is impractical in practice.

It would be nice if the author can demonstrate potential ways to apply the algorithm in practice. For example, can the algorithm be used together with function approximations?

**Questions:**

Does the proposed method achieve better empirical performance on tabular environments compared to baselines?

Are there potential ways to make the algorithm more practical?

**Limitations:**

The authors discussed certain limitations in Section 6.

---

> ### Author Rebuttal · Authors · 2024-08-05
>
> We thank the reviewer for the helpful and insightful feedback. In the following, we provide point-to-point responses to the weaknesses and questions.
>
>
>
> **Question 1. Empirical performance in tabular environments** We first refer the reviewer to Section A of the appendix, where we compare our algorithms with two baselines. Additional results for large environments are provided in the rebuttal PDF, Fig. 2. Our method outperforms the non-robust method and demonstrates performance similar to the LCB approach [35], consistent with our theoretical results. Furthermore, as shown in Fig. 1 of the rebuttal PDF, in the tabular setting, our algorithms have a lower execution time and computational complexity than the LCB baseline. Hence we claim that our algorithms perform better than the baselines under tabular environments.
>
> It is also important to note that there is no practical implementation of the baseline [5] due to the NP-hardness of their models. Consequently, we did not include a comparison with [5] (please also see our response to Reviewer 2 Weakness 2).
>
>
>
> **Weakness \& Q2. Extend to practical settings.** We want to emphasize that our major contribution is to develop a methodological framework for offline RL with model mismatch, which incorporates both uncertainties from the limited dataset and model mismatch as a **single** distributional uncertainty set and tackles them through a unified DRO framework. To better illustrate our novelty and technique contributions, we present our results in the tabular setting, where we show the sample complexity of our algorithms improves upon or matches the SOTA.
>
> However, our method can be easily extended to large-scale problems with function approximation. For example, in d-rectangular robust linear MDPs [26,5], the nominal transition kernel is the inner product of some d-dimensional feature functions $\phi$ and a d-dim vector $\theta$: $\mathsf{P}^a_{s,s'}=\phi(s,a)^\top \theta_{s'}$. The uncertainty set for the model mismatch is set as $\mathcal{P}=\\{(\phi(s,a)^\top \tau_{s'}): D(\theta_i,\tau_i)\leq R, i\leq d\\}$. Our method can be adapted to solve offline robust linear MDP problems. Specifically, after estimating the parameter $\hat{\theta}$ from the offline dataset as in [5], we can design a similar enlarged uncertainty set $\tilde{\mathcal{P}}=\\{(\phi(s,a)^\top \tau_{s'})_{s,a,s'}: D(\hat{\theta}_i,\tau_i)\leq R+\kappa_i \\}$, where $\kappa_i$ is some enlarged term that accounts for the uncertainty of the dataset. Hence our method will result in a standard robust linear MDP formulation, which can be solved as in [26].
>
> Our method can be further extended to robust MDPs with latent structure or function approximation. Specifically, the nominal transition kernel can be represented or approximated by some parameter function $\mathsf{P}\approx f_\theta$, and the robust MDP for the model mismatch will be set as $\mathcal{P}=\\{ f_{\theta'}: D(\theta,\theta')\leq R\\}$. Similarly, after estimating the parameter $\hat{\theta}$ from the dataset and designing a dataset dependent term $\kappa$, solving the enlarged robust MDP $\mathcal{P}=\\{ f_{\theta'}: D(\hat{\theta},\theta')\leq R+\kappa\\}$ will result in a policy tackling both two sources of uncertainty. Also, it is worth mentioning that such a robust MDP can be solved empirically using adversarial training approaches, as in [a,b].
>
>
> Another extension to make our method more practical is to design model-free algorithms for our DRO formulation. Unlike the model-based approach presented in our paper, model-free algorithms do not require the model estimation or solving the Bellman equation but can directly learn the optimal policy in an online bootstrapping fashion, making them more suitable for large-scale problems. Existing robust Q-learning algorithms, e.g., [c,d], can be directly adapted by setting the uncertainty set radius to $R+\kappa$ and solving our formulation, which further makes our method more practical.
>
>
> To verify that our method can also be extended to practical settings, we further implement a model-free robust Q-learning as described in [c] to solve the American Option environment ([c], [64]) in the offline setting. For each dataset, we ran the robust Q-learning algorithm with the uncertainty set constructed as described in our paper and plotted the robust value function of the learned policy. The results, presented in Fig. 3 of the rebuttal PDF, demonstrate that our method can be combined with model-free algorithms to effectively solve offline problems under model mismatch, particularly for large-scale problems. We will add more discussion in our paper.
>
>
> [a] Rigter, Marc, et al. "Rambo-rl: Robust adversarial model-based offline reinforcement learning." Advances in neural information processing systems 35 (2022): 16082-16097.
>
> [b] Pinto, Lerrel, et al. "Robust adversarial reinforcement learning." International conference on machine learning. PMLR, 2017.
>
> [c] Liang, Zhipeng, et al. "Single-trajectory distributionally robust reinforcement learning." arXiv preprint arXiv:2301.11721 (2023).
>
> [d] Liu, Zijian, et al. "Distributionally Robust $ Q $-Learning." International Conference on Machine Learning. PMLR, 2022.

---

> > ### Comment · Reviewer_3D3w · 2024-08-12
> >
> > I thank the authors for their rebuttal and for the additional analysis using transition kernels as well as for discussing potential use cases of the proposed algorithm. I believe this strengthens the paper and will increase my score to 6.

---

> > > ### Author Response · Authors · 2024-08-12
> > >
> > > Thank you very much for your positive feedback on our submission!
> > >
> > > We’re glad to hear that you found our responses helpful, and we deeply appreciate your efforts and support. Thank you again!

---

### Official Review · Reviewer_TydZ · 2024-07-12

**Soundness:** 3
**Presentation:** 3
**Contribution:** 3
**Rating:** 7
**Confidence:** 3

**Summary:**

The authors propose a unified principle of pessimism using distributionally robust Markov decision processes (MDPs) to handle both data sparsity and model mismatch. They construct a robust MDP with a single uncertainty set and demonstrate that the optimal robust policy achieves a near-optimal sub-optimality gap under the target environment across three uncertainty models: total variation, χ2 divergence, and KL divergence. The proposed approach improves or matches state-of-the-art performance under these models and provides the first result for the χ2 divergence model.

**Strengths:**

- The paper is well-structured and clearly written, making it easy to follow the main arguments and methodologies.
- The paper provides detailed theoretical analysis and guarantees, showing that the proposed method achieves near-optimal performance.
- The proposed framework is versatile as well as easier to implement.

**Weaknesses:**

- While the theoretical guarantees are strong, the empirical validation could be more comprehensive. The experiments are conducted on relatively simple tasks (Frozen-Lake and Gambler problems). It would be beneficial to see results on more complex and diverse benchmarks.
- Computational Cost: The paper claims that the proposed method has a lower computational cost compared to existing methods, but this claim is not empirically validated. A comparison of computational costs (e.g., runtime) with other methods would strengthen this claim.

**Questions:**

To summarize the main points/questions raised in the weaknesses section:

- Could the authors provide empirical results on more complex tasks to validate the effectiveness of their method?
- Could the authors compare the computational costs of their method with other existing methods?

---

> ### Author Rebuttal · Authors · 2024-08-05
>
> We thank the reviewer for the helpful and insightful feedback. In the following, we provide point-to-point responses to the weaknesses and questions.
>
> **Weakness 1. Experiment results on more complex and diverse benchmarks.**
> We further provide two more experiments on large-scale environments in the rebuttal PDF. We implement our algorithm with the two baselines under the KL divergence uncertainty set.
>
> In this rebuttal, we further add the results under the Garnet problem G(64,16) [a] in Fig 2(a), where there are 64 states and 16 actions. The transition kernels and reward functions are generated following some Gaussian distribution $\mathcal{N}(\mu_{s,a},\sigma_{s,a})$. The comparison under the N-Chain problem is presented in Fig 2(b). And the results under the Cartpole problem are in Fig 2(c).  We also implement a model-free algorithm under the American Option problem in Fig 3, where no model estimation is required, indicating the potential of our method in solving large-scale problems.
>
>
> As the results show, our method performs similarly to the LCB algorithm, aligning with our theoretical results. Moreover, our approach also shows a great improvement compared to the non-robust method, which verifies the robustness of our method.
>
>
>
>
> We will include more experiments under different environments and uncertainty sets in the final version.
>
>
>
>
>
> **Weakness 2. Computational cost.** Our algorithms are better in terms of computational complexity or practical implementations than both baselines.
>
> The two-layer optimization problem in [5] is an extension of the model studied in [41] to the robust setting, both involving non-rectangular uncertainty sets that are NP-hard to solve [49]. This creates uncertainty regarding the solvability of their models. Specifically, due to the unsolvability of the non-robust model in [41], an adversarial training-based algorithm is designed in [b] with only an experimental convergence guarantee, highlighting the implementation challenges of [5]. In contrast, our algorithm can be implemented with polynomial complexity. Specifically, the total computational complexity of our algorithm under the TV, CS, and KL models is shown in [13] to be $\mathcal{O}(S^2A\log S)$,$\mathcal{O}(S^2A\log S)$, and $\tilde{\mathcal{O}}(S^2A)$.
>
> Compared to [35], our algorithms also offer better computational complexity. Specifically, the penalty term in [35] requires a complicated computation involving a minimum operator, resulting in an additional max-min structure in their algorithm update. The comparison/max-min operator in the LCB algorithm is executed $SA$ times per step, significantly increasing computational complexity. In contrast, our algorithms have a simple structure and do not require additional operators, making them more computationally efficient.
>
>
> To illustrate our computational efficiency, we include numerical experiments in the rebuttal PDF. We implemented the LCB algorithm from [35] and our DRO algorithm under the KL model, monitoring the execution time of both methods while learning the same policy from the same dataset. We plotted the execution time for each dataset versus the size of the dataset in three environments: Gambler's game, Frozen Lake, and N-chain. As shown in Fig. 1, our algorithm consistently requires less execution time across all three environments, demonstrating lower computational complexity.
>
> Therefore, our methods exhibit superior computational complexity compared to both baselines.
>
>
> [a] Archibald, T. W., et al. "On the generation of markov decision processes." Journal of the Operational Research Society 46.3 (1995): 354-361.
> [b] Rigter, Marc, et al. "Rambo-rl: Robust adversarial model-based offline reinforcement learning." Advances in neural information processing systems 35 (2022): 16082-16097.

---

> > ### Comment · Reviewer_TydZ · 2024-08-11
> >
> > Thank you for your reply. I still appreciate the contributions of the work and I am in favor of the acceptance of the paper. I have no further questions and will remain my positive score.

---

> > > ### Author Response · Authors · 2024-08-12
> > >
> > > Thank you very much for your positive feedback on our submission!
> > >
> > > We’re glad to hear that you found our responses helpful, and we deeply appreciate your efforts and support. Thank you again!

---

### Official Review · Reviewer_cT4V · 2024-07-12

**Soundness:** 3
**Presentation:** 2
**Contribution:** 2
**Rating:** 6
**Confidence:** 3

**Summary:**

This paper studies offline reinforcement learning (RL) under model mismatch. The authors propose a unified distributionally robust optimization (DRO) framework that effectively tackles both the uncertainty from limited dataset coverage and the model mismatch between the training and deployment environments.

**Strengths:**

A Unified DRO Formulation: The authors provide unified results for robust Markov Decision Processes (MDPs) with several types of robust sets.

The authors have carefully designed the uncertainty set radius, incorporating both model mismatch and data sparsity factors. The tight theoretical guarantees provided for three widely studied uncertainty set models (total variation, χ2 divergence, and Kullback-Leibler divergence) demonstrate the robustness and versatility of their approach.

The authors present the first finite-sample analysis for the χ2 divergence uncertainty set in the offline reinforcement learning (RL) setting with model mismatch. This contribution advances our understanding of this particular model and its applications in robust RL.

The paper is well-structured, with a clear problem formulation, comprehensive algorithmic description, and rigorous theoretical analysis.

**Weaknesses:**

It would be beneficial if you could elaborate on the advantages of your work compared to [35] and [5], especially regarding sample complexity under KL and total variation settings. A more detailed comparison would help readers fully appreciate the advancements your method brings to the field.

Given that [5] presents another unified framework for offline distributionally robust RL, it would be interesting to know if your framework can be extended to the χ2 case. If this extension is not possible, highlighting this limitation could actually strengthen your paper by emphasizing the unique contributions of your approach.

Addressing the computational efficiency of your proposed algorithm, particularly in comparison to [35], would be valuable. If your method is computationally efficient, including experimental results similar to those in [35] could provide empirical support for your theoretical findings.

**Questions:**

na

**Limitations:**

The authors have discussed the limitations in Section 6.

---

> ### Author Rebuttal · Authors · 2024-08-05
>
> We thank the reviewer for the helpful and insightful feedback. In the following, we provide point-to-point responses to the weaknesses and questions.
>
> **Weakness 1. Advantages compared to [5] and [35].**
> In summary, our method has three major advantages compared to both baselines. Firstly, our work is applicable under three different uncertainty set models, whereas [35] is limited to the KL model and [5] addresses only the KL and TV models (please see Weakness 2 for further discussion). Secondly, our algorithms achieve better sample complexity results compared to [5], and match the SOTA complexity under the KL model in [35] (please see discussion in Weakness 1-1). Finally, our algorithms demonstrate superior computational complexity compared to both baselines, and are the most efficient (please also see Weakness 3).
>
> These improvements further strengthens our paper, highlighting our novelty and contributions.
>
>
> **Weakness 1-1. Comparison of sample complexity with [5] and [35].**
> We refer to Table 1 on page 8 for a comparison of the sample complexity results between ours and the two baselines. A more detailed comparison is provided below.
>
> Compared with [5], our methods and results achieve better sample complexity in both KL and TV models. In the TV model, our sample complexity outperforms [5] in terms of dependence on $S$ and $(1-\gamma)$; For the KL model, our complexity is linearly dependent on $S$, while [5] has a quadratic dependence. Furthermore, as noted in [5], their result's exponential term can be replaced by utilizing both $P_{\min}$ and $\mu_{\min}$, while our (asymptotic) complexity result depends solely on $P_{\min}$.
>
> Our results match exactly those of [35] in the KL model, indicating that we are achieving SOTA. Additionally, we provide results for two other models, which cannot be obtained directly using the algorithm in [35] (also see Weakness 2).
>
>
>
>
> **Weakness 2. Extension of baselines to other models.**
> Although extending the method in [5] may seem straightforward at first glance, the concrete extension requires additional effort. More importantly, the extended sample complexity result under the CS uncertainty set is also expected to be proportional to $S^2$, similar to the results under the other two models, whereas our method exhibits a linear dependence on $S$. This difference arises because the method in [5] is distribution-based, designing the first-level uncertainty set to cover the true nominal kernel, which results in worse complexity than ours (a more detailed discussion can be found in  [46]).
>
> In [35], the LCB-based algorithm is only designed for the KL model and requires additional efforts to extend to other models. Specifically, a careful study of the optimal solution to the corresponding DRO problem is needed to design the penalty term, ensuring a pessimistic estimation of the robust value functions. In contrast, our method can be directly applied from a uniform framework under all three uncertainty sets.
>
>
>
>
> **Weakness 3. Computational complexity comparison.**
> Our algorithms enjoy better computational complexity than both baselines.
>
> The two-layer optimization problem in [5] is an extension of the model studied in [41] to the robust setting, both involving non-rectangular uncertainty sets that are NP-hard to solve [49]. This creates uncertainty regarding the solvability of their models. Specifically, due to the unsolvability of the non-robust model in [41], an adversarial training-based algorithm is designed in [a] with only an experimental convergence guarantee, highlighting the implementation challenges of [5]. In contrast, our algorithm can be implemented with polynomial complexity. Specifically, the total computational complexity of our algorithm in the TV, CS and KL models is shown in [13] to be $\mathcal{O}(S^2A\log S)$,$\mathcal{O}(S^2A\log S)$, and $\tilde{\mathcal{O}}(S^2A)$.
>
> Compared to [35], our algorithms also offer better computational complexity. Specifically, the penalty term in [35] requires a complicated computation involving a minimum operator, resulting in an additional max-min structure in their algorithm update. The comparison/max-min operator in the LCB algorithm is executed $SA$ times per step, significantly increasing computational complexity. In contrast, our algorithms have a simple structure and do not require additional operators, making them more computationally efficient.
>
>
> To illustrate our computational efficiency, we include numerical experiments in the rebuttal PDF. We implemented the LCB algorithm from [35] and our DRO algorithm under the KL model, monitoring the execution time of both methods while learning the same policy from the same dataset. We plotted the execution time for each dataset versus the size of the dataset in three environments: Gambler's game, Frozen Lake, and N-chain. As shown in Fig. 1, our algorithm consistently requires less execution time across all three environments, demonstrating lower computational complexity.
>
> Therefore, our methods exhibit superior computational complexity compared to both baselines.
>
>
>
>
> [a] Rigter, Marc, et al. "Rambo-rl: Robust adversarial model-based offline reinforcement learning." Advances in neural information processing systems 35 (2022): 16082-16097.

---

> > ### Comment · Reviewer_cT4V · 2024-08-13
> >
> > Thank you for your response. My concerns have been addressed, and I would like to raise my score.

---

> > > ### Author Response · Authors · 2024-08-13
> > >
> > > Thank you very much for your positive feedback on our submission!
> > >
> > > We’re glad to hear that you found our responses helpful, and we deeply appreciate your efforts and support. Thank you again!

---

> ### Author Response · Authors · 2024-08-12
>
> Thank you again for reviewing our paper!!
>
> As the discussion period ends soon, we would like to ask if you find our rebuttal and response useful to address your questions and concerns. If not, we are more than willing to clearify them for you!
>
> Thank you very much for your time and support!

---

### Author Rebuttal · Authors · 2024-08-05

We thank the three reviewers for the helpful and insightful feedback. Besides the point-to-point responses, we provide some additional numerical results in the PDF, and a few responses to address some common questions.


**Compare with baselines [5], [35].**
Our method has three major advantages compared to both baselines.  **(1). Improved Sample Complexity.** Our algorithms achieve better sample complexity results compared to [5] (under TV and KL models), and match the SOTA complexity under the KL model in [35]. **(2). Higher Computational Efficiency.** Our method is more computationally efficient compared to both baselines. Specifically, the approach in [5] is NP-hard, whereas ours can be implemented in polynomial time. Additionally, due to the complexity of the penalty term design in [35], our algorithms demonstrate superior computational efficiency. We developed three numerical experiments to demonstrate this, as shown in Figure 1. **(3). Broader Applicability.** Our work is applicable under three different uncertainty set models, whereas [35] is limited to the KL model and [5] addresses only the KL and TV models. Extending their methods to other models requires additional effort.


**Experiments under more environments.**
We conducted three experiments with larger problem scales: the Garnet problem with 64 states and 16 actions, the N-Chain problem, and CartPole. The results, shown in Figure 2, indicate that our DRO-based approach achieves the best performance.


**Extension to practical settings.**
Since our main contribution is to develop a universal methodology to unify the two sources of uncertainty in offline RL with model mismatch, we mainly focus on the tabular setting to highlight our algorithm design and theoretical results in our paper. However, our method and framework can be easily extended to large scale problems with function approximation or model-free algorithm design (please also see our response to R3 for a detailed discussion).  We also conducted an experiment on the American option problem under the offline setting with model mismatch. Our method can be directly combined with model-free robust Q-learning to develop a more efficient and practical algorithm. The results, shown in Figure 3, demonstrate that our robust Q-learning outperforms the non-robust baseline, highlighting the potential of our method in practical applications.

---

### Author Response · Authors · 2024-08-10

Dear Reviewers,

As the discussion period has started for a few days, we would sincerely appreciate your review of our response to ensure that your concerns have been thoroughly addressed. If so, we kindly inquire if the reviewer could consider providing stronger support for the paper. Moreover, we are more than willing to address any further questions you may have. Thank you very much for your time and effort!

Best,

The Authors

---

### Decision · Program_Chairs · 2024-09-25

**Decision:**

Accept (poster)

**Comment:**

This paper has a solid algorithmic/theoretical contribution,